# Decentralized Randomly Distributed Multi-agent Multi-armed Bandit with Heterogeneous Rewards

**Mengfan Xu**[1]    **Diego Klabjan**[1]

[1]Department of Industrial Engineering and Management Sciences, Northwestern University
`MengfanXu2023@u.northwestern.edu, d-klabjan@northwestern.edu`

## Abstract

We study a decentralized multi-agent multi-armed bandit problem in which multiple clients are connected by time dependent random graphs provided by an environment. The reward distributions of each arm vary across clients and rewards are generated independently over time by an environment based on distributions that include both sub-exponential and sub-Gaussian distributions. Each client pulls an arm and communicates with neighbors based on the graph provided by the environment. The goal is to minimize the overall regret of the entire system through collaborations. To this end, we introduce a novel algorithmic framework, which first provides robust simulation methods for generating random graphs using rapidly mixing Markov chains or the random graph model, and then combines an averaging-based consensus approach with a newly proposed weighting technique and the upper confidence bound to deliver a UCB-type solution. Our algorithms account for the randomness in the graphs, removing the conventional doubly stochasticity assumption, and only require the knowledge of the number of clients at initialization. We derive optimal instance-dependent regret upper bounds of order $\log T$ in both sub-Gaussian and sub-exponential environments, and a nearly optimal mean-gap independent regret upper bound of order $\sqrt{T} \log T$ up to a $\log T$ factor. Importantly, our regret bounds hold with high probability and capture graph randomness, whereas prior works consider expected regret under assumptions and require more stringent reward distributions.

## 1 Introduction

Multi-armed Bandit (MAB) [Auer et al., 2002a,b] is an online sequential decision-making process that balances exploration and exploitation while given partial information. In this process, a single player (agent, client) aims to maximize a cumulative reward or, equivalently, minimize the cumulative loss, known as regret, by pulling an arm and observing the reward of that arm at each time step. The two variants of MAB are adversarial and stochastic MAB, depending on whether rewards are chosen arbitrarily or follow a time-invariant distribution, respectively. Recently, motivated by the development of federated learning [McMahan et al., 2017], multi-agent stochastic multi-armed bandit has been drawing increasing attention (commonly referred to as multi-agent MAB). In this variant, multiple clients collaboratively work with multiple stochastic MABs to maximize the overall performance of the entire system. Likewise, regret is an important performance measure, which is the difference between the cumulative reward of always pulling the global optimal arm by all clients and the actual cumulative reward gained by the clients at the end of the game, where global optimality is defined with respect to the average expected reward values of arms across clients. Thereafter, the question for each client to answer is essentially how to guarantee an optimal regret with limited observations of arms and insufficient information of other clients. Assuming the existence of a central server, also known as the controller, [Bistritz and Leshem, 2018, Zhu et al., 2021b, Huang et al., 2021, Mitra et al., 2021, Réda et al., 2022, Yan et al., 2022], allow a controller-client framework where the controller integrates and distributes the inputs from and to clients, adequately addressing the challenge

posed by the lack of information of other clients. However, this centralization implicitly requires all clients to communicate with one another through the central server and may fail to include common networks with graph structures where clients perform only pair-wise communications within the neighborhoods on the given graphs. Non-complete graphs capture the reality of failed communication links. Removing the centralization assumption leads to a decentralized multi-agent MAB problem, which is a challenging but attracting direction as it connects the bandit problem and graph theory, and precludes traditional centralized processing.

In the field of decentralized multi-agent MAB, it is commonly assumed that the mean reward value of an arm for different clients is the same, or equivalently, homogeneous. This assumption is encountered in [Landgren et al., 2016a,b, 2021, Zhu et al., 2020, Martínez-Rubio et al., 2019, Agarwal et al., 2022, Wang et al., 2022, 2020, Li and Song, 2022, Sankararaman et al., 2019, Chawla et al., 2020]. However, this assumption may not always hold in practical scenarios. In recent years, there has been an increasing emphasis on heterogeneous reward settings, where clients can retain different mean values for the rewards of the same arm. The transition to heterogeneous reward settings presents additional technical challenges. Clients are unable to infer the global optimal arms without sequential communications regarding the rewards of the same arm at other clients. Such communications, however, are limited by the partially observed rewards, as other clients may not pull the same arm, and constrained by the underlying graph structure. We study the heterogeneous setting with time varying graphs.

Traditionally, rewards are assumed to be sub-Gaussian distributed. However, there has been a recent focus on MAB with heavy-tailed reward distributions. This presents a non-trivial challenge as it is harder to concentrate reward observations in sublinear time compared to the light-tailed counterpart [Tao et al., 2022]. In the work of [Jia et al., 2021], sub-exponential rewards are considered and analyzed in the single-agent MAB setting with newly proposed upper confidence bounds. Meanwhile, for multi-agent MAB, heavy-tailed distributions are examined in a homogeneous setting in [Dubey et al., 2020]. However, the heterogeneous setting studied herein has not yet been formulated or analyzed, posing more challenges compared to the homogeneous setting, as discussed earlier.

Besides rewards, the underlying graph assumptions are essential to the decentralized multi-agent MAB problem, as increased communication among clients leads to better identification of global optimal arms and smaller regret. The existing works [Sankararaman et al., 2019, Chawla et al., 2020] relate regret with graph structures and characterize the dependency of regret on the graph complexity with respect to conductance. When considering two special cases, [Chawla et al., 2020] demonstrates the theoretical improvement achieved by circular ring graphs compared to complete graphs, and [Li and Song, 2022] numerically shows that the circular ring graph presents the most challenging scenario with the largest regret, while the complete graph is the simplest. There are two types of graphs from a time perspective: time-invariant graphs, which remain constant over time, and time-varying graphs, which depend on time steps and are more challenging but more general. Assumptions on time-invariant graphs include complete graphs [Wang et al., 2021] where all clients can communicate, regular graphs [Jiang and Cheng, 2023] where each client has the same number of neighbors, and connected graphs under the doubly stochasticity assumption [Zhu et al., 2020, 2021a,b]. Independently from our work, recent work [Zhu and Liu, 2023] has focused on time-varying $B$-connected graphs, where the composition of every $l$ consecutive graphs is a strongly connected graph. However, their doubly stochasticity assumption, where all elements of edge probability also called weight matrices are uniformly bounded by a positive constant, can be violated in several cases. Additionally, their graphs may be strongly correlated to meet the connectivity condition when $l > 1$, which may not always hold in practice. No research has been conducted on time-varying graphs with only connectivity constraints or without constraints on connectivity. Additionally, current time-varying graphs do not provide insight into how the graphs change over time. As the graphs are generated by the environment, similar to the generation of rewards, it remained unexplored considering random graphs in an i.i.d manner, such as random edge failures or random realizations as pointed out for future research in [Martínez-Rubio et al., 2019]. We also address this situation.

Traditionally, random graphs have often been formulated using the Erdős–Rényi (E-R) model, which has been widely adopted in various research domains. The model, described by $G(M, c)$, consists of $M$ vertices with each pair of vertices being connected with probability $c$. Notably, the E-R model is 1) not necessarily connected and 2) stochastic that allows for random edge failures, and has found applications in mean-field game [Delarue, 2017] and majority vote settings [Lima et al., 2008]. Though it has only been used in numerical experiments for the decentralized multi-agent MAB

setting with homogeneous rewards [Dubey et al., 2020], the theoretical study of this model in this context remained unexplored until this work, let alone with heterogeneous rewards and time-varying graphs. Alternatively, one can consider all connected graphs (there are exponentially many of them), and the environment can randomly sample a connected graph and produce i.i.d. samples of such random connected graphs. This approach mimics the behavior of stochastic rewards and allows the environment to exhaust the sample space of connected graphs independently, without the doubly stochasticity assumption, which, however, has not yet been studied and it is also addressed herein.

For the multi-agent MAB framework, methods in MAB are a natural extension. [Zhu et al., 2021b] adapt the UCB algorithm to the multi-agent setting. This algorithm uses weighted averages to achieve consensus among clients and is shown to have a regret of order $\log T$ for time-invariant graphs. A follow-up study in [Zhu and Liu, 2023] re-analyzes this algorithm for time-varying $B$-connected graphs under the aforementioned assumptions under the doubly stochasticity assumption by adding an additional term compared to UCB. An effective UCB-based method for random graphs without doubly stochasticity assumption and for sub-exponential distributed rewards remained unexplored.

This paper presents a novel contribution to the decentralized multi-agent MAB problem by studying both heterogeneous rewards and time-varying random graphs, where the distributions of rewards and graphs are independent of time. To the best of our knowledge, this is the first work to consider this problem and to investigate it with heavy-tailed reward distributions. Specifically, the paper investigates 1) heterogeneous sub-exponential and sub-Gaussian distributed rewards and 2) random graphs including the possibly disconnected E-R model and random connected graphs, and applies them to the decentralized multi-agent MAB framework. This work bridges the gap between large-deviation theories for sub-exponential distributions and multi-agent MAB with heterogeneous rewards, and the gap between random graphs and decentralized multi-agent MAB.

To this end, we propose a brand new algorithmic framework consisting of three main components: graph generation, DrFed-UCB: burn-in period, and DrFed-UCB: learning period. For the learning period, we modify the algorithm by [Zhu et al., 2021b] by introducing new UCB quantities that are consistent with the conventional UCB algorithm and generalize to sub-exponential settings. We also introduce a newly proposed stopping time and a new weight matrix without the doubly stochasticity assumption to leverage more information in random graphs. A burn-in period is crucial in estimating the graph distribution and initializing the weight matrix. We embed and analyze techniques from random graphs since the number of connected graphs is exponentially large in the number of vertices, and directly sampling such a graph is an NP-hard problem. In particular, we use the Metropolis-Hastings method with rapidly mixing Markov chains, as proposed in [Gray et al., 2019], to approximately generate random connected graphs in polynomial time. We additionally demonstrate its theoretical convergence rate, making it feasible to consider random connected graphs in the era of large-scale inference.

We present comprehensive analyses of the regret of the proposed algorithm, using the same regret definition as in existing literature. Firstly, we show that algorithm DrFed-UCB achieves optimal instance-dependent regret upper bounds of order $\log T$ with high probability, in both sub-Gaussian and sub-exponential settings, consistent with prior works. We add that although both [Zhu et al., 2020] and our analyses use the UCB framework, the important algorithmic steps are different and thus also the analyses. Secondly, we demonstrate that with high probability, the regret is universally upper bounded by $O(\sqrt{T}\log T)$ in sub-exponential settings, including sub-Gaussian settings. This upper bound matches the upper and lower bounds in single-agent settings up to a $\log T$ factor, establishing its tightness.

The paper is organized as follows. We first introduce the notations used throughout the paper, present the problem formulation, and propose algorithms for solving the problem. Following that, we provide theoretical results on the regret of the proposed algorithm in various settings.

## 2 Problem Formulation and Methodologies

### 2.1 Problem Formulation

Throughout, we consider a decentralized system with $M$ clients that are labeled as nodes $1, 2, \ldots, M$ on a time-varying network, which is described by an undirected graph $G_t$ for $1 \leq t \leq T$ where parameter $T$ denotes the time horizon of the problem. Formally, at time step $t$, $G_t = (V, E_t)$ where $V = \{1, 2, \ldots, M\}$ and $E_t$ denotes the edge set consisting of pair-wise nodes and representing the

neighborhood information in $G_t$. The neighbor set $\mathcal{N}_m(t)$ include all neighbors of client $m$ based on $G_t$. Equivalently, the graph $G_t$ can be represented by the adjacency matrix $(X_{i,j}^t)_{1 \leq i,j \leq M}$ where the element $X_{i,j}^t = 1$ if there is an edge between clients $i$ and $j$ and $X_{i,j}^t = 0$ otherwise. We let $X_{i,i} = 1$ for any $1 \leq i \leq M$. With this notation at hand, we define the empirical graph (adjacency matrix) $P_t$ as $P_t = \frac{(\sum_{s=1}^t X_{i,j}^s)_{1 \leq i,j \leq M}}{t}$. It is worth emphasizing that 1) the matrix $P_t$ is not necessarily doubly stochastic, 2) the matrix captures more information about $G_t$ than the prior works based on $|\mathcal{N}_m(t)|$, and 3) each client $m$ only knows the $m$-th row of $P_t$ without knowledge of $G_t$, i.e. $\{P_t(m,j)\}_j$ are known to client $m$, while $\{P_t(k,j)\}_j$ for $k \neq m$ are always unknown. Let us denote the set of all possible connected graphs on $M$ nodes as $\mathcal{G}_\mathcal{M}$.

We next consider the bandit problems faced by the clients. In the MAB setting, the environment generates rewards. Likewise, we again use the term, the environment, to represent the source of graphs $G_t$ and rewards $r_i^m(t)$ in the decentralized multi-agent MAB setting. Formally, there are $K$ arms faced by each client. At each time step $t$, for each client $1 \leq m \leq M$, let the reward of arm $1 \leq i \leq K$ be $r_i^m(t)$, which is i.i.d. distributed across time with the mean value $\mu_i^m$, and is drawn independently across the clients. Here we consider a heterogeneous setting where $\mu_i^m$ is not necessarily the same as $\mu_i^j$ for $m \neq j$. At each time step $t$, client $m$ pulls an arm $a_m^t$, only observes the reward of that arm $r_{a_m^t}^m(t)$ from the environment, and exchanges information with neighbors in $G_t$ given by the environment. In other words, two clients communicate only when there is an edge between them.

By taking the average over clients as in the existing literature, we define the global reward of arm $i$ at each time step $t$ as $r_i(t) = \frac{1}{M} \sum_{m=1}^M r_i^m(t)$ and the subsequent expected value of the global reward as $\mu_i = \frac{1}{M} \sum_{m=1}^M \mu_i^m$. We define the global optimal arm as $i^* = \arg\max_i \mu_i$ and arm $i \neq i^*$ is called global sub-optimal. Let $\Delta_i = \mu_{i^*} - \mu_i$ be the sub-optimality gap. This enables us to quantify the regret of the action sequence (policy) $\{a_m^t\}_{1 \leq m \leq M}^{1 \leq t \leq T}$ as follows. Ideally, clients would like to pull arm $i^*$ if knowledge of $\{\mu_i\}_i$ were available. Given the partially observed rewards due to bandits (dimension $i$) and limited accesses to information from other clients (dimension $m$), the regret is defined as $R_T = T\mu_{i^*} - \frac{1}{M} \sum_{t=1}^T \sum_{m=1}^M \mu_{a_m^t}^m$ which measures the difference of the cumulative expected reward between the global optimal arm and the action sequence. The main objective of this paper is to develop theoretically robust solutions to minimize $R_T$ for clients operating on time-varying random graphs that are vulnerable to random communication failures, which only require knowledge of $M$.

## 2.2 Algorithms

In this section, we introduce a new algorithmic framework that incorporates two graph generation algorithms, one for the E-R model and the other for uniformly distributed connected graphs. More importantly, the framework includes a UCB-variant algorithm that runs a learning period after a burn-in period, which is commonly referred to as a warm-up phase in statistical procedures.

### 2.2.1 Graph Generation

We investigate two types of graph dynamics as follows, for which we propose simulation methods that enable us to generate and analyze the resulting random graphs.

**E-R random graphs** At each time step $t$, the adjacency matrix of graph $G_t$ is generated by the environment by element-wise sampling $X_{i,j}^t$ according to a Bernoulli distribution. Specifically, $X_{i,j}^t$ follows a Bernoulli distribution with parameter $c$.

**Uniformly distributed connected graphs** At each time step $t$, the random graph $G_t$ is generated by the environment by uniformly sampling a graph from the sample space of all connected graphs $\mathcal{G}_M$, which yields the adjacency matrix $(X_{i,j}^t)_{1 \leq i \neq j \leq M}$ corresponding to $G_t$. Generating uniformly distributed connected graphs is presented in Algorithm 3 as in Appendix. It is computationally infeasible to exhaust the sample space $\mathcal{G}_M$ since the number of connected graphs is exponentially large. To this end, we import the Metropolis-Hastings method in [Gray et al., 2019] and leverage rapidly mixing Markov chains. Remarkably, by adapting the algorithm into our setting which yields a new algorithm, we construct a Markov chain that converges to the target distribution after a finite number of burn-in steps. This essentially traverses graphs in $\mathcal{G}_M$ through step-wise transitions, with a complexity of $O(M^2)$ from previous states. More precisely, at time step $s$ with connected graph $G_s$, we randomly choose a pair of nodes and check whether it exists in the edge set. If this is the case, we

remove the edge from $G_s$ and check whether the remaining graph is connected, and only accept the graph as $G_{s+1}$ in the connected case. If the edge does not exist in the edge set, we add it to $G_s$ and get $G_{s+1}$. In this setting, let $c = c(M)$ be the number of times an edge is present among all connected graphs divided by the total number of connected graphs. It is known that $c = 2\frac{\log M}{M-1}$ (1) [Trevisan]. The distribution of $G_s$ eventually converges to the uniform distribution in $\mathcal{G}_M$. The formal statements are in Appendix.

### 2.2.2 Main Algorithm

In the following, we present the proposed algorithm, DrFed-UCB, which comprises of a burn-in period and a learning period described in Algorithm 1 and Algorithm 2, respectively.

We start by introducing the variables used in the algorithm with respect to client $m$. We use $\bar{\mu}_i^m(t), n_{m,i}(t)$ to denote reward estimators and counters based on client $m$'s own pulls of arm $i$, respectively, and use $\tilde{\mu}_i^m, N_{m,i}(t)$ to denote reward estimators and counters based on the network-wide pulls of arm $i$, respectively. By network-wide, we refer to the clients in $\mathcal{N}_m(t)$. Denote the stopping time for the filtration $\{G_s\}_{s=1}^t$ as $h_{m,j}^t = max_{s \leq t}\{(m,j) \in E_s\}$; it represents the most recent communication between clients $m$ and $j$. The weights for the network-wide and local estimators are $P_t'(m,j)$ and $d_{m,t}$ defined later, respectively.

There are two stages in Algorithm 1 where $t \leq L$ as follows. For the first $\tau_1$ steps, the environment generates graphs based on one of the aforementioned graph generation algorithms to arrive at the steady state, while client $m$ pulls arms randomly and updates local estimators $\{\bar{\mu}_i^m, n_i^m\}_i$. Afterwards, the environment generates the graph $G_t$ that follows the distribution of interest, while client $m$ updates $\{\bar{\mu}_i^m, n_i^m\}_i$ and row $m$ of $P_t$ by exchanging information with its neighbors in the graph $G_t$. Note that client $m$ does not have any global information about $G_t$. At the end of the burn-in period, client $m$ computes the network-wide estimator $\tilde{\mu}_i^m$ by taking the weighted average of local estimators of other clients (including itself), where the weights are given by the $m$-th row of weight matrix $P'$ and $d$, which depend on $P$ and knowledge of $M$ and satisfy $\sum_{j=1}^M P_t'(m,j) + d_{m,t}M = 1$.

Subsequently, we describe Algorithm 2 where $t \geq L+1$. There are four phases in one iteration enumerated below in the order indicated. A flowchart of this algorithm is provided in Appendix.

**UCB**  Given the estimators $\tilde{\mu}_i^m(t), n_{m,i}(t), N_{m,i}(t), \bar{\mu}_i^m(t)$, client $m$ either randomly samples an arm or pulls the arm that maximizes the upper confidence bound using $\tilde{\mu}_i^m(t), n_{m,i}(t)$, depending on whether $n_{m,i}(t) \leq N_{m,i}(t) - K$ holds for some arm $i$. This additional condition ensures consensus among clients regarding which arm to pull. The upper confidence bound $F(m,i,t)$ is specified as $F(m,i,t) = \sqrt{\frac{C_1 \ln t}{n_{m,i}(t)}}$ and $F(m,i,t) = \sqrt{\frac{C_1 \ln T}{n_{m,i}(t)}} + \frac{C_2 \ln T}{n_{m,i}(t)}$ in settings with sub-Gaussian and sub-exponential rewards, respectively. Constants $C_1$ and $C_2$ are determined in the analyses and they depend on $\sigma$ which is an upper bound of standard deviations of the reward values (it is formally defined later).

**Environment and client interaction**  After client $m$ pulls arm $a_m^t$, the environment sends the reward $r_{m,a_m^t}^t$ and the neighbor set $\mathcal{N}_m(t)$ in $G_t$ to client $m$. Client $m$ does not know the whole $G_t$ and obtains only the neighbor set $\mathcal{N}_m(t)$.

**Transmission**  Client $m$ sends the maintained local and network-wide estimators $\{\bar{\mu}_i^m(t), \tilde{\mu}_i^m(t), n_{m,i}(t), N_{m,i}(t)\}_i$ to all clients in $\mathcal{N}_m(t)$ and receives the ones from them.

**Update estimators**  At the end of an iteration, client $m$ first updates the $m$-th row of matrix $P_t$. Subsequently, client $m$ updates the quantities $\{\bar{\mu}_i^m(t), \tilde{\mu}_i^m(t), n_{m,i}(t), N_{m,i}(t)\}_{1 \leq i \leq K}$ adhering to: $t_{m,j} = max_{s \geq \tau_1}\{(m,j) \in E_s\}$ and 0 if such an $s$ does not exist

$$n_{m,i}(t+1) = n_{m,i}(t) + \mathbb{1}_{a_m^t = i}, N_{m,i}(t+1) = \max\{n_{m,i}(t+1), \hat{N}_{i,j}^m(t), j \in \mathcal{N}_m(t)\} \quad (2)$$

$$\bar{\mu}_i^m(t+1) = \frac{\bar{\mu}_i^m(t) \cdot n_{m,i}(t) + r_{m,i}(t) \cdot \mathbb{1}_{a_m^t = i}}{n_{m,i}(t+1)}, P_t'(m,j) = \frac{M-1}{M^2} \text{ if } P_t(m,j) > 0 \text{ and } 0 \text{ otherwise}$$

$$\tilde{\mu}_i^m(t+1) = \sum_{j=1}^M P_t'(m,j)\hat{\tilde{\mu}}_{i,j}^m(t_{m,j}) + d_{m,t}\sum_{j \in N_m(t)} \hat{\tilde{\mu}}_{i,j}^m(t) + d_{m,t}\sum_{j \notin N_m(t)} \hat{\tilde{\mu}}_{i,j}^m(t_{m,j})$$

with $d_{m,t} = (1 - \sum_{j=1}^M P_t'(m,j))/M$

Similar to [Zhu et al., 2021b, Zhu and Liu, 2023], the algorithm balances between exploration and exploitation by the upper confidence bound and a criterion on $n_{m,i}(t)$ and $N_{m,i}(t)$ that ensures that all clients explore arms at similar rates and thereby "staying on the same page." After interacting

---

**Algorithm 1:** DrFed-UCB: Burn-in period

---

Initialization: The length of the burn-in period is $L$ and we are also given $\tau_1 < L$; In the time
 step $t = 0$, the estimates are initialized as $\bar{\mu}_i^m(0) = 0$, $n_{m,i}(0) = 0$, $\hat{\mu}_{i,j}^m(0) = 0$, and
 $P_0(m, j) =$ for any arm $i$ and clients $m, j$;

**for** $1 \le t \le \tau_1$ **do**
   | The environment generates a sample graph $G_t = (V, E_t)$ based on either E-R or Algorithm 3;
**end**

**for** $1 \le t \le \tau_1$ **do**
   | **for** *each client* $m$ **do**
      | | Sample arm $a_t^m = (t \mod K)$;
      | | Receive reward $r_{a_t^m}^m(t)$ and update $n_{m,i}(t) = n_{m,i}(t-1) + \mathbb{1}_{a_m^t = i}$;
      | | Update the local estimate for any arm $i$: $\bar{\mu}_i^m(t) = \frac{n_{m,i}(t-1)\bar{\mu}_i^m(t-1) + r_{a_t^m}^m(t) \cdot 1_{a_t^m = i}}{n_{m,i}(t-1) + 1_{a_t^m = i}}$;
   | **end**
**end**

**for** $\tau_1 < t \le L$ **do**
   | The environment generates a sample graph $G_t = (V, E_t)$ based on either E-R or Algorithm 3;
   | **for** *each client* $m$ **do**
      | | Sample arm $a_t^m = (t \mod K)$;
      | | Receive rewards $r_{a_t^m}^m(t)$ and update $n_{m,i}(t) = n_{m,i}(t-1) + \mathbb{1}_{a_m^t = i}$;
      | | Update the local estimates for any arm $i$: $\bar{\mu}_i^m(t) = \frac{n_{m,i}(t-1)\bar{\mu}_i^m(t-1) + r_{a_t^m}^m(t) \cdot 1_{a_t^m = i}}{n_{m,i}(t-1) + 1_{a_t^m = i}}$;
      | | Update the maintained matrix $P_t(m, j) = \frac{(t-1)P_{t-1}(m,j) + X_{m,j}^t}{t}$ for each $j \in V$;
      | | Send $\{\bar{\mu}_i^m(t)\}_{i=1}^{i=K}$ to all clients in $\mathcal{N}_m(t)$;
      | | Receive $\{\bar{\mu}_i^j(t)\}_{i=1}^{i=K}$ from all clients $j \in \mathcal{N}_m(t)$ and store them as $\hat{\mu}_{i,j}^m(t)$.
   | **end**
**end**

**for** *each client* $m$ *and arm* $i$ **do**
   | For client $1 \le j \le M$, set $h^L(m, j) = \max_{s \ge \tau_1}\{(m, j) \in E_s\}$ or 0 if such $s$ does not exist
   | $\tilde{\mu}_i^m(L+1) = \sum_{j=1}^M P'_{m,j}(L)\hat{\mu}_{i,j}^m(h_{m,j}^L)$ where $P'_{m,j}(L) = \begin{cases} \frac{1}{M} & \text{if } P_L(m,j) > 0 \\ 0 & \text{otherwise} \end{cases}$;
**end**

---

with the environment, clients move to the transmission stage, where they share information with the neighbors on $G_t$, as a preparation for the update stage.

Different from the upper confidence bound approach in [Zhu and Liu, 2023], which has an extra term of $\frac{1}{t}$ in the UCB criterion, our proposal is aligned with the conventional UCB algorithm. Meanwhile, our update rule differs from that in [Zhu et al., 2021b, Zhu and Liu, 2023] in three key aspects: (1) maintaining a stopping time $t_{m,j}$ that tracks the most recent communication to client $j$, and (2) updating $\tilde{\mu}_i^m$ based on both $\tilde{\mu}_i^j$ and $\bar{\mu}_i^j$ for $j \in \mathcal{N}_m(t)$, and (3) using a weight matrix based on $P'_t$ and $P_t$ computed from the trajectory $\{G_s\}_{s \le t}$ in the previous steps. The first point ensures that the latest information from other clients is leveraged, in case there is no communication at the current time step. The second point ensures proper integration of both network-wide and local information, smoothing out biases from local estimators and reducing variances through averaging. The third point distills the information carried by the time-varying graphs and determines the weights of the available local and network-wide estimators, removing the need for the doubly stochasticity assumption. The algorithm assumes that the clients know $M$ and $\sigma^2$.

We note that $t_{m,j}$ is the stopping time by definition and that $\bar{\mu}_i^m$ is an unbiased estimator for $\mu_i^m$ with a decaying variance proxy. Meanwhile, the matrices $P'_t$ and $P_t$ are not doubly stochastic and keep track of the history of the random graphs. By introducing $t_{m,j}$ and $P_t$, we can show that the global estimator $\tilde{\mu}_i^m(t)$ behaves similarly to a sub-Gaussian/sub-exponential random variable with an expectation of $\mu_i$ and a time-decaying variance proxy proportional to $\frac{1}{\min_j n_{j,i}(t)}$. This ensures that the concentration inequality holds for $\tilde{\mu}_i^m(t)$ with respect to $\mu_i$ and that client $m$ tends to identify the

**Algorithm 2:** DrFed-UCB: Learning period

---

Initialization: For each client $m$ and arm $i \in \{1, 2, \ldots, K\}$, we have $\tilde{\mu}_i^m(L+1)$,
  $N_{m,i}(L+1) = n_{m,i}(L)$; all other values at $L+1$ are initialized as 0;

**for** $t = L+1, L+2, \ldots, T$ **do**

    **for** *each client m* **do**                                                          // UCB

        **if** *there is no arm $i$ such that $n_{m,i}(t) \leq N_{m,i}(t) - K$* **then**

            $a_m^t = \arg\max_i \tilde{\mu}_{m,i}(t) + F(m,i,t)$

        **else**

            Randomly sample an arm $a_m^t$.

        **end**

        Pull arm $a_m^t$ and receive reward $r_{m,a_m^t}(t)$;

    **end**

    The environment generates a sample graph $G_t = (V, E_t)$

     based on E-R or Algorithm 3;                              // Env

    Each client $m$ sends $\mu_i^m(t), N_{j,i}(t), \bar{\mu}_i^m(t), \tilde{\mu}_i^m(t)$ to each client in $\mathcal{N}_m(t)$;

    Each client $m$ receives $\mu_i^j(t), N_{j,i}(t), \bar{\mu}_i^j(t), \tilde{\mu}_i^j(t)$ from all clients $j \in \mathcal{N}_m(t)$ and stores

     them as $\hat{\mu}_{i,j}^m(t), \hat{N}_{i,j}^m(t), \hat{\bar{\mu}}_{i,j}^m(t), \hat{\tilde{\mu}}_{i,j}^m(t)$;              // Transmission

    **for** *each client m* **do**

        **for** $i = 1, \ldots, K$ **do**

            Update $P_t$ for $1 \leq j \leq M$ by $P_t(m,j) = \frac{(t-1)P_{t-1}(m,j) + X_{m,j}^t}{t}$;

            Update $P_t'$ for $1 \leq j \leq M$ by $P_t'(m,j) = \begin{cases} 1 & \text{if } P_t(m,j) > 0 \\ 0 & \text{if } P_t(m,j) = 0 \end{cases}$;

            Update $n_{m,i}(t), N_{m,i}(t)$ and $\tilde{\mu}_i^m(t)$ based on equations (2);

        **end**

    **end**

**end**

---

global optimal arms with high probability, which plays an important role in minimizing regret. The formal statements are presented in the next section.

## 3 Regret Analyses

In this section, we show the theoretical guarantees of the proposed algorithm, assuming mild conditions on the environment. Specifically, we consider various settings with different model assumptions. We prove that the regret of Algorithm 2 has different instance-dependent upper bounds of order $\log T$ for settings with sub-Gaussian and sub-exponential distributed rewards, and a mean-gap independent upper bound of order $\sqrt{T} \log T$ across settings. Many researchers call such a bound instance independent but we believe such a terminology is misleading and thus we prefer to call it man-gap independent, given that it still has dependency on parameters pertaining to the problem instance. The results are consistent with the regret bounds in prior works.

### 3.1 Model Assumptions

By definition, the environment is determined by how the graphs (E-R or uniform) and rewards are generated. For reward we consider two cases.

**Sub-g** At time step $t$, the reward of arm $i$ at client $m$ has bounded support $[0, 1]$, and is drawn from a sub-Gaussian distribution with mean $0 \leq \mu_i^m \leq 1$ and variance proxy $0 \leq (\sigma_i^m)^2 \leq \sigma^2$.

**Sub-e** At time step $t$, the reward of arm $i$ at client $m$ has bounded support $[0, 1]$, and follows a sub-exponential distribution with mean $0 \leq \mu_i^m \leq 1$ and parameters $0 \leq (\sigma_i^m)^2 \leq \sigma^2, 0 \leq \alpha_i^m \leq \alpha$.

With these considerations, we investigate four different environments (settings) based on the two graph assumptions and the two reward assumptions: Setting 1.1 corresponds to E-R and Sub-g, Setting 1.2 to Uniform and Sub-g, Setting 2.1 to E-R and Sub-e, and Setting 2.2 to Uniform and Sub-e. For each setting, we derive upper bounds on the regret in the next section.

## 3.2 Regret Analyses

In this section, we establish the regret bounds formally when clients adhere to Algorithm 2 in various settings. We denote Setting 1.1, Setting 1.2 with $M < 11$, and Setting 1.2 with $M \geq 11$ as $s_1, s_2$ and $s_3$, respectively. Likewise, we denote Setting 2.1, Setting 2.2 with $M < 11$, and Setting 2.2 with $M \geq 11$ as $S_1, S_2$ and $S_3$, respectively. See Table 1 in Appendix for a tabular view of the various settings.

Note that the randomness of $R_T$ arises from both the reward and graph observations. Considering $S_1, S_2, S_3$ differ in the reward assumptions compared to $s_1, s_2, s_3$, we define an event $A$ that preserves the properties of the variables with respect to the random graphs. Given the length of the burn-in period $L_i$ for $i \in \{s_1, s_2, s_3\}$ and the fact that $L_{s_i} = L_{S_i}$ since it only relies on the graph assumptions, we use $L$ to denote $\max_i L_{s_i}$. Parameters $0 < \delta, \epsilon < 1$ are any constants, and the parameter $c = c(M)$ represents the mean value of the Bernoulli distribution in $s_1, S_1$ and the probability of an edge in $s_2, S_2, s_3$, and $S_3$ among all connected graphs (see (1)). We define events $A_1 = \{\forall t \geq L, \|P_t - cE\|_\infty \leq \delta\}, A_2 = \{\exists t_0, \forall t \geq L, \forall j, \forall m, t + 1 - \min_j t_{m,j} \leq t_0 \leq c_0 \min_l n_{l,i}(t+1)\}$, and $A_3 = \{\forall t \geq L, G_t \text{ is connected}\}$. Here $E$ is the matrix with all values of 1. Constant $c_0 = c_0(K, \min_{i \neq i^*} \Delta_i, M, \epsilon, \delta)$ is defined later. Since $c = c(M)$ this implies that $G_t$ depends on $M$. We define $A = A_{\epsilon,\delta} = A_1 \cap A_2 \cap A_3$, which yields $A \in \Sigma$ with $\Sigma$ being the sub-$\sigma$-algebra formed by $\{\Omega, \emptyset, A, A^c\}$. This implies $E[\cdot|A_{\epsilon,\delta}]$ and $P[\cdot|A_{\epsilon,\delta}]$ are well-defined, since $A$ only relies on the graphs and removes the differences among $s_1, s_2, s_3$ ($S_1, S_2, S_3$), enabling universal regret upper bounds.

Next, we demonstrate that event $A$ holds with high probability.

**Theorem 1.** *For event $A_{\epsilon,\delta}$ and any $1 > \epsilon, \delta > 0$, we have $P(A_{\epsilon,\delta}) \geq 1 - 7\epsilon$.*

*Proof Sketch.* The complete proof is deferred to Appendix; we discuss the main logic here. The proof relies on bounding the probabilities of $A_1, A_2, A_3$ separately. For $A_1$, its upper bound holds by the analysis of the mixing time of the Markov chain underlying $G_t$ and on the matrix-form Hoeffding inequality. We obtain an upper bound on $P(A_2)$ by analyzing the stopping time $t_{m,j}$ and the counter $n_{m,i}(t)$. For the last term $P(A_3)$, we show that the minimum degree of $G_t$ has a high probability lower bound that is sufficient for claiming the connectivity of $G_t$. To put all together, we use the Bonferroni's inequality and reach the lower bound of $P(A_{\epsilon,\delta})$. □

Subsequently, we have the following general upper bound on the regret $R_T$ of Algorithm 2 in the high probability sense, which holds on $A$ in any of the settings $s_1, s_2, s_3$ with sub-Gaussian rewards.

**Theorem 2.** *Let $f$ be a function specific to a setting and detailed later. For every $0 < \epsilon < 1$ and $0 < \delta < f(\epsilon, M, T)$, in setting $s_1$ with $c \geq \frac{1}{2} + \frac{1}{2}\sqrt{1 - (\frac{\epsilon}{MT})^{\frac{2}{M-1}}}$, $s_2$ and $s_3$, with the time horizon $T$ satisfying $T \geq L$, the regret of Algorithm 2 with $F(m, i, t) = \sqrt{\frac{C_1 \ln t}{n_{m,i}(t)}}$ satisfies that $E[R_T|A_{\epsilon,\delta}] \leq L + \sum_{i \neq i^*}(\max\{\lceil \frac{4C_1 \log T}{\Delta_i^2} \rceil, 2(K^2 + MK)\} + \frac{2\pi^2}{3P(A_{\epsilon,\delta})} + K^2 + (2M - 1)K) = O(\ln T)$ where the length of the burn-in period is explicitly $L =$*

$$\max\left\{\underbrace{\frac{\ln \frac{T}{2\epsilon}}{2\delta^2}, \frac{4K \log_2 T}{c_0}}_{L_{s_1}}, \underbrace{\frac{\ln \frac{\delta}{10}}{\ln p^*} + 25\frac{1+\lambda}{1-\lambda}\frac{\ln \frac{T}{2\epsilon}}{2\delta^2}, \frac{4K \log_2 T}{c_0}}_{L_{s_2}}, \underbrace{\frac{\ln \frac{\delta}{10}}{\ln p^*} + 25\frac{1+\lambda}{1-\lambda}\frac{\ln \frac{T}{2\epsilon}}{2\delta^2}, \frac{K \ln(\frac{MT}{\epsilon})}{\frac{\ln(\frac{1}{1 - \frac{2 \log M}{M-1}})}{c_0}}}_{L_{s_3}}\right\}$$

*with $\lambda$ being the spectral gap of the Markov chain in $s_2, s_3$ that satisfies $1 - \lambda \geq \frac{1}{2\frac{\ln 2}{\ln 2p^*}\ln 4 + 1}$, $p^* = p^*(M) < 1$ and $c_0 = c_0(K, \min_{i \neq i^*}\Delta_i, M, \epsilon, \delta)$, and the instance-dependent constant $C_1 = 8\sigma^2 \max\{12\frac{M(M+2)}{M^4}\}$.*

*Proof Sketch.* The proof is carried out in Appendix; here we describe the main ideas as follows. We note that the regret is proportional to the total number of pulling global sub-optimal arms by the end of round $T$. We fix client $m$ for illustration without loss of generality. We tackle all the possible cases when clients pull such a sub-optimal arm - (i) the condition $n_{m,i}(t) \leq N_{m,i}(t) - K$ is met, (ii) the upper confidence bounds of global sub-optimal arms deviate from the true means, (iii) the upper confidence bounds of global optimal arms deviate from the true means, and (iv) the mean values of global sub-optimal arms are greater than the mean values of global optimal arms. The technical novelty of our proof is in that 1) we deduce that the total number of events (ii) and (iii) occurring can

be bounded by some constants using newly derived conditional concentration inequalities that hold by our upper bounds on the conditional moment generating functions and by the unbiasedness of the network-wide estimators and 2) we control (i) by analyzing the scenarios where the criteria are met, which do not occur frequently. □

**Remark** (**Specification of the parameters**). *Note that the choice of $f$ depends on the problem settings. Specifically, in setting $s_1$, we set $f(\epsilon, M, T) = \frac{1}{2} + \frac{1}{4}\sqrt{1 - (\frac{\epsilon}{MT})^{\frac{2}{M-1}}}$. By the definition of $c$, we have $f(\epsilon, M, T) < c$. In setting $s_2$ with $M < 11$, we specify $f(\epsilon, M, T) = \frac{1}{2}$ which meets $f < c$ due to (1). Lastly, in setting $s_3$ with $M \geq 11$, we choose $f(\epsilon, M, T) = \frac{1}{2}\frac{2\log M}{M-1}$ and again we have $f < c$ due to (1). We observe that the regret bound is dependent on the transition kernel $\pi$ and the spectral gap $\lambda$ of the underlying Markov chain associated with $\pi$. This indicates the significance of graph complexities and distributions within the framework of the random graph model when deriving the regret bounds, in a similar manner as the role of graph conductance in the regret bounds established in [Sankararaman et al., 2019, Chawla et al., 2020] for time-invariant graphs.*

To proceed, we show a high probability upper bound on the regret $E[R_T | A_{\epsilon,\delta}]$ of Algorithm 2 for settings $S_1, S_2, S_3$ with sub-exponential rewards.

**Theorem 3.** *Let $f$ be a function specific to a setting and defined in the above remark. For every $0 < \epsilon < 1$ and $0 < \delta < f(\epsilon, M, T)$, in settings $S_1$ with $c \geq \frac{1}{2} + \frac{1}{2}\sqrt{1 - (\frac{\epsilon}{MT})^{\frac{2}{M-1}}}, S_2, S_3$ with the time horizon $T$ satisfying $T \geq L$, the regret of Algorithm 2 with $F(m, i, t) = \sqrt{\frac{C_1 \ln T}{n_{m,i}(t)}} + \frac{C_2 \ln T}{n_{m,i}(t)}$ satisfies $E[R_T | A_{\epsilon,\delta}] \leq L + \sum_{i \neq i^*} (\Delta_i + 1) \cdot (\max([\frac{16C_1 \log T}{\Delta_i^2}], [\frac{4C_2 \log T}{\Delta_i}], 2(K^2 + MK)) + \frac{4}{P(A_{\epsilon,\delta})T^3} + K^2 + (2M-1)K) = O(\ln T)$ where $L, C_1$ are specified as in Theorem 2 and $\frac{C_2}{C_1} \geq \frac{3}{2}$.*

*Proof Sketch.* The proof is detailed in Appendix. The proof logic is similar to that of Theorem 2. However, the main differences lie in the upper confidence bounds, which require proving new concentration inequalities and moment generating functions for the network-wide estimators. □

In addition to the instance-dependent regret bounds of order $O(\frac{\log T}{\Delta_i})$ that depend on the sub-optimality gap $\Delta_i$ which may be arbitrarily small and thereby leading to large regret, we also establish a universal, mean-gap independent regret bound that applies to settings with sub-exponential and sub-Gaussian rewards. A formal proof is deferred to Appendix.

**Theorem 4.** *Assume the same conditions as in Theorems 2 and 3. The regret of Algorithm 2 satisfies that $E[R_T | A_{\epsilon,\delta}] \leq L_1 + \frac{4}{P(A_{\epsilon,\delta})T^3} + (\sqrt{\max(C_1, C_2)\ln T} + 1)\frac{4M}{P(A_{\epsilon,\delta})T^3} + K(C_2(\ln T)^2 + C_2 \ln T + \sqrt{C_1 \ln T}\sqrt{T(\ln T + 1)}) = O(\sqrt{T}\ln T)$. where $L_1 = \max(L, K(2(K^2 + MK)))$, $L, C_1$ is specified as in Theorem 2, and $\frac{C_2}{C_1} \geq \frac{3}{2}$. The involved constants depend on $\sigma^2$ but not on $\Delta_i$.*

### 3.3 Other Performance Measures

**Communication cost** Assuming a constant cost of establishing a communication link, as defined in [Wang et al., 2020, Li and Song, 2022], denoted as $c_1$, the communication cost $C_T$ can be calculated as $C_T = c_1 \cdot \sum_{t=1}^{T} |E_t|$, which is proportional to $\sum_{t=1}^{T} |E_t|$. Alternatively, following the framework proposed in [Wang et al., 2022, Li and Wang, 2022, Sankararaman et al., 2019, Chawla et al., 2020], the communication cost can be defined as the total number of communications among clients, which can be represented as $C_T = \sum_{t=1}^{T} |E_t|$, similar to the previous definition. Regarding the quantity $C_T = \sum_{t=1}^{T} |E_t|$, the number of edges $E_t$ could be $O(M)$ for sparse graphs, and at most $O(M^2)$. In the random graph model, the expected number of edges is $\frac{M(M-1)}{2}c$, which implies $O(M^2)$ in the worst case scenario and the total communication cost, in a worst-case scenario, is of order $O(TM^2)$. This analysis holds also for the random connected graph case where $c$ represents the probability of having an edge. This cost aligns with the existing work of research on decentralized distributed multi-agent MAB problems without a focus on the communication cost, where edge-wise communication is a standard practice. Optimizing communication costs to achieve sublinearity, as discussed in [Sankararaman et al., 2019, Chawla et al., 2020], is a subject for future research.

**Complexity and privacy guarantee** At each time step, the time complexity of the graph generation algorithm is $O(M^2 + M + |E_t|) \leq O(M^2)$, where the first term accounts for edge selection, and the second and third terms are for graph connectivity verification using Algorithm 1 (benefitting from the use of Markov chains for graph generation). The main algorithm comprises various stages involving

multiple agents. The overall time complexity is calculated as $O(MK + M^2 + MK + M^2) = O(M^2 + MK)$, consistent with that of [Zhu et al., 2021b, Dubey et al., 2020]. It is noteworthy that most procedures can operate in parallel (synchronously), such as arm pulling, broadcasting, estimator updating, and E-R model generation, with the exception being random connected graph generation due to the Markovian property. Privacy guarantees are also important. Here, we note that clients only communicate aggregated values of raw rewards. Differential privacy is not the focus of this work but may be considered in future research.

## 4 Numerical Results

In this section, we present a numerical study of the proposed algorithm. Specifically, we first demonstrate the regret performance of Algorithms 2 and 3, in comparison with existing benchmark methods from the literature, in a setting with time-invariant graphs. Moreover, we conduct a numerical experiment with respect to time-varying graphs, comparing the proposed algorithm with the most recent work [Zhu and Liu, 2023]. Furthermore, the theoretical regret bounds of the proposed algorithm, as discussed in the previous section, exhibit different dependencies on the parameters that determine the underlying problem settings. Therefore, we examine these dependencies through simulations to gain insights into the exact regret incurred by the algorithm in practice. The benchmark algorithms include GoSInE [Chawla et al., 2020], Gossip_UCB [Zhu et al., 2021b], and Dist_UCB [Zhu and Liu, 2023]. Notably, GoSInE and Gossip_UCB are designed for time-invariant graphs, while Dist_UCB and the proposed algorithm (referred to as DrFed-UCB) are tailored for time-varying graphs. Details about numerical experiments are refered to Appendix.

**Benchmark comparison results** The visualizations for both time-invariant and time-varying graphs are in Appendix. We evaluated regret by averaging over 50 runs, along with the corresponding 95% confidence intervals. In time-invariant graphs, DrFed-UCB consistently demonstrates the lowest average regret, showcasing significant improvements. Dist_UCB and DrFed-UCB exhibit larger variances (Dist_UCB having the largest variances), which might have resulted from the communication mechanisms designed for time-varying graphs. In time-varying graphs, our regret is substantially lower compared to that of Dist_UCB. In terms of time complexity, DrFed-UCB and GoSInE are approximately six times faster than Dist_UCB.

**Regret dependency results** Additionally, we illustrate how the regret of DrFed-UCB depends on several factors, including the number of clients $M$, the number of arms $K$, the Bernoulli parameter $c$ for the E-R model, and heterogeneity measured by $\max_{i,j,k} |\mu_i^k - \mu_j^k|$. The visualizations are available in Appendix. We observe that regret monotonically increases with the level of heterogeneity and the number of arms, while it monotonically decreases with connectivity, which is equivalent to an increase in graph complexity. However, this monotonic dependency does not hold with respect to $M$ due to the accumulation of the information gain.

## 5 Conclusions

In this paper, we consider a decentralized multi-agent multi-armed bandit problem in a fully stochastic environment that generates time-varying random graphs and heterogeneous rewards following sub-gaussian and sub-exponential distributions, which has not yet been studied in existing works. To the best of our knowledge, this is the first theoretical work on random graphs including the E-R model and random connected graphs, and the first work on heterogeneous rewards with heavy tailed rewards. To tackle this problem, we develop a series of new algorithms, which first simulate graphs of interest, then run a warm-up phase to handle graph dynamics and initialization, and proceed to the learning period using a combination of upper confidence bounds (UCB) with a consensus mechanism that relies on newly proposed weight matrices and updates, and using a stopping time to handle randomly delayed feedback. Our technical novelties in the results and the analyses are as follows. We prove high probability instance-dependent regret bounds of the order of $\log T$ in both sub-gaussian and sub-exponential cases, consistent with the regret bound in the existing works that only consider the expected regret. Moreover, we establish a nearly tight instance-free regret bound of order $\sqrt{T} \log T$ for both sub-exponential and sub-gaussian distributions, up to a $\log T$ factor. We leverage probabilistic graphical methods on random graphs and draw on theories related to rapidly mixing Markov chains, which allows us to eliminate the doubly stochasticity assumption through new weight matrices and a stopping time. We construct new network-wide estimators and invent new concentration inequalities for them, and subsequently incorporate the seminal UCB algorithm into this distributed setting. A discussion on future work is refered to Appendix.

## Acknowledgement

We deeply appreciate the NeurIPS anonymous reviewers and the meta-reviewer for their valuable and insightful suggestions and discussions. This final version of the paper has been made possible through their great help. Additionally, we are much obliged to the authors of the papers [Chawla et al., 2020, Zhu et al., 2021b], especially Ronshee Chawla from [Chawla et al., 2020] and Zhaowei Zhu from [Zhu et al., 2021b], for promptly sharing the code of their algorithms, which has helped us to run the benchmarks presented in this work. Their gesture definitely improves the advancement of the field.

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

# A  Additional Background Work

Developing efficient algorithms for decentralized systems has been a popular research area in recent years. Among them, gossiping algorithms have been proven to be successful [Scaman et al., 2017, Duchi et al., 2011, Nedic and Ozdaglar, 2009]. In this approach, each client computes iteratively a weighted average of local estimators and network-wide estimators obtained from neighbors. The goal is to derive an estimator that converges to the average of the true values across the entire system. The weights are represented by a matrix that respects the graph structure under certain conditions. The gossiping-based averaging approach enables the incorporation of MAB methods in decentralized settings. In particular, motivated by the success of the UCB algorithm [Auer et al., 2002a] in stochastic MAB, [Landgren et al., 2016a,b, 2021, Zhu et al., 2020, 2021a,b, Martínez-Rubio et al., 2019, Chawla et al., 2020, Wang et al., 2021] import it to various decentralized settings under the assumption of sub-Gaussianity, including homogeneous or heterogeneous rewards, different graph assumptions, and various levels of global information. The regret bounds obtained are typically of order $\log T$. However, most existing works assume that the graph is time-invariant under further conditions, which is often not the case. For example, [Wang et al., 2021] provide a optimal regret guarantee for complete graphs which are essentially a centralized batched bandit problem [Perchet et al., 2016]. Connected graphs are also considered, but [Zhu et al., 2020] assume that the rewards are homogeneous and graphs are time-invariant related to doubly stochastic matrices. In addition, [Martínez-Rubio et al., 2019] propose the DDUCB algorithm for settings with time-invariant graphs and homogeneous rewards, dealing with deterministically delayed feedback and assuming knowing the number of vertices and the spectral gap of the given graph. Meanwhile, [Jiang and Cheng, 2023] propose an algorithm C-CBGE that is robust to Gaussian noises and deals with client-dependent MAB, but requires time-invariant regular graphs. [Zhu et al., 2021b] propose a gossiping-based UCB-variant algorithm for time-invariant graphs. In this approach, each client maintains a weighted averaged estimator by gossiping, uses doubly stochastic weight matrices depending on global information of the graph, and adopts a UCB-based decision rule by constructing upper confidence bounds. Recently, [Zhu and Liu, 2023], revisit the algorithm and add an additional term to the UCB rule for time-varying repeatedly strongly connected graphs, assuming no global information. However, the doubly stochasticity assumption excludes many graphs from consideration. Our algorithm builds on the approach proposed by [Zhu et al., 2021b] with new weight matrices that do not require the doubly stochasticity assumption. Our weight matrices leverage more local graph information, rather than just the size of the vertex set as in [Zhu and Liu, 2023, Zhu et al., 2021b]. We introduce the terminology of the stopping time for randomly delayed feedback, along with new upper confidence bounds that consider random graphs and sub-exponentiality. This leads to smaller high probability regret bounds, and the algorithm only requires knowledge of the number of vertices that can be obtained at initialization or estimated as in [Martínez-Rubio et al., 2019].

In the context of bandits with heavy-tailed distributed rewards, the UCB algorithm continues to play a significant role. [Dubey et al., 2020] are the first to consider the multi-agent MAB setting with homogeneous heavy-tailed rewards. They develop a UCB-based algorithm with an instance-dependent regret bound of order $\log T$. They achieve this by adopting larger upper confidence bounds and finding cliques of vertices, even though the graphs are time-invariant and known to clients. In a separate line of work, [Jia et al., 2021] consider the single-agent MAB setting with sub-exponential rewards, and propose a UCB-based algorithm that enlarges or pretrains the upper confidence bounds, achieving a mean-gap independent regret bound of order $\sqrt{T \log T}$. We extend this technique to the decentralized multi-agent MAB setting with heterogeneous sub-exponential rewards, using a gossiping approach, and establish both an optimal instance-dependent regret bound of $O(\log T)$ and a nearly optimal mean-gap independent regret bound of $O(\sqrt{T} \log T)$, up to a $\log T$ factor.

Our work draws on the classical literature on random graphs. From the perspective of generating random connected graphs, we build upon a numerically efficient algorithm introduced in [Gray et al., 2019], which is based on the Metropolis-Hastings algorithm [Chib and Greenberg, 1995], despite its lack of finite convergence rate for non-sparse graphs. We follow their algorithm and, in addition, provide a new analysis on the convergence rate and mixing time of the underlying Markov chain. In terms of the E-R model, it has been thoroughly examined in various areas, such as mean-field games [Delarue, 2017] and majority vote settings [Lima et al., 2008]. However, these random graphs have not yet been applied to the decentralized multi-agent MAB setting that is partially determined by the underlying graphs. Our formulation and analyses bridge this gap, providing insights into the dynamics of decentralized multi-agent MAB in the context of random graphs.

# B Future work

Recent advancements have been made in reducing communication costs with respect to the dependency in multi-agent MAB with homogeneous rewards (in the generalized linear bandit setting [Li and Wang, 2022], the ground truth of the unknown parameter is the same for all clients), such as achieving $O(\sqrt{T}M^2)$ in [Li and Wang, 2022] for centralized settings or $O(M^3 \log T)$ through Global Information Synchronization (GIS) communication protocols assuming time-invariant graphs in [Li and Song, 2022]. Likewise, [Sankararaman et al., 2019, Chawla et al., 2020] improve the communication cost of order $\log T$ or $o(T)$ through asynchronous communication protocols and balancing the trade-off between regret and communication cost. More recently, [Wang et al., 2022] establish a novel communication protocol, TCOM, which is of order $\log \log T$ by means of concentrating communication around sub-optimal arms and performing aggregation of estimators across time steps. Furthermore, [Wang et al., 2020] develops a new leader-follower communication protocol, which selects a leader that communicates to the followers. Here the communication cost is independent of $T$ which is much smaller. The incorporation of random graph structures and heterogeneous rewards introduces its own complexities, which poses challenges to reductions in communication costs. These great advancements introduce a promising direction for communication efficiency as a next step within the context herein.

# C Details on numerical experiments in Section 4

We report the experimental details in Section 4, including both benchmarking and regret properties of the algorithms. The implementation details of the experiments are as follows, including the data generation, benchmarks, and the evaluation metrics.

The process of data generation involves both reward generation and graph generation. First we generate different numbers of arms and clients, denoted as $K$ and $M$, respectively. Specifically, we generate rewards using the Bernoulli distribution in the sub-Gaussian distribution family, varying the mean values $\mu_i^m$ by introducing multiple levels of heterogeneity denoted as $h = \max_{i,j,m} |\mu_i^m - \mu_j|$ and then for each arm $k$, partitioning the range $[0.1, 0.1 + (k + 1) \cdot h/K]$ into $M$ intervals. In terms of graph generation, we generate E-R models with varying values of $c$, to capture graph complexity. Specifically, for the benchmarking experiment with time-invariant graphs, we set $K = 2, M = 5, h = 0.1, c = 1$, i.e. complete graphs. For the benchmarking experiment with time-varying graphs, we set $K = 2, M = 5, h = 0.1, c = 0.9$. For the regret experiments, the parameters are $h \in \{0.1, 0.2, 0.3\}$, $M \in \{5, 8, 12\}$, $c \in \{0.2, 0.5, 0.9, 1\}$, and $K \in \{2, 3, 4\}$. We selected the least positive number of arms $K = 2$ to keep computational times low and $M = 5$ to have small graphs but still a variety of them.

We compare the new method DrFed-UCB with the classical methods, such as the Gossiping Insert-Eliminate algorithm (GoSInE) in [Chawla et al., 2020] which focuses on deterministic graphs and sub-Gaussian rewards and motivated our work. We also include the Gossip UCB algorithm (Gossip_UCB) [Zhu et al., 2021b] as a benchmark. Meanwhile, in terms of time-varying graphs, we implement the algorithm, Distributed UCB (Dist_UCB) in [Zhu and Liu, 2023] that has been developed for time-varying graphs, and compare our algorithm to this benchmark.

**Evaluation** The evaluation metric is the regret measure as defined in Section 4. More specifically, for the experiments, we use the average regret over 50 runs for each benchmark and also report the 95% confidence intervals across the 50 runs. With respect to the communication cost as another performance measure, it is computed explicitly. Additionally, the runtime can provide insights into the time complexity of the models.

**Benchmark comparison results** The results for time-invariant and time-varying graphs are presented in Figure 1 (a) and Figure 1 (b), respectively. The x-axis represents time steps, while the y-axis shows the average regret up to that time step. Figure 1 (a) demonstrates that DrFed-UCB exhibits the smallest average regret among all methods in time-invariant graphs, with significant improvements. More precisely, with respect to the Area Under the Curve (AUC) of the regret curve, the improvements of DrFed-UCB over GoSInE, Gossip_UCB, and Dist_UCB are 132%, 158%, and 128%, respectively, showcasing the regret improvement of the newly proposed algorithm compared to the benchmarks. Notably, both Dist_UCB and DrFed-UCB result in larger variances, primarily observed in Dist_UCB. This phenomenon may be attributed to the communication mechanisms designed for time-varying graphs. In Figure 1 (b), we observe that our regret is notably smaller compared to Dist_UCB in settings with time-varying graphs. Specifically, the AUC of Dist_UCB is

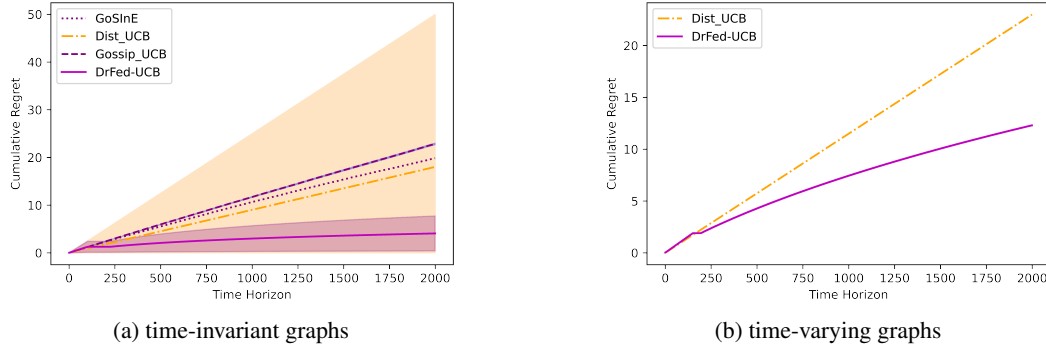

(a) time-invariant graphs

(b) time-varying graphs

Figure 1: The regret of different methods in settings with both time-invariant and time-varying graphs

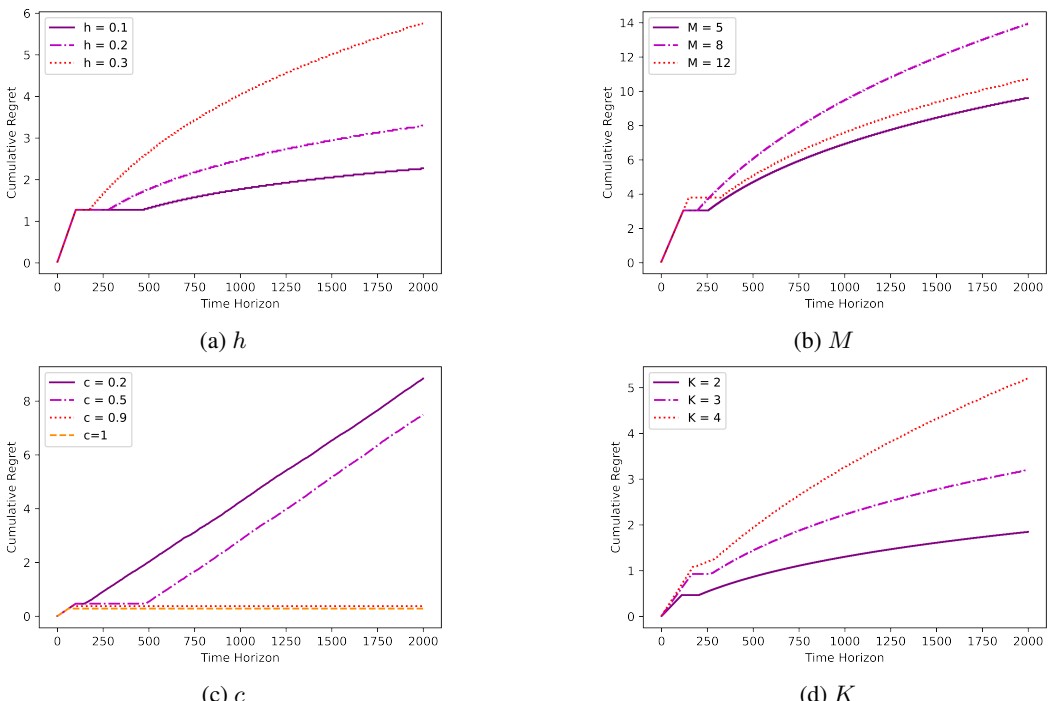

(a) $h$

(b) $M$

(c) $c$

(d) $K$

Figure 2: The regret of the proposed algorithm in problem settings with different parameters

96.6% larger than that of our regret curve, which implies the significant improvement in this setting with time-varying graphs. Furthermore, we perform a time complexity comparison, revealing that DrFed-UCB and GoSInE are approximately six times faster than Dist_UCB. Lastly, communication cost is directly computed by the total number of communication rounds and follows an explicit formula. Specifically, the communication costs of DrFed-UCB, Gossip_UCB, and Dist_UCB are of order $T$, whereas GoSInE exhibits only $o(T)$, suggesting a potential direction for optimizing communication costs.

**Regret dependency results** Meanwhile, we illustrate how DrFed-UCB's regret depends on several factors: the number of clients ($M$), the number of arms ($K$), the Bernoulli parameter ($c$) for the E-R model, and heterogeneity measured by $h$. The regret metrics are presented as (a), (b), (c), and (d) in Figure 2, respectively. We observe that regret monotonically increases with the level of heterogeneity and the number of arms, while decreasing with connectivity, which is equivalent to an increase in graph complexity. However, this monotonic trend does not apply to the number of clients. This is due to the following considerations. On one hand, a large $M$ implies a greater number of incident edges

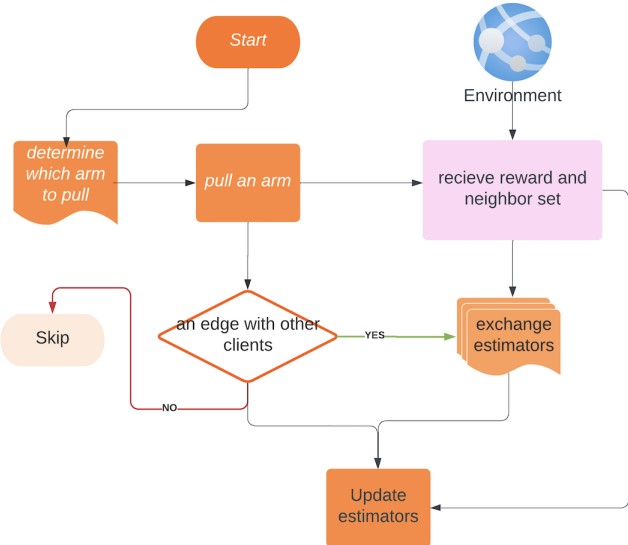

Figure 3: Flowchart of Algorithm 2

of each client, providing more global information access and potentially leading to smaller regret. On the other hand, a large $M$ also weakens the Chernoff-Hoeffding inequality for clients transmitting information, which might result in larger regret.

## D    Algorithms and Tables in Section 3

The algorithm for generating random connected graphs is presented in Algorithm 3 as follows.

---

**Algorithm 3:** Generate a uniformly distributed connected graph

---

Initialization: Let $\tau_1$ be given; Generate a random graph $G^{init}$ by selecting each edge with probability $\frac{1}{2}$;

Connectivity: make $G^{init}$ connected by adding the least many edges to get $G_0$ ;

**for** $t = 0, 1, 2, \dots, \tau_1$ **do**

    Randomly sample an edge pair $e = (i, j)$;

    Denote the edge set of $G_s$ as $E_s$;

    **if** $e \in E_s$ **then**

        Remove $e$ from $E_s$ to get $G'_s = (V, E_s \backslash \{e\})$;

        **if** $G'_s$ *is connected* **then**

            $G_{s+1} = G'_s$;

        **else**

            reject $G'_s$ and set $G_{s+1} = G_s$;

        **end**

    **else**

        $G_{s+1} = (V, E_s \cup \{e\})$;

    **end**

**end**

---

The flowchart of Algorithm 2 is presented in Figure 3 to illustrate the information flow in the algorithm. The table below displays the various settings we consider for the regret analysis.

Table 1: Settings

| | E-R | uniform | M | reward |
|---|---|---|---|---|
| $s_1$ | ✓ | | any | sub-g |
| $s_2$ | | ✓ | $[1, 10]$ | sub-g |
| $s_3$ | | ✓ | $[11, \infty)$ | sub-g |
| $S_1$ | ✓ | | any | sub-e |
| $S_2$ | | ✓ | $[1, 10]$ | sub-e |
| $S_3$ | | ✓ | $[11, \infty)$ | sub-e |

# E    Remarks on the theoretical results in Section 3.2

## E.1    Remarks on Theorem 2

**Remark** (**The condition on the time horizon**). *Although the above regret bound holds for any $T > L$, the same bound applies to $T \leq L$ as follows. Assuming $T \leq L$, we obtain $E[R_T|A_{\epsilon,\delta}] \leq T \leq L$ where the first inequality is by noting that the rewards are within the range of $[0, 1]$.*

**Remark** (**The upper bound on the expected regret**). *Theorem 2 states a high probability regret bound, while the expected regret is often considered in the existing literature. As a corollary of Theorem 2, we establish the upper bound on $E[R_T]$ if $\epsilon = \frac{\log T}{MT}$ as follows. Note that*

$$E[R_T] = E[R_T|A_{\epsilon,\delta}]P(A_{\epsilon,\delta}) + E[R_T|A_{\epsilon,\delta}^c]P(A_{\epsilon,\delta}^c) \leq P(A_{\epsilon,\delta}) \cdot E[R_T|A_{\epsilon,\delta}] + T \cdot (1 - P(A_{\epsilon,\delta}))$$

$$\leq (1 - 7\epsilon)(L + \sum_{i \neq i^*}(\max\{[\frac{4C_1 \log T}{\Delta_i^2}], 2(K^2 + MK)\} + \frac{2\pi^2}{3P(A_{\epsilon,\delta})} + K^2 + (2M-1)K)) + 7\epsilon T$$

$$\leq l_1 + l_2 \log T + \sum_{i \neq i^*}(\max\{[\frac{4C_1 \log T}{\Delta_i^2}], 2(K^2 + MK)\} + \frac{2\pi^2}{3(1 - 7\epsilon)} + K^2 + (2M-1)K) + 7\frac{\log T}{M}$$

*where the first inequality uses $E[R_T|A_{\epsilon,\delta}] \leq T$ and the second inequality follows by Theorem 1. Here $l_1$ and $l_2$ are constants depending on $K, M, \delta, \min_{i \neq i^*} \Delta_i$, and $\lambda$.*

**Remark** (**Comparison with previous work**). *A comparison to the regret bounds in the existing literature considering sub-Gaussian rewards is as follows. Our regret bounds are consistent with the prior works where the expected regret bounds are of order $\log T$. Note that the regret bounds in [Zhu and Liu, 2023] cannot be used here since the update rule and the settings are different. Their update rule and analyses cannot carry over to our settings, which explains why we invent these modifications and proofs. On the one hand, the time-varying graphs they consider do not include the E-R model, and we can find counter-examples where their doubly stochastic weight matrices $W_t$ result in the divergence of $W_1 \cdot W_2 \ldots W_T$. This makes the key proof step invalid in our framework. On the other hand, their time-varying graphs include the connected graphs when $l = 1$, but they also make an additional assumption of doubly stochastic weight matrices, which is not applicable to regular graphs. Furthermore, they study an expected regret upper bound, while we prove a high probability regret bound that captures the dynamics in the random graphs. The graph assumptions in other works, however, are stronger, such as [Zhu et al., 2021b] consider time-invariant graphs and [Wang et al., 2021] assume graphs are complete [Perchet et al., 2016]. In contrast to some work that focuses on homogeneous rewards in decentralized multi-agent MAB, we derive regret bounds of the same order $\log T$ in a heterogeneous setting. If we take a closer look at the coefficients in terms of $K, M, \lambda, \Delta_i$, our regret bound is determined by $O(\max(K, \frac{1+\lambda}{1-\lambda}, \frac{1}{M^2 \Delta_i}) \log T)$. The work of [Zhu and Liu, 2023] arrives at $O(\max\{\frac{\log T}{\Delta_i}, K_1, K_2\})$ where $K_1, K_2$ are related to $T$ without explicit formulas. Our regret is smaller when $K\Delta_i \leq 1$ and $\frac{1+\lambda}{1-\lambda}\Delta_i \leq 1$, which can always hold by rescaling $\Delta_i$, i.e. for many cases we get substantial improvement.*

### E.2 Remarks on Theorem 4

**Remark.** *Based on the expression of $L_1$, we obtain that $L_1$ is independent of the sub-optimality gap $\Delta_i$. Meanwhile, we have $C_1 = 8\sigma^2 \cdot 12\frac{M(M+2)}{M^4}$ and $C_2 = \frac{3}{2}C_1 = 12\sigma^2 \cdot 12\frac{M(M+2)}{M^4}$. This implies that the established regret bound in Theorem 4 does not rely on $\Delta_i$ but does depend on $\sigma^2$. To this end, we use the terminology, mean-gap independent bounds, to only represent bounds having no dependency on $\Delta_i$, rather than instance independent that seems to be an overclaim in this case.*

**Remark** (**Comparison with previous work**). *For decentralized multi-agent MAB with homogeneous heavy-tailed rewards and time-invariant graphs, [Dubey et al., 2020] provide an instance-dependent regret bound of order $\log T$. In contrast, our regret bound has the same order for heterogeneous settings with random graphs, as shown in Theorem 3. Additionally, we provide a mean-gap independent regret bound as in Theorem 4. In the single-agent MAB setting, [Jia et al., 2021] consider sub-exponential rewards and derive a mean-gap independent regret upper bound of order $\sqrt{T \log T}$. Our regret bound of $\sqrt{T} \log T$ is consistent with theirs, up to a logarithmic factor. Furthermore, our result is consistent with the regret lower bound as proposed in [Slivkins et al., 2019], up to a $\log T$ factor, indicating the tightness of our regret bound.*

**Remark.** *The discussion regarding the conditions on $T$, the expected regret $E[R_T]$, and the parameter specifications follow the same logic as those in Theorem 2. We omit the details here.*

## F Proof of results in Section 3.2

### F.1 Lemmas and Propositions

**Lemma 1.** *For any $m, i, t > L$, we have*

$$n_{m,i}(t) \geq N_{m,i}(t) - K(K + 2M)$$

*Proof of Lemma 1.* The proof is referred to [Zhu and Liu, 2023]. □

**Lemma 2.** *For any $m, i, t > L$, if $n_{m,i}(t) \geq 2(K^2 + KM + M)$ and graph $G_t$ is connected, then we have*

$$n_{m,i}(t) \leq 2\min_j n_{j,i}(t).$$

*where the min is taken over all clients, not just the neighbors.*

*Proof of Lemma 2.* The proof is referred to [Zhu and Liu, 2023]. □

**Lemma 3** (Generalized Holder's inequality). *For any $r > 0$ and measurable functions $h_i$ for $i = 1, \ldots, n$, if $\sum_{i=1}^{n} \frac{1}{p_i} = \frac{1}{r}$, then*

$$||\Pi_{i=1}^{n} h_i||_r \leq \Pi_{i=1}^{n} ||h_i||_{p_k}.$$

The proof follows from the Young's inequality for products.

**Lemma 4.** *Suppose that random variables $X_1, X_2, \ldots, X_n$ are such that $Y_i = E[(X_1, \ldots, X_n)|(X_1, \ldots, X_{i-1}, X_{i+1}, \ldots, X_n)]$ are sub-Gaussian distributed with variance proxy $\sigma_1, \sigma_2, \ldots, \sigma_n$, respectively. Then the sum of these sub-Gaussian random variables, $\sum_{i=1}^{n} X_i$, is again sub-Gaussian with variance proxy $\left(\sum_{i=1}^{n} \sigma_i\right)^2$.*

*Proof.* First, without loss of generality, let us assume $E[X_i] = 0$. Otherwise, we can always construct a random variable $X_i - E[X_i]$ which has the same variance proxy with a difference up to a constant.

Defining $p_i = \frac{\sum_{k=1}^n \sigma_k}{\sigma_i}$ gives $\sum_{i=1}^n \frac{1}{p_i} = 1$. Let $\mu$ be the distribution function of random vector $(X_1, \ldots, X_n)$. By specifying $h_i(x) = \exp(\lambda x)$ and $r = 1$, we obtain that for any $\lambda > 0$ we have

$$E[\exp\{\lambda(\sum_{i=1}^n X_i)\}]$$

$$= E[\Pi_{i=1}^n \exp\{\lambda X_i\}]$$

$$= \int_0^\infty \Pi_{i=1}^n \exp\{\lambda X_i\} d\mu$$

$$\leq \Pi_{i=1}^n \|\exp\{\lambda X_i\}\|_{\frac{\sum_{k=1}^n \sigma_k}{\sigma_i}}$$

$$= \Pi_{i=1}^n (\int_0^\infty \exp\{\lambda X_i\}^{\frac{\sum_{k=1}^n \sigma_k}{\sigma_i}} d\mu)^{\frac{\sigma_i}{\sum_{k=1}^n \sigma_k}}$$

$$= \Pi_{i=1}^n (E_{Y_i}[\exp\{\lambda X_i \frac{\sum_{k=1}^n \sigma_k}{\sigma_i}\}]^{\frac{\sigma_i}{\sum_{k=1}^n \sigma_k}}$$

$$\leq \Pi_{i=1}^n [\exp\{\frac{1}{2}\sigma_i^2 \lambda^2 \frac{(\sum_{k=1}^n \sigma_k)^2}{\sigma_i^2}\}]^{\frac{\sigma_i}{\sum_{k=1}^n \sigma_k}}$$

$$= [\exp\{\frac{1}{2}\lambda^2(\sum_{k=1}^n \sigma_k)^2\}]^{\sum_{i=1}^n \frac{\sigma_i}{\sum_{k=1}^n \sigma_k}}$$

$$= \exp\{\frac{1}{2}\lambda^2(\sum_{k=1}^n \sigma_k)^2\}$$

where the first inequality is by Lemma 3 and the second inequality follows the definition of sub-Gaussian random variables.

$\square$

**Lemma 5.** *Suppose that random variables $X_1, X_2, \ldots, X_n$ are independent sub-Gaussian distributed with variance proxy $\sigma_1, \sigma_2, \ldots, \sigma_n$, respectively. Then we have that the sum of these sub-Gaussian random variables, $\sum_{i=1}^n X_i$, is again sub-Gaussian with variance proxy $\sum_{i=1}^n \sigma_i^2$.*

*Proof.* For any $\lambda > 0$ note that

$$E[\exp\{\lambda(\sum_{i=1}^n X_i)\}]$$

$$= E[\Pi_{i=1}^n \exp\{\lambda X_i\}].$$

Since $X_1, X_2, \ldots, X_n$ are independent random variables, we further have

$$E[\exp\{\lambda(\sum_{i=1}^n X_i)\}]$$

$$= \Pi_{i=1}^n E[\exp \lambda X_i]$$

$$\leq \Pi_{i=1}^n \exp\{\frac{1}{2}\lambda^2 \sigma_i^2\}$$

$$= \exp\{\frac{1}{2}\lambda^2 \sum_{i=1}^n \sigma_i^2\}$$

where the inequality is by the definition of sub-Gaussian random variables.

This concludes the proof.

$\square$

**Proposition 1.** *Under E-R, for any $1 > \delta, \epsilon > 0$, and any fixed $t$, $t \geq L_{s1}$, the maintained matrix $P_t$ satisfies*

$$\|P_t - cE\|_\infty \leq \delta$$

*with probability $1 - \frac{\epsilon}{T}$. This implies that with probability at least $1 - \epsilon$ for any $t \geq L_{s_1}$, we have*

$$||P_t - cE||_\infty \leq \delta.$$

*Proof.* We start with the convergence rate of matrix $P_t$ for fixed $t$.

We recall that in E-R, the indicator function $X_{i,j}^s$ for edge $(i,j)$ at time step $s$ follows a Bernoulli distribution with mean value $c$. This implies that $\{X_{i,j}^s\}_s$ are i.i.d. random variables which allows us to use the Chernoff-Hoeffding inequality

$$P(|\frac{\sum_{s=1}^t X_{i,j}^s}{t} - c| > \delta) \leq 2\exp\{-2t\delta^2\}.$$

For the probability term, we note that for any $t \geq L_{s_1}$,

$$2\exp\{-2t\delta^2\} \leq \frac{\epsilon}{T}$$

since $t \geq L_{s_1} \geq \frac{\ln\frac{T}{2\epsilon}}{2\delta^2}$ by the choice $L_{s_1}$ of the burn-in period in setting 1.

As a result, the maintained matrix $P_t$ satisfies with probability at least $1 - \frac{\epsilon}{T}$ that

$$||P_t - cE||_\infty$$
$$= \max_{i,j}|\frac{\sum_{s=1}^t X_{i,j}^s}{t} - c|$$
$$\leq \delta$$

which concludes the first part of the statement.

Subsequently, consider the probability $P(||P_t - cE||_\infty < \delta, \forall t > L_{s_1})$. We obtain

$$P(||P_t - cE||_\infty < \delta, \forall t > L_{s_1})$$
$$= 1 - P(\cup_{t \geq L_{s_1}}||P_t - cE||_\infty < \delta)$$
$$\geq 1 - \sum_{t \geq L_{s_1}} P(||P_t - cE||_\infty < \delta)$$
$$\geq 1 - (T - L_{s_1})\frac{\epsilon}{T} \geq 1 - \epsilon$$

where the first inequality uses the Bonferroni's inequality.

This completes the second part of the statement.

$\square$

We next pin down the Markov chain governing Algorithm 1. Its states compound to all connected graphs if $G$ and $G'$ are connected, then the transition probability is defined by

$$\pi(G'|G) = \begin{cases} 0 \text{ if } |E(G')\Delta E(G)| > 1 \\ \frac{2}{M(M-1)} \text{ if } |E(G')\Delta E(G)| = 1 \\ 1 - \frac{2\alpha(G)}{M(M-1)} \text{ if } G' = G. \end{cases}$$

Here $\Delta$ denotes the symmetric difference and $\alpha(G)$ is the number of all connected graph that differ with $G$ by at most one edge. Algorithm 1 is a random walk in the Markov chain denoted as $CG - MC$. The intriguing question is if the stationary distribution corresponds to the uniform distribution on all connected graphs on $M$ nodes and if it is rapidly mixing. The next paragraph gives affirmative answers.

**Proposition 2.** *In $CG - MC$, for any time step $n \geq 1$ and initial connected graph $G^{init}$, we have*

$$||\pi^n(\cdot|G^{init}) - \pi^*(\cdot)||_{TV} \leq 2(p^*)^n$$

*where $p^* = p^*(M) < 1$ and $\pi^*$ is the uniform distribution on all connected graphs.*

*Proof.* Based on the definition of $\pi^*$, we have

$$\pi^* = \frac{1}{\#\{connected \quad graphs\}}.$$

Therefore, there exists a constant $0 < C_f < 1$ such that for any two connected graphs $G$, $G'$ with $|E(G)\Delta E(G')| = 1$ we have

$$\pi(G|G') \geq C_f \pi^*.$$

In essence $C_f = \frac{1}{\pi^*} \min_{G,G'} \pi(G|G') < 1$.

If $G = G'$, then there are two possible cases. First, if $\alpha(G) < \frac{(M(M-1)}{2}$, then $\pi(G|G) > \frac{2}{M(M-1)} > \pi^*$ and $\pi(G|G) > 0$. Otherwise, we have $\pi(G|G) = 0$. In other words, the set $G \notin \{G' : \pi(G'|G) \leq \pi^*(G'), \pi(G'|G) > 0\}$.

This implies that for $G' \in \{G' : \pi(G'|G) \leq \pi^*(G'), \pi(G'|G) > 0\}$, we have $|E(G)\Delta E(G')| = 1$ and subsequently $\pi(G|G') \geq C_f \pi^*$.

We start with the one-step transition and obtain

$$||\pi(\cdot|G) - \pi^*(\cdot)||_{TV}$$

$$= 2 \sup_A |\int_A (\pi(G'|G) - \pi^*(G'))dG'|$$

$$\leq 2 \int_{\{G':\pi(G'|G)-\pi^*(G')\leq 0\}} (-\pi(G'|G) + \pi^*(G'))dG'$$

$$\leq 2 \int_{\{G':\pi(G'|G)=0\}} (-\pi(G'|G) + \pi^*(G'))dG' +$$

$$2 \int_{\{G':\pi(G'|G)>0,\pi(G'|G)-\pi^*(G')\leq 0\}} (-\pi(G'|G) + \pi^*(G'))dG'$$

$$= 2 \int_{\{G':\pi(G'|G)=0\}} (\pi^*(G'))dG' + 2(1-C_f) \int_{\{G':\pi(G'|G)>0,\pi(G'|G)-\pi^*(G')\leq 0\}} (\pi^*(G'))dG'$$

$$\leq 2P(\{G' : \pi(G'|G) = 0\}) + 2(1-C_f)(1 - P(\{G' : \pi(G'|G) = 0\}))$$

$$\leq 2(1 - \frac{1}{\#\{connected \quad graphs\}}) + 2(1-C_f)(1 - (1 - \frac{1}{\#\{connected \quad graphs\}}))$$

$$\doteq 2p' + 2(1-C_f)(1-p') = 2(p' + (1-C_f)(1-p'))$$

$$\doteq 2p^*$$

where we denote the term $1 - \frac{1}{\#\{connected \quad graphs\}}$ and the term $p' + (1-C_f)(1-p')$ by $p'$ and $p^*$, respectively. It is worth noting that $p^* = p^*(M)$ and $p^* < 1$ since $p', C_f < 1$. Here the third inequality uses the above argument on the graphs in the set $\{G' : \pi(G'|G) \leq \pi^*(G'), \pi(G'|G) > 0\}$, and the last inequality uses the following result. By definition,

$$P(\{G' : \pi(G'|G) = 0\})$$

$$= 1 - P(\{G' : \pi(G'|G) > 0\})$$

$$\leq 1 - P(\{G' : |E(G)\Delta E(G')| = 1\})$$

$$= 1 - \frac{\alpha(G)}{\#\{connected \quad graphs\}} \leq 1 - \frac{1}{\#\{connected \quad graphs\}}$$

where the last inequality uses $\alpha(G) \geq 1$ by the definition of $\alpha(G)$.

Suppose at time step $n$, the result holds, i.e. for any $G$

$$||\pi^n(\cdot|G) - \pi^*(\cdot)||_{TV} \leq 2(p^*)^n.$$

Then we consider the transition kernel at the $n + 1$ step. Note that

$$||\pi^{n+1}(\cdot|G) - \pi^*(\cdot)||_{TV}$$

$$= 2 \sup_A |\int_A (\pi^{n+1}(G'|G) - \pi^*(G'))d_{G'}|$$

$$\leq 2 \sup_A |\int_S \int_A (\pi^n(G'|S) - \pi^*(G'))(\pi(S|G) - \pi^*(S))d_{G'}d_S|$$

$$= 2 \sup_A |\int_S (\pi(S|G) - \pi^*(S))(\int_A (\pi^n(G'|S) - \pi^*(G'))d_{G'})d_S|$$

$$\leq 2 \cdot \frac{1}{2}||\pi^n(\cdot|S) - \pi^*(\cdot)||_{TV} \cdot \frac{1}{2}||\pi(\cdot|G) - \pi^*(\cdot)||_{TV}$$

$$= \frac{1}{2}||\pi^n(\cdot|S) - \pi^*(\cdot)||_{TV}||\pi(\cdot|G) - \pi^*(\cdot)||_{TV}$$

$$\leq \frac{1}{2} \cdot 2(p^*)^n \cdot 2p^* = 2(p^*)^{n+1}$$

where the second inequality is by using the definition of $|| \cdot ||_{TV}$ and the last inequality holds by the results in the basis step and the induction step, respectively. The first inequality requires more arguments as follows. Consider the integral

$$\int_S \int_A (\pi^n(G'|S) - \pi^*(G'))(\pi(S|G) - \pi^*(S))d_{G'}d_S$$

$$= \int_S \int_A (\pi^n(G'|S) - \pi^*(G'))\pi(S|G)d_{G'}d_S - (\pi^n(G'|S) - \pi^*(G'))\pi^*(S)d_{G'}d_S$$

$$= \int_S \int_A \pi^n(G'|S)\pi(S|G)d_{G'}d_S - \int_S \int_A \pi^*(G')\pi^n(S|G)d_{G'}d_S -$$

$$\int_S \int_A \pi^n(G'|S)\pi^*(S)d_{G'}d_S + \int_S \int_A \pi^*(G')\pi^*(S)d_{G'}d_S$$

$$\geq \int_A \pi^{n+1}(G'|G)d_{G'} - \int_A \pi^*(G')d_{G'} -$$

$$\int_S \pi^*(S)d_S + \int_S \pi^*(G')d_{G'}$$

$$= \int_A \pi^{n+1}(G'|G)d_{G'} - \int_A \pi^*(S)d_S = \int_A (\pi^{n+1}(G'|G) - \pi^*(G'))d_{G'}$$

where the results hold by exchanging the orders of the integrals as a result of Funibi's Theorem and the inequality uses the fact that $\int_A \pi^n(G'|S)d_{G'} \leq 1$.

This completes the proof by concluding the mathematical induction.

$\square$

**Proposition 3.** *For any $1 > \delta, \epsilon > 0$, we obtain that for setting 2, for any fixed $t \geq L_{s_2}$, the maintained matrix $P_t$ satisfies with probability $1 - 2\frac{\epsilon}{T}$*

$$||P_t - cE||_\infty \leq \delta.$$

*Meanwhile, the graph generated by Algorithm 3 converges to the stationary distribution with*

$$||\pi^t(\cdot|G) - \pi^*(\cdot)||_{TV} \leq \frac{1\delta}{5},$$

*where $\pi^*$ is the uniform distribution on all connected graphs.*

*Proof.* Suppose we run the rapidly mixing markov chain for a time period of length $\tau_1 = \frac{\ln \frac{\zeta}{2}}{\ln p^*}$ where $\zeta < \frac{\delta}{5}$. By applying Proposition 2, we obtain that for any time step $t > \tau_1$,

$$||\pi^t(\cdot|G) - f(\cdot)||_{TV} \leq 2(p^*)^{\tau_1}.$$

For the second phase, we reset the counter of time step and denote the starting point $t = \tau_1 + 1$ as $t = 1$. Based on the definition of $P_t$, we have $P_t = (\frac{\sum_{s=1}^{t} X_{i,j}^s}{t})_{1 \leq i \neq j \leq M}$ where $X_{i,j}^s$ is the indicator function for edge $(i,j)$ on Graph $G_s$ that follows the distribution $\pi^{\tau_1+s}(G, \cdot)$. Let us denote $Y_{i,j}^s$ the indicator function for edge $(i,j)$ on Graph $G_s^{obj}$ following the distribution $\pi^*(\cdot)$ and $P_t^{obj} = (\frac{\sum_{s=1}^{t} Y_{i,j}^s}{t})_{1 \leq i \neq j \leq M}$.

By the Chernoff-Hoeffding inequality and specifying $\zeta = 2(p^*)^{\tau_1}$, we derive

$$P(|E[Y_{i,j}^1] - \frac{\sum_{s=1}^{t} Y_{i,j}^s}{t}| \geq \zeta) \leq 2 \exp\{-2t\zeta^2\}, \tag{3}$$

i.e.

$$||P_t^{obj} - cE||_\infty \leq \zeta$$

holds with probability $1 - 2\exp\{-2t\zeta^2\}$.

Consider the difference between $P_t$ and $P_t^{obj}$ and we obtain that

$$P_t - P_t^{obj}$$
$$= (\frac{\sum_{s=1}^{t} X_{i,j}^s - \sum_{s=1}^{t} Y_{i,j}^s}{t})_{1 \leq i \neq j \leq M}$$
$$= (\frac{\sum_{s=1}^{t} X_{i,j}^s}{t} - E[X_{i,j}^1] + E[X_{i,j}^1 - Y_{i,j}^1] + E[Y_{i,j}^1] - \frac{\sum_{s=1}^{t} Y_{i,j}^s}{t})_{1 \leq i \neq j \leq M}$$

where the last term is bounded by (3).

For the second quantity $E[(X_{i,j}^s - Y_{i,j}^s)]$, we have that for any $s, i, j$

$$E[(X_{i,j}^s - Y_{i,j}^s)|G_s, G_s^{obj}]$$
$$= 1_{G_s \text{ contains edge } (i,j) \& G_s^{obj} \text{ does not contain edge } (i,j)} - 1_{(G_s \text{ does not contain edge } (i,j) \& G_s^{obj} \text{ contains edge } (i,j))}$$

and subsequently by the law of total expectation, we further obtain

$E[(X_{i,j}^s - Y_{i,j}^s)]$

$= E[E[(X_{i,j}^s - Y_{i,j}^s)|G_s, G_s^{obj}]]$

$= E[1_{G_s \text{ contains edge } (i,j) \& G_s^{obj} \text{ does not contain edge } (i,j)} - 1_{(G_s \text{ does not contain edge } (i,j) \& G_s^{obj} \text{ contains edge } (i,j))}]$

$= E[1_{G_s \text{ contains edge } (i,j)}] \cdot E[1_{G_s^{obj} \text{ does not contain edge } (i,j)}] -$

$\qquad E[1_{(G_s \text{ does not contain edge } (i,j))}] \cdot E[1_{G_s^{obj} \text{ contains edge } (i,j)}]$ since $G_s$ and $G_s^{obj}$ are independent

$= \int_A 1_{S \text{ contains edge } (i,j)} \pi^{\tau_1+s}(S|G)dS \cdot \int_A 1_{S \text{ does not contain edge } (i,j)} \pi^*(S)dS -$

$\qquad \int_A 1_{S \text{ does not contain edge } (i,j)} \pi^{\tau_1+s}(S|G)dS \cdot \int_A 1_{S \text{ contains edge } (i,j)} \pi^*(S)dS$

$= \int_A 1_{S \text{ contains edge } (i,j)} \pi^{\tau_1+s}(S|G)dS \cdot \int_A 1_{S \text{ does not contain edge } (i,j)} \pi^*(S)dS -$

$\qquad \int_A 1_{S \text{ contains edge } (i,j)} \pi^*(S)dS \cdot \int_A 1_{S \text{ does not contain edge } (i,j)} \pi^*(S)dS +$

$\qquad \int_A 1_{S \text{ contains edge } (i,j)} \pi^*(S)dS \cdot \int_A 1_{S \text{ does not contain edge } (i,j)} \pi^*(S)dS -$

$\qquad \int_A 1_{S \text{ does not contain edge } (i,j)} \pi^{\tau_1+s}(S|G)dS \cdot \int_A 1_{S \text{ contains edge } (i,j)} \pi^*(S)dS$

$= \int_A 1_{S \text{ does not contain edge } (i,j)} \pi^*(S)dS (\int_A 1_{S \text{ contains edge } (i,j)} (\pi^{\tau_1+s}(S|G) - \pi^*(S))dS +$

$\qquad \int_A 1_{S \text{ contains edge } (i,j)} \pi^*(S)dS (\int_A 1_{S \text{ contains edge } (i,j)} (-\pi^{\tau_1+s}(S|G) + \pi^*(S))dS)$

$\leq \int_A 1_{S \text{ does not contain edge } (i,j)} \pi^*(S)dS \cdot ||\pi^{\tau_1+s}(\cdot|G) - \pi^*(\cdot)||_{TV} +$

$\qquad \int_A 1_{S \text{ contains edge } (i,j)} \pi^*(S)dS \cdot ||\pi^{\tau_1+s}(\cdot|G) - \pi^*(\cdot)||_{TV}$

$= ||\pi^{\tau_1+s}(\cdot|G) - \pi^*(\cdot)||_{TV}$

$\leq 2(p^*)^{\tau_1}.$

In like manner, we achieve

$$E[(-X_{i,j}^s + Y_{i,j}^s)]$$
$$\leq 2(p^*)^{\tau_1}. \tag{4}$$

We now proceed to the analysis on the first term. Though $\{X_{i,j}^s\}_s$ are neither independent or identically distributed random variables, the difference $\frac{\sum_{s=1}^t X_{i,j}^s}{t} - E[X_{i,j}^1]$ can be upper bounded by the convergence property of $\pi^n$. Note that $X_{i,j}^s$ is only different from $X_{i,j}^{s+1}$ when edge $(i,j)$ is sampled at time step $s$ and the generated graph is accepted.

We observe that

$$P(G_{s+1}|G_s, G_{s-1}, \ldots, G_1)$$
$$= P(G_{s+1}|G_s).$$

Meanwhile, we can write $X_{i,j}^{s+1} = 1_{X_{i,j}^{s+1}=1} = 1_{G_{s+1} \text{ contains edge } (i,j)}$ and similarly, $X_{i,j}^s = 1_{X_{i,j}^s=1} = 1_{G_s \text{ contains edge } (i,j)}$. Denote event $E$ as $E = \{\text{connected graph G contains edge } (i,j)\}$. This gives us

$$X_{i,j}^{s+1} = 1_{G_{s+1} \in E},$$
$$X_{i,j}^s = 1_{G_s \in E}.$$

Furthermore, the quantity $\int_A 1_{S \in E} \pi^*(S) dS$ can be simplified as

$$E[Y_{i,j}^1] = E[1_{G_s^{obj} \in E}]$$
$$= \int_A 1_{S \in E} \pi^*(S) dS$$

since $G_s^{obj}$ follows a distribution with density $\pi^*(\cdot)$.

A new Hoeffding lemma for markov chains has been recently shown as follows in [Fan et al., 2021]. Let $a(\lambda) = \frac{1+\lambda}{1-\lambda}$ where $\lambda$ is the spectrum of the Markov chain $CG - MC$ and by the Theorem 2.1 in [Fan et al., 2021], we obtain that

$$P(|\sum_{s=1}^t 1_{G_s \in E} - tE[Y_{i,j}^1]| > t\zeta) \leq 2\exp\{-2a(\lambda)^{-1}t\zeta^2\}$$

$$i.e.P(|\frac{\sum_{s=1}^t X_{i,j}^s}{t} - E[Y_{i,j}^1]| > \zeta) \leq 2\exp\{-2a(\lambda)^{-1}t\zeta^2\} \tag{5}$$

since $X_{i,j}^s = 1_{G_s \in E}$ satisfies $0 \leq 1_{G_s \in E} \leq 1$, i.e. the values are within the range of $[0,1]$.

By the result $E[X_{i,j}^1] - \zeta \leq E[Y_{i,j}^1] \leq E[X_{i,j}^1] + \zeta$, we obtain

$$P(|\frac{\sum_{s=1}^t X_{i,j}^s}{t} - E[X_{i,j}^1]| > 2\zeta) \leq 2\exp\{-2a(\lambda)^{-1}t\zeta^2\}. \tag{6}$$

Putting the results (3), (4) and (6) together, we derive

$$||P_t - P_t^{obj}||_\infty$$
$$= \max_{i,j}|\frac{\sum_{s=1}^t X_{i,j}^s}{t} - E[X_{i,j}^1] + E[X_{i,j}^1 - Y_{i,j}^1] + E[Y_{i,j}^1] - \frac{\sum_{s=1}^t Y_{i,j}^s}{t}|$$
$$\leq \max_{i,j}(|\frac{\sum_{s=1}^t X_{i,j}^s}{t} - E[X_{i,j}^1]| + |E[X_{i,j}^1 - Y_{i,j}^1]| + |E[Y_{i,j}^1] - \frac{\sum_{s=1}^t Y_{i,j}^s}{t}|$$
$$\leq 2\zeta + \zeta + \zeta = 4\zeta$$

which holds with probability at least $1 - 2\exp\{-2a(\lambda)^{-1}t\zeta^2\} - 2\exp\{-2t\zeta^2\}$.

For the probability term $1 - 2\exp\{-2a(\lambda)^{-1}t\zeta^2\} - 2\exp\{-2t\zeta^2\}$, we have

$$2\exp\{-2a(\lambda)^{-1}t\zeta^2\} \leq \frac{\epsilon}{T},$$
$$2\exp\{-2t\zeta^2\} \leq \frac{\epsilon}{T}$$

which holds by

$$t \geq L_{s_2} - \tau_1 = a(\lambda)\frac{\ln\frac{T}{2\epsilon}}{2\zeta^2}.$$

Therefore, the distance between the empirical matrix and the constant matrix reads as with probability at least $1 - 2\frac{\epsilon}{T}$,

$$||P_t - cE||_\infty$$
$$\leq ||P_t - P_t^{obj}||_\infty + ||P_t^{obj} - cE||_\infty$$
$$\leq 4\zeta + \zeta = 5\zeta < \delta$$

where $\zeta = 2(p^*)^{\tau_1} < \frac{1}{5}\delta$ by the choice of parameter $\delta$ and $\zeta$. This completes the proof of Proposition 3.

$\square$

Next, we proceed to explicitly characterize the spectrum of $CG - MC$ which plays a role in the length of burning period $L_{s_2}$ and $L_{s_3}$.

**Proposition 4.** *In setting 2, the spectral gap* $1 - \lambda$ *of* $CG - MC$ *satisfies that for* $\theta > 1$,

$$1 - \lambda \geq \frac{1}{2^{\frac{\ln \theta}{\ln 2p^*}} \ln 2\theta + 1}.$$

*Proof.* It is worth noting that for $G \neq G'$ and $|E(G)\Delta E(G')| = 1$, we have

$$\pi^*(G)\pi(G'|G) = \pi^*(G')\pi(G|G')$$

by the fact that $\pi^*(G) = \pi^*(G')$ and $\pi(G'|G) = \pi(G|G') = \frac{2}{M(M-1)}$.

For $G \neq G'$ and $|E(G)\Delta E(G')| > 1$ or $|E(G)\Delta E(G')| = 0$, we have $\pi^*(G)\pi(G'|G) = \pi^*(G')\pi(G|G')$ since $\pi(G'|G) = \pi(G|G') = 0$.

For $G = G'$, we have $\pi^*(G)\pi(G'|G) = \pi^*(G')\pi(G|G')$ by the expression.

As a result, $CG - MC$ is reversible. Meanwhile, it is ergodic since it has a stationary distribution $\pi^*$ as stated in Proposition 2.

Henceforth, by the result of Theorem 1 in [McNew, 2011] that holds for any ergodic and reversible Markov chain, we have

$$\frac{1}{2 \ln 2e} \frac{\lambda_2}{1 - \lambda_2} \leq \tau(e)$$

where $\tau(e)$ is the mixing time for an error tolerance $e$ and $\lambda_2$ is the second largest eigenvalue of $CG - MC$. Choosing $e > \frac{1}{2}$ immediately gives us

$$\lambda_2 \leq \frac{2\tau(e) \ln 2 \frac{1}{5}\delta}{2\tau(e) \ln 2e + 1}. \tag{7}$$

Again by Proposition 2, we have

$$||\pi^{\tau(e)}(\cdot|G) - \pi^*(\cdot)||_{TV} \leq 2(p^*)^{\tau(e)} = e.$$

Consequently, we arrive at

$$\tau(e) = \frac{\ln e}{\ln 2p^*}$$

and subsequently

$$\lambda_2 \leq \frac{2^{\frac{\ln e}{\ln 2p^*}} \ln 2e}{2^{\frac{\ln e}{\ln 2p^*}} \ln 2e + 1}.$$

by plugging $\tau(e)$ into (7).

This completes the proof of the lower bound on the spectral gap $1 - \lambda_2$.

$\square$

In the following proposition, we show the sufficient condition for graphs generated by the E-R model being connected.

**Proposition 5.** *Assume $c$ in setting 1 meets the condition*

$$1 \geq c \geq \frac{1}{2} + \frac{1}{2}\sqrt{1 - (\frac{\epsilon}{MT})^{\frac{2}{M-1}}},$$

*where $0 < \epsilon < 1$. Then, with probability $1 - \epsilon$, for any $t > 0$, $G_t$ following the E-R model is connected.*

*Proof.* For $1 \leq j \leq M$, we denote the degree of client $j$ as $d_j$.

It is straightforward to have 1) $\sum_{j=1}^{M} d_j = 2 \cdot$ total number of edegs, 2) $E[\text{total number of edges}] = c \cdot \frac{M(M-1)}{2}$ and 3) random variables $d_1, d_2, \ldots, d_M$ are dependent but follow the same distribution.

Note that $d_j$ follows a binomial distribution with $E[d_j] = c \cdot (M-1)$ where $c$ is the probability of an edge. Then by the Chernoff Bound inequality, we have

$$P(d_j < \frac{M-1}{2}) \leq \exp\{-(M-1) \cdot KL(0.5||c)\}$$

where $KL(0.5||c)$ denotes the KL divergence between Bernoulli(0.5) and Bernoulli($c$).

For the term $KL(0.5||c)$, we can further show that

$$KL(0.5||c) = \frac{1}{2}\log\frac{\frac{1}{2}}{c} + \frac{1}{2}\log\frac{\frac{1}{2}}{1-c} = \frac{1}{2}\log\frac{1}{4c(1-c)}$$

which leads to $P(d_j < \frac{M-1}{2}) \leq \exp\{(M-1) \cdot \frac{1}{2}\log 4c(1-c)\}$.

Meanwhile, we have specified the choice of $c$ as

$$\frac{1}{2} + \frac{1}{2}\sqrt{1 - (\frac{\epsilon}{MT})^{\frac{2}{M-1}}}\} \leq c < 1$$

which guarantees $\exp\{(M-1) \cdot \frac{1}{2}\log 4c(1-c)\} \leq \frac{\epsilon}{MT}$ as follow. We observe that

$$c \geq \frac{1}{2} + \frac{1}{2}\sqrt{1 - (\frac{\epsilon}{MT})^{\frac{2}{M-1}}}$$

$$\implies 4c(1-c) \leq (\frac{\epsilon}{MT})^{\frac{2}{M-1}}$$

$$\implies \log 4c(1-c) \leq \frac{2\log\frac{\epsilon}{MT}}{M-1}$$

$$\implies (M-1) \cdot \frac{1}{2}\log 4c(1-c) \leq \log\frac{\epsilon}{MT}$$

$$\implies \exp\{(M-1) \cdot \frac{1}{2}\log 4c(1-c)\} \leq \frac{\epsilon}{MT}.$$

This is summarized as for any $j$

$$P(d_j < \frac{M-1}{2}) \leq \exp\{(M-1) \cdot \frac{1}{2}\log 4c(1-c)\} \leq \frac{\epsilon}{MT}. \tag{8}$$

Meanwhile, it is known as if $\delta(G_t) \geq \frac{M-1}{2}$, then we have that graph $G_t$ is connected where $\delta(G_t) = \min_m d_m$.

As a result, consider the probability and we obtain that

$$P(\text{graph } G_t \text{ is connected})$$

$$\geq P(\min_j d_j \geq \frac{M-1}{2})$$

$$= P(\bigcap_j \{d_j \geq \frac{M-1}{2}\})$$

$$= 1 - P(\bigcup_j \{d_j < \frac{M-1}{2}\})$$

$$\geq 1 - \sum_j P(d_j < \frac{M-1}{2})$$

$$= 1 - MP(d_j < \frac{M-1}{2})$$

$$\geq 1 - M\frac{\epsilon}{MT} = 1 - \frac{\epsilon}{T}$$

where the second inequality holds by the Bonferroni's inequality and the third inequality uses (8).

Consequently, we obtain

$$
\begin{aligned}
&P(\text{graph } G_t \text{ is connected}) \\
&= P(\cap_t \{G_t \text{ is connected}\}) \\
&\geq 1 - \sum_t P(G_t \text{ is not connected}) \\
&= 1 - \sum_t (1 - P(G_t \text{ is connected})) \\
&\geq 1 - \sum_t (1 - (1 - \frac{\epsilon}{T}) = 1 - \epsilon
\end{aligned}
$$

where the first inequality holds again by the Bonferroni's inequality and the second inequality results from the above derivation.

This completes the proof.

$\square$

On graphs with the established properties, we next show the results on the transmission gap between two consecutive rounds of communication for any two clients and the number of arm pulls for all clients.

**Proposition 6.** *We have that with probability $1 - \epsilon$, for any $t > L$ and any $m$, there exists $t_0$ such that*

$$
t + 1 - \min_j t_{m,j} \leq t_0, t_0 \leq c_0 \min_l n_{l,i}(t + 1)
$$

*where $c_0 = c_0(K, \min_{i \neq i^*} \Delta_i, M, \epsilon, \delta)$.*

*Proof.* The edges in setting 1 follow a Bernoulli distribution with a given parameter $c$ by definition. Though setting 2 does not explicitly define the edge distribution, the probability of an edge existing in a connected graph, denoted as $c$, is deterministic, independent of time since graphs are i.i.d. over time and homogeneous among edges.

Henceforth, it is straightforward that $c$ satisfies

$$
\frac{M(M - 1)}{2} c = E(N)
$$

and equivalently $c = \frac{2E(N)}{M(M-1)}$ where $N$ denotes the number of edges in a random connected graph.

We observe that $0 \leq N \leq \frac{M(M-1)}{2}$. Furthermore, the existing result in [Trevisan] yields

$$
E[N] = M \log M.
$$

Consequently, the probability term $c$ has an explicit expressions $c = \frac{2E[N]}{M(M-1)} = \frac{2 \log M}{M-1}$.

For setting $s_2, S_2$ we have $c = \frac{2 \log M}{M-1} \geq \frac{1}{2}$ since $M < 10$, while in setting $s_1, S_1$, the condition on $c$ guarantees $c > \frac{1}{2}$. Note that $t + 1 - t_{m,j}$ follows a geometric distribution since each edge follows a Bernoulli distribution, which holds by

$$
\begin{aligned}
&P(t + 1 - t_{m,j} = 1 | t_{m,j}) \\
&= \frac{P(\text{there is an edge between m and j at time step t+1 and } t_{m,j})}{P(\text{there is an edge between m and j at time step } t_{m,j})} \\
&= \frac{(c)^2}{c} = c
\end{aligned}
$$

and

$$P(t + 1 - t_{m,j} = k | t_{m,j})$$

$$= \frac{P(\text{there is an edge between m and j at time step t+k and } t_{m,j}, \text{ no edge at time step} t + 1, \ldots, t + k - 1)}{P(\text{there is an edge between m and j at time step } t_{m,j})}$$

$$= \frac{(1 - c)^{k-1} c^2}{c} = c(1 - c)^{k-1}.$$

Note that $P(t + 1 - \min_j t_{m,j} \geq t_0)$ which denotes the tail of a geometric distribution depends on the choice of $c$. More precisely, the tail probability $P_0$ is monotone decreasing in $c$.

When $c = \frac{1}{2}$, we obtain that

$$P_0 = P(t + 1 - \min_j t_{m,j} > t_0) \leq \sum_{s > t_0} (\frac{1}{2})^s \leq (\frac{1}{2})^{t_0}. \tag{9}$$

Choosing $t_0 = \frac{\ln \frac{M^2 T}{\epsilon}}{\ln 2}$ leads to $P_0 = 1 - (\frac{1}{2})^{t_0} = 1 - \frac{\epsilon}{M^2 T}$ and

$$c_0 \min_l n_{l,i}(t + 1) \geq c_0 \min_l n_{l,i}(L)$$

$$\geq c_0 \frac{L}{K} \geq c_0 \frac{\ln \frac{M^2 T}{\epsilon}}{c_0 \ln 2} = \frac{\ln \frac{M^2 T}{\epsilon}}{\ln 2} = t_0$$

where the last inequality holds by the choice of $L$. This implies $\min_l n_{l,i}(t + 1) - t_0 \geq (1 - c_0) \min_l n_{l,i}(t + 1)$, i.e. $t_0 \leq c_0 \min_l n_{l,i}(t + 1)$.

Therefore, with probability $1 - \frac{\epsilon}{M^2 T}$,

$$\min_l n_{l,i}(t_{m,l})$$
$$\geq \min_l n_{l,i}(\min_j t_{m,j})$$
$$\geq \min_l n_{l,i}(t + 1 - t_0)$$
$$\geq \min_l n_{l,i}(t + 1) - t_0$$
$$\geq (1 - c_0) \cdot \min_l n_{l,i}(t + 1)$$

where the first inequality results from the fact $t_{m,l} \geq \min_j t_{m,j}$, the second inequality uses the fact from (9), the third inequality applies the definition of $n$, and the last inequality holds by the choice of $t_0$.

Consider setting $s_3, S_3$ where $M > 10$. Generally, for a given parameter $c$, we obtain

$$P(t + 1 - \min_j t_{m,j} = 1) = c,$$
$$P(t + 1 - \min_j t_{m,j} = 2) = c(1 - c),$$
$$\ldots,$$
$$P(t + 1 - \min_j t_{m,j} = n) = c(1 - c)^{n-1}$$

and subsequently

$$P_0 = P(t + 1 - \min_j t_{m,j} > t_0) \leq \sum_{s > t_0} c(1 - c)^{s-1} \leq c(\frac{1}{c} - \frac{1 - (1 - c)^{t_0}}{c}) = (1 - c)^{t_0}.$$

For the probability term $P_0$, we further arrive at

$$P_0 \geq 1 - \frac{\epsilon}{M^2 T}$$

by the choice of $t_0 \geq \frac{\ln(\frac{\epsilon}{M^2 T})}{\ln(1-c)}$.

Meanwhile, we claim that the choice of $t_0$ satisfies

$$t_0 \leq c_0 \min_l n_{l,i}(t+1)$$

since $\frac{\ln(\frac{\epsilon}{M^2 T})}{\ln(1-c)} \leq c_0 \min_l n_{l,i}(t+1)$ holds by noting $n_{l,i}(t+1) \geq n_{l,i}(L) \geq \frac{L}{K}$ and

$$L \geq \frac{K \ln(\frac{\epsilon}{M^2 T})}{c_0 \ln(1-c)} = \frac{K \ln(\frac{M^2 T}{\epsilon})}{c_0 \ln(\frac{1}{1-c})}.$$

To summarize, in all the settings, we have that with probability at least $1 - \frac{\epsilon}{M^2 T}$,

$$t + 1 - \min_j t_{m,j} \leq t_0,$$
$$t_0 \leq c_0 \min_l n_{l,i}(t+1). \tag{10}$$

Therefore, we obtain that in setting $s_1, S_1$

$$P(\forall m, t+1 - \min_j t_{m,j} \leq t_0 \leq c_0 \min_l n_{l,i}(t+1))$$
$$\doteq (1 - P_0)^M \geq 1 - M P_0 = 1 - \frac{\epsilon}{MT} \tag{11}$$

where the inequality is a result of the Bernoulli's inequality.

In setting $s_2, S_2, s_3, S_3, \{t+1 - \min_j t_{1,j}, \ldots, t+1 - \min_j t_{M,j}\}$ follow the same distribution, but are dependent since they construct a connected graph. However, we have the following result

$$P(\forall m, t+1 - \min_j t_{m,j} \leq t_0 \leq c_0 \min_l n_{l,i}(t+1))$$
$$= 1 - P(\cup_m \{t+1 - \min_j t_{m,j} \geq t_0\})$$
$$\geq 1 - \sum_m P(t+1 - \min_j t_{m,j} \geq t_0)$$
$$= 1 - M P_0 = 1 - \frac{\epsilon}{MT} \tag{12}$$

by the Bonferroni's inequality.

As a consequence, we arrive at that in setting $s_1, S_1, s_2, S_2, s_3, S_3$,

$$P(\forall t, \forall m, t+1 - \min_j t_{m,j} \leq t_0 \leq c_0 \min_l n_{l,i}(t+1))$$
$$\geq 1 - \sum_t \sum_m P(t+1 - \min_j t_{m,j} \leq t_0 \leq c_0 \min_l n_{l,i}(t+1))$$
$$\geq 1 - MT(1 - (1 - \frac{\epsilon}{MT})) = 1 - \epsilon$$

where the first inequality again uses the Bonferroni's inequality and the second inequality holds by applying (11) and (12).

$\square$

After establishing the transmissions among clients, we next proceed to show the concentration properties of the network-wide estimators maintained by the clients.

The first is to demonstrate the unbiasedness of these estimators with respect to the global expected rewards.

**Proposition 7.** *Assume the parameter $\delta$ satisfies that $0 < \delta < c = f(\epsilon, M, T)$. For any arm $i$ and any client $m$, at every time step $t$, we have*

$$E[\tilde{\mu}_i^m(t)|A_{\epsilon,\delta}] = \mu_i.$$

*Proof.* The result can be shown by induction as follows. We start with the basis step by considering any time step $t \leq L + 1$. By the definition of $\tilde{\mu}_i^m(t) = \tilde{\mu}_i^m(L+1)$, we arrive at

$$
\begin{aligned}
&E[\tilde{\mu}_i^m(t)|A_{\epsilon,\delta}]\\
&= E[\tilde{\mu}_i^m(L+1)|A_{\epsilon,\delta}]\\
&= E[\sum_{j=1}^{M} P'_{m,j}(L)\hat{\tilde{\mu}}_{i,j}^m(h_{m,j}^L)|A_{\epsilon,\delta}]
\end{aligned}
\tag{13}
$$

where $P'_{m,j}(L) = \begin{cases} \frac{1}{M}, & \text{if } P_L(m,j) > 0 \\ 0, & \text{else} \end{cases}$. The definition of $A_{\epsilon,\delta}$ and the choice of $\delta$ guarantee that $|P_L - cE| < \delta < c$ on event $A_{\epsilon,\delta}$, i.e. we have for any $t \geq L$, $P_t > 0$ and thereby obtaining

$$
P'_{m,j}(L) = \frac{1}{M}.
\tag{14}
$$

Therefore, we continue with (13) and have

$$
\begin{aligned}
(13) &= E[\sum_{j=1}^{M} \frac{1}{M}\hat{\tilde{\mu}}_{i,j}^m(h_{m,j}^L)|A_{\epsilon,\delta}]\\
&= \frac{1}{M}\sum_{j=1}^{M} E[\bar{\mu}_i^j(h_{m,j}^L)|A_{\epsilon,\delta}]\\
&= \frac{1}{M}\sum_{j=1}^{M} E[\frac{\sum_s r_i^j(s)}{n_{j,i}(h_{m,j}^L)}|A_{\epsilon,\delta}]\\
&= \frac{1}{M}\sum_{j=1}^{M} E[E[\frac{\sum_s r_i^j(s)}{n_{j,i}(h_{m,j}^L)}|\sigma(n_{j,i}(l))_{l \leq h_{m,j}^L}, A_{\epsilon,\delta}|A_{\epsilon,\delta}]
\end{aligned}
$$

where the last equality uses the law of total expectation.

With the derivations, we further have

$$
\begin{aligned}
(13) &= \frac{1}{M}\sum_{j=1}^{M} E[\frac{1}{n_{j,i}(h_{m,j}^L)}E[\sum_{s:n_{j,i}(s)-n_{j,i}(s-1)=1} r_i^j(s)|\sigma(n_{j,i}(l))_{l \leq h_{m,j}^L}, A_{\epsilon,\delta}|A_{\epsilon,\delta}]\\
&= \frac{1}{M}\sum_{j=1}^{M} E[\frac{1}{n_{j,i}(h_{m,j}^L)}\sum_{s:n_{j,i}(s)-n_{j,i}(s-1)=1} E[r_i^j(s)|\sigma(n_{j,i}(l))_{l \leq h_{m,j}^L}, A_{\epsilon,\delta}|A_{\epsilon,\delta}] \qquad (15)\\
&= \frac{1}{M}\sum_{j=1}^{M} E[\frac{1}{n_{j,i}(h_{m,j}^L)}\sum_{s:n_{j,i}(s)-n_{j,i}(s-1)=1} \mu_i^j|A_{\epsilon,\delta}] \qquad (16)\\
&= \frac{1}{M}\sum_{j=1}^{M} E[\mu_i^j|A_{\epsilon,\delta}] = \mu_i \qquad (17)
\end{aligned}
$$

where the second equality (15) uses the fact that $\{s : n_{j,i}(s) - n_{j,i}(s-1) = 1\}$ is contained in $\sigma(n_{j,i}(l))_{l \leq L_{m,j}}$ and the third equality (16) results from that $r_i^j(s)$ is independent of everything else given $s$ and $E[r_i^j(s)] = \mu_i^j$.

The induction step follows a similar analysis as follows. Suppose that for any $s \leq t$ we have $E[\tilde{\mu}_i^m(s)|A_{\epsilon,\delta}] = \mu_i$.

For time step $t+1$, we first write it as

$$E[\tilde{\mu}_i^m(t+1)|A_{\epsilon,\delta}]$$

$$= E[\sum_{j=1}^{M} P'_t(m,j)\hat{\tilde{\mu}}_{i,j}^m(t_{m,j}) + d_{m,t}\sum_{j\in N_m(t)}\hat{\tilde{\mu}}_{i,j}^m(t) + d_{m,t}\sum_{j\notin N_m(t)}\hat{\tilde{\mu}}_{i,j}^m(t_{m,j})|A_{\epsilon,\delta}]$$

$$= E[E[\sum_{j=1}^{M} P'_t(m,j)\tilde{\mu}_i^j(t_{m,j}) + d_{m,t}\sum_{j\in N_m(t)}\bar{\mu}_i^j(t) + d_{m,t}\sum_{j\notin N_m(t)}\bar{\mu}_i^j(t_{m,j})|\sigma(n_{j,i}(t))_{j,i,t}, A_{\epsilon,\delta}|A_{\epsilon,\delta}]$$

$$\tag{18}$$

where $P'_t(m,j)$ and $d$ are constants since $P_t(m,j) > 0$ for $t \geq L$ on event $A_{\epsilon,\delta}$ and the last equality is again by the law of total expectation.

This gives us that by the law of total expectation

$$(18) = E[\sum_{j=1}^{M} P'_t(m,j)E[\tilde{\mu}_i^j(t_{m,j})|\sigma(n_{j,i}(t))_{j,i,t}, A_{\epsilon,\delta}+$$

$$d_{m,t}\sum_{j\in N_m(t)} E[\bar{\mu}_i^j(t)|\sigma(n_{j,i}(t))_{j,i,t}, A_{\epsilon,\delta}+$$

$$d_{m,t}\sum_{j\notin N_m(t)} E[\bar{\mu}_i^j(t_{m,j})|\sigma(n_{j,i}(t))_{j,i,t}, A_{\epsilon,\delta}|A_{\epsilon,\delta}]$$

$$= \sum_{j=1}^{M} P'(m,j)E[E[\tilde{\mu}_i^j(t_{m,j})|\sigma(n_{j,i}(t))_{j,i,t}, A_{\epsilon,\delta}|A_{\epsilon,\delta}]+$$

$$E[d_{m,t}\sum_{j\in N_m(t)} E[\bar{\mu}_i^j(t)|\sigma(n_{j,i}(t))_{j,i,t}, A_{\epsilon,\delta}+$$

$$d_{m,t}\sum_{j\notin N_m(t)} E[\bar{\mu}_i^j(t_{m,j})|\sigma(n_{j,i}(t))_{j,i,t}, A_{\epsilon,\delta}|A_{\epsilon,\delta}]$$

$$= \sum_{j=1}^{M} P'(m,j)E[\tilde{\mu}_i^j(t_{m,j})|A_{\epsilon,\delta}] + E[d_{m,t}\sum_{j} E[\bar{\mu}_i^j(t_{m,j})|\sigma(n_{j,i}(t))_{j,i,t}, A_{\epsilon,\delta}|A_{\epsilon,\delta}]$$

$$= \sum_{j=1}^{M} P'(m,j)E[\tilde{\mu}_i^j(t_{m,j})|A_{\epsilon,\delta}]+$$

$$d_{m,t}\sum_{j} E[E[\frac{1}{n_{j,i}(t_{m,j})}\sum_{s:n_{j,i}(s)-n_{j,i}(s-1)=1} E[r_i^j(s)|\sigma(n_{j,i}(t))_{j,i,t}, A_{\epsilon,\delta}|A_{\epsilon,\delta}]$$

$$= \sum_{j=1}^{M} P'(m,j)\mu_i + d_{m,t}M\mu_i = (\sum_{j=1}^{M} P'(m,j) + Md_{m,t})\mu_i = \mu_i$$

where the first equality uses $P'_t(m,j)$ and $d$ are constants on event $A_{\epsilon,\delta}$, the second equality is derived by re-organizing the terms, the third equality again uses the law of total expectation and integrates the second term by $t_{m,j}$, the fourth equality elaborate the second term and the equality in the last line follows from the induction and (15, 16 17).

This completes the induction step and thus shows the unbiasedness of the network-wide estimators conditional on event $A_{\epsilon,\delta}$.

$$\square$$

Then we characterize the moment generating functions of the network-wide estimators and conclude that they have similar properties as their local rewards.

**Proposition 8.** *Assume the parameter $\delta$ satisfies that $0 < \delta < c = f(\epsilon, M, T)$. In setting $s_1, s_2, s_3$ where rewards follow sub-gaussian distributions, for any $m, i, \lambda$ and $t > L$ where $L$ is the length of*

*the burn-in period, the global estimator $\tilde{\mu}_i^m(t)$ is sub-Gaussian distributed. Moreover, the conditional moment generating function satisfies that with $P(A_{\epsilon,\delta}) = 1 - 7\epsilon$,*

$$E[\exp\{\lambda(\tilde{\mu}_i^m(t) - \mu_i)\}1_{A_{\epsilon,\delta}}|\sigma(\{n_{m,i}(t)\}_{t,i,m})]$$
$$\leq \exp\{\frac{\lambda^2}{2}\frac{C\sigma^2}{\min_j n_{j,i}(t)}\}$$

*where $\sigma^2 = \max_{j,i}(\tilde{\sigma}_i^j)^2$ and $C = \max\{\frac{4(M+2)(1-\frac{1-c_0}{2(M+2)})^2}{3M(1-c_0)}, (M+2)(1+4Md_{m,t}^2)\}$.*

*Proof.* We prove the statement on the conditional moment generating functions by induction. Let us start with the basis step.

Note that the definition of $A_{\epsilon,\delta}$ and the choice of $\delta$ again guarantee that for $t \geq L$, $|P_t - cE| < \delta < c$ on event $A_{\epsilon,\delta}$. This implies that for any $t \geq L$, $m$ and $j$, $P_t(m,j) > 0$, and if $t = L$

$$P_t'(m,j) = \frac{1}{M} \tag{19}$$

and if $t > L$

$$P_t'(m,j) = \frac{M-1}{M^2}. \tag{20}$$

Consider the time step $t \leq L + 1$. The quantity satisfies that

$$E[\exp\{\lambda(\tilde{\mu}_i^m(t) - \mu_i)\}1_{A_{\epsilon,\delta}}|\sigma(\{n_{m,i}(t)\}_{t,i,m})$$
$$= E[\exp\{\lambda(\tilde{\mu}_i^m(L+1) - \mu_i)\}1_{A_{\epsilon,\delta}}|\sigma(\{n_{m,i}(t)\}_{t,i,m})$$
$$= E[\exp\{\lambda(\sum_{j=1}^M P_{m,j}'(L)\hat{\tilde{\mu}}_{i,j}^m(h_{m,j}^L) - \mu_i)\}1_{A_{\epsilon,\delta}}|\sigma(\{n_{m,i}(t)\}_{t,i,m})]$$
$$= E[\exp\{\lambda(\sum_{j=1}^M \frac{1}{M}\hat{\tilde{\mu}}_{i,j}^m(h_{m,j}^L) - \mu_i)\}1_{A_{\epsilon,\delta}}|\sigma(\{n_{m,i}(t)\}_{t,i,m})]$$
$$= E[\exp\{\lambda\sum_{j=1}^M \frac{1}{M}(\hat{\tilde{\mu}}_{i,j}^m(h_{m,j}^L) - \mu_i^j)\}1_{A_{\epsilon,\delta}}|\sigma(\{n_{m,i}(t)\}_{t,i,m})]$$
$$\leq \Pi_{j=1}^M (E[(\exp\{(\lambda\frac{1}{M}(\bar{\mu}_i^j(h_{m,j}^L) - \mu_i^j)\}1_{A_{\epsilon,\delta}})^M|\sigma(\{n_{m,i}(t)\}_{t,i,m})])^{\frac{1}{M}} \tag{21}$$

where the third equality holds by (19), the fourth equality uses the definition $\mu_i = \frac{1}{M}\sum_{i=1}^M \mu_i^j$, and the last inequality results from the generalized hoeffding inequality as in Lemma 3 and the fact that $\hat{\tilde{\mu}}_{i,j}^m(h_{m,j}^L) = \bar{\mu}_i^j(h_{m,j}^L)$.

Note that for any client $j$, we have

$$E[(\exp\{(\lambda\frac{1}{M}(\bar{\mu}_i^j(h_{m,j}^L) - \mu_i^j)\}1_{A_{\epsilon,\delta}})^M|\sigma(\{n_{m,i}(t)\}_{t,i,m}))]$$
$$= E[\exp\{(\lambda(\bar{\mu}_i^j(h_{m,j}^L) - \mu_i^j)\}1_{A_{\epsilon,\delta}})|\sigma(\{n_{m,i}(t)\}_{t,i,m})]$$
$$= E[\exp\{(\lambda\frac{\sum_s(r_i^j(s) - \mu_i^j)}{n_{j,i}(h_{m,j}^L)}\}1_{A_{\epsilon,\delta}})|\sigma(\{n_{m,i}(t)\}_{t,i,m})]$$
$$= E[\exp\{\sum_s(\lambda\frac{(r_i^j(s) - \mu_i^j)}{n_{j,i}(h_{m,j}^L)}\}1_{A_{\epsilon,\delta}})|\sigma(\{n_{m,i}(t)\}_{t,i,m})]. \tag{22}$$

It is worth noting that given $s$, $r_i^j(s)$ is independent of everything else, which gives us

$$(22) = \Pi_s E[\exp\{\lambda \frac{(r_i^j(s) - \mu_i^j)}{n_{j,i}(h_{m,j}^L)}\} 1_{A_{\epsilon,\delta}} | \sigma(\{n_{m,i}(t)\}_{t,i,m})]$$

$$= \Pi_s E[\exp\{\lambda \frac{(r_i^j(s) - \mu_i^j)}{n_{j,i}(h_{m,j}^L)}\} | \sigma(\{n_{m,i}(t)\}_{t,i,m})] \cdot E[1_{A_{\epsilon,\delta}} | \sigma(\{n_{m,i}(t)\}_{t,i,m})]$$

$$= \Pi_s E_r[\exp\{\lambda \frac{(r_i^j(s) - \mu_i^j)}{n_{j,i}(h_{m,j}^L)}\}] \cdot E[1_{A_{\epsilon,\delta}} | \sigma(\{n_{m,i}(t)\}_{t,i,m})]$$

$$\leq \Pi_s \exp\{\frac{(\frac{\lambda}{n_{j,i}(h_{m,j}^L)})^2 \sigma^2}{2}\} \cdot E[1_{A_{\epsilon,\delta}} | \sigma(\{n_{m,i}(t)\}_{t,i,m})]$$

$$\leq (\exp\{\frac{(\frac{\lambda}{n_{j,i}(h_{m,j}^L)})^2 \sigma^2}{2}\})^{n_{j,i}(h_{m,j}^L)}$$

$$= \exp\{\frac{\frac{\lambda^2}{n_{j,i}(h_{m,j}^L)} \sigma^2}{2}\}$$

$$\leq \exp\{\frac{\lambda^2 \sigma^2}{2 \min_j n_{j,i}(h_{m,j}^L)}\} \qquad (23)$$

where the first inequality holds by the definition of sub-Gaussian random variables $r_i^j(s) - \mu_i^j$ with an mean value 0, the second inequality results from $1_{A_{\epsilon,\delta}} \leq 1$, and the last inequality uses $n_{j,i}(h_{m,j}^L) \geq \min_j n_{j,i}(h_{m,j}^L)$ for any $j$.

Therefore, we obtain that by plugging (23) into (21)

$$(21) \leq \Pi_{j=1}^M (\exp\{\frac{\lambda^2 \sigma^2}{2 \min_j n_{j,i}(h_{m,j}^L)}\})^{\frac{1}{M}}$$

$$= ((\exp\{\frac{\lambda^2 \sigma^2}{2 \min_j n_{j,i}(h_{m,j}^L)}\})^{\frac{1}{M}})^M$$

$$= \exp\{\frac{\lambda^2 \sigma^2}{2 \min_j n_{j,i}(h_{m,j}^L)}\}$$

which completes the basis step.

Now we proceed to the induction step. Suppose that for any $s < t + 1$ where $t \geq L$, we have

$$E[\exp\{\lambda(\tilde{\mu}_i^m(s) - \mu_i)\} 1_{A_{\epsilon,\delta}} | \sigma(\{n_{m,i}(s)\}_{s,i,m})]$$

$$\leq \exp\{\frac{\lambda^2}{2} \frac{C\sigma^2}{\min_j n_{j,i}(s)}\}. \qquad (24)$$

The update rule of $\tilde{\mu}_i^m$ implies that

$$E[\exp\{\lambda(\tilde{\mu}_i^m(t+1) - \mu_i)\}1_{A_{\epsilon,\delta}}|\sigma(\{n_{m,i}(s)\}_{s,i,m})]$$

$$= E[\exp\{\lambda(\sum_{j=1}^M P'_t(m,j)(\hat{\tilde{\mu}}_{i,j}^m(t_{m,j}) - \mu_i) + d_{m,t}\sum_{j\in N_m(t)}(\hat{\tilde{\mu}}_{i,j}^m(t) - \mu_i^j)$$

$$+ d_{m,t}\sum_{j\notin N_m(t)}(\hat{\tilde{\mu}}_{i,j}^m(t_{m,j}) - \mu_i^j))\}1_{A_{\epsilon,\delta}}|\sigma(\{n_{m,i}(s)\}_{s,i,m})]$$

$$= E[\exp\{\lambda(\sum_{j=1}^M P'_t(m,j)(\tilde{\mu}_i^j(t_{m,j}) - \mu_i) + d_{m,t}\sum_{j\in N_m(t)}(\bar{\mu}_i^j(t) - \mu_i^j)$$

$$+ d_{m,t}\sum_{j\notin N_m(t)}(\bar{\mu}_i^j(t_{m,j}) - \mu_i^j))\}1_{A_{\epsilon,\delta}}|\sigma(\{n_{m,i}(s)\}_{s,i,m})]$$

$$= E[\Pi_{j=1}^M \exp\{\lambda P'_t(m,j)(\tilde{\mu}_i^j(t_{m,j}) - \mu_i)\}1_{A_{\epsilon,\delta}} \cdot \Pi_{j\in N_m(t)}\exp\{\lambda d_{m,t}(\bar{\mu}_i^j(t) - \mu_i^j)\}1_{A_{\epsilon,\delta}}$$

$$\cdot \Pi_{j\notin N_m(t)}\exp\{\lambda d_{m,t}(\bar{\mu}_i^j(t_{m,j}) - \mu_i^j)1_{A_{\epsilon,\delta}}\}|\sigma(\{n_{m,i}(s)\}_{s,i,m})]$$

$$\leq \Pi_{j=1}^M(E[(\exp\{\lambda P'_t(m,j)(\tilde{\mu}_i^j(t_{m,j}) - \mu_i)\})^{M+2}1_{A_{\epsilon,\delta}}|\sigma(\{n_{m,i}(s)\}_{s,i,m})])^{\frac{1}{M+2}} \cdot$$

$$E[\Pi_{j\in N_m(t)}(\exp\{\lambda d_{m,t}(\bar{\mu}_i^j(t) - \mu_i^j)\})^{M+2}1_{A_{\epsilon,\delta}}|\sigma(\{n_{m,i}(s)\}_{s,i,m})]^{\frac{1}{M+2}} \cdot$$

$$E[\Pi_{j\notin N_m(t)}(\exp\{\lambda d_{m,t}(\bar{\mu}_i^j(t_{m,j}) - \mu_i^j)\})^{M+2}1_{A_{\epsilon,\delta}}|\sigma(\{n_{m,i}(s)\}_{s,i,m})]^{\frac{1}{M+2}}$$

$$= \Pi_{j=1}^M(E[(\exp\{\lambda P'_t(m,j)(M+2)(\tilde{\mu}_i^j(t_{m,j}) - \mu_i)\})1_{A_{\epsilon,\delta}}|\sigma(\{n_{m,i}(s)\}_{s,i,m})])^{\frac{1}{M+2}} \cdot$$

$$E[\Pi_{j\in N_m(t)}(\exp\{\lambda d_{m,t}(M+2)(\bar{\mu}_i^j(t) - \mu_i^j)\})1_{A_{\epsilon,\delta}}|\sigma(\{n_{m,i}(s)\}_{s,i,m})]^{\frac{1}{M+2}} \cdot$$

$$E[\Pi_{j\notin N_m(t)}(\exp\{\lambda d_{m,t}(M+2)(\bar{\mu}_i^j(t_{m,j}) - \mu_i^j)\})1_{A_{\epsilon,\delta}}|\sigma(\{n_{m,i}(s)\}_{s,i,m})]^{\frac{1}{M+2}}$$

$$\leq \Pi_{j=1}^M(\exp\{\frac{\lambda^2(P'_t(m,j))^2(M+2)^2}{2}\frac{C\sigma^2}{\min_j n_{j,i}(t_{m,j})}\})^{\frac{1}{M+2}} \cdot$$

$$\Pi_{j\in N_m(t)}\Pi_s(E_r[\exp\{\lambda d_{m,t}(M+2)\frac{(r_i^j(s) - \mu_i^j)}{n_{j,i}(t)}\}] \cdot E[1_{A_{\epsilon,\delta}}|\sigma(\{n_{m,i}(t)\}_{t,i,m})])^{\frac{1}{M+2}} \cdot$$

$$\Pi_{j\notin N_m(t)}\Pi_s(E_r[\exp\{\lambda d_{m,t}(M+2)\frac{(r_i^j(s) - \mu_i^j)}{n_{j,i}(t_{m,j})}\}] \cdot E[1_{A_{\epsilon,\delta}}|\sigma(\{n_{m,i}(t)\}_{t,i,m})])^{\frac{1}{M+2}}$$

$$\tag{25}$$

where the first inequality uses Lemma 3 and the second inequality applies (24) as the assumption for the induction step and holds by exchanging the expectations with the multiplication since again given $s$ the reward $(r_i^j(s) - \mu_i^j)$ is independent of other random variables.

We continue bounding the last two terms by using the definition of sub-Gaussian random variables $(r_i^j(s) - \mu_i^j)$ and obtain

$$(25) \leq (\exp\{\frac{\lambda^2(P'_t(m,j))^2(M+2)^2}{2}\frac{C\sigma^2}{\min_j n_{j,i}(t_{m,j})}\})^{\frac{M}{M+2}} \cdot$$

$$\Pi_{j\in N_m(t)}\Pi_s(\exp\frac{\lambda^2 d_{m,t}^2(M+2)^2\sigma^2}{2n_{j,i}^2(t)} \cdot E[1_{A_{\epsilon,\delta}}|\sigma(\{n_{m,i}(t)\}_{t,i,m})])^{\frac{1}{M+2}} \cdot$$

$$\Pi_{j\notin N_m(t)}\Pi_s(\exp\frac{\lambda^2 d_{m,t}^2(M+2)^2\sigma^2}{2n_{j,i}^2(t_{m,j})} \cdot E[1_{A_{\epsilon,\delta}}|\sigma(\{n_{m,i}(t)\}_{t,i,m})])^{\frac{1}{M+2}}$$

$$= (\exp\{\frac{\lambda^2(P'_t(m,j))^2(M+2)^2}{2}\frac{C\sigma^2}{\min_j n_{j,i}(t_{m,j})}\})^{\frac{M}{M+2}} \cdot$$

$$\Pi_{j\in N_m(t)}\exp\{\frac{n_{j,i}(t)}{M+2}\frac{\lambda^2 d_{m,t}^2(M+2)^2\sigma^2}{2n_{j,i}^2(t)}\} \cdot E[1_{A_{\epsilon,\delta}}|\sigma(\{n_{m,i}(t)\}_{t,i,m})] \cdot$$

$$\Pi_{j\notin N_m(t)}\exp\{\frac{n_{j,i}(t_{m,j})}{M+2}\frac{\lambda^2 d_{m,t}^2(M+2)^2\sigma^2}{2n_{j,i}^2(t_{m,j})}\} \cdot E[1_{A_{\epsilon,\delta}}|\sigma(\{n_{m,i}(t)\}_{t,i,m})]$$

Building on that, we establish

$$(25) \leq \left(\exp\left\{\frac{\lambda^2(P_t'(m,j))^2(M+2)^2}{2}\frac{C\sigma^2}{\min_j n_{j,i}(t_{m,j})}\right\}\right)^{\frac{M}{M+2}} \cdot$$

$$\left(\exp\left\{\frac{\lambda^2 d_{m,t}^2(M+2)\sigma^2}{2\min_j n_{j,i}(t)}\right\}\right)^{|N_m(t)|} \cdot E[1_{A_{\epsilon,\delta}}|\sigma(\{n_{m,i}(t)\}_{t,i,m})] \cdot$$

$$\left(\exp\left\{\frac{\lambda^2 d_{m,t}^2(M+2)\sigma^2}{2\min_j n_{j,i}(t_{m,j})}\right\}\right)^{|M-N_m(t)|} \cdot E[1_{A_{\epsilon,\delta}}|\sigma(\{n_{m,i}(t)\}_{t,i,m})]$$

$$= E\left[\left(\exp\left\{\frac{\lambda^2(P_t'(m,j))^2 M(M+2)}{2}\frac{C\sigma^2}{\min_j n_{j,i}(t_{m,j})}\right\}\right) \cdot \left(\exp\left\{\frac{\lambda^2 d_{m,t}^2(M+2)|N_m(t)|}{2\min_j n_{j,i}(t)}\right\}\right)\right.$$

$$\left. \cdot \left(\exp\left\{\frac{\lambda^2 d_{m,t}^2(M+2)\sigma^2|M-N_m(t)|}{2\min_j n_{j,i}(t_{m,j})}\right\}\right)1_{A_{\epsilon,\delta}}|\sigma(\{n_{m,i}(t)\}_{t,i,m})\right]$$

$$\leq E\left[\left(\exp\left\{\frac{\lambda^2(P_t'(m,j))^2 M(M+2)}{2(1-c_0)}\frac{C\sigma^2}{\min_j n_{j,i}(t+1)}\right\}\right) \cdot \left(\exp\left\{\frac{\lambda^2 d_{m,t}^2(M+2)|N_m(t)|\sigma^2}{2\frac{L/K}{L/K+1}\min_j n_{j,i}(t+1)}\right\}\right)\right.$$

$$\left. \cdot \left(\exp\left\{\frac{\lambda^2 d_{m,t}^2(M+2)|M-N_m(t)|\sigma^2}{2(1-c_0)\min_j n_{j,i}(t+1)}\right\}\right)1_{A_{\epsilon,\delta}}|\sigma(\{n_{m,i}(t)\}_{t,i,m})\right]$$

where the first inequality uses the fact that for any $j$, $n_{j,i}(t) \geq \min_j n_{j,i}(t)$ and $n_{j,i}(t_{m,j}) \geq \min_j n_{j,i}(t_{m,j})$. For the second inequality, the first term is a result of $\frac{\min_j n_{j,i}(t)}{\min_j n_{j,i}(t+1)} \geq \frac{\min_j n_{j,i}(t)}{\min_j n_{j,i}(t)+1} \geq \frac{L/K}{L/K+1}$ since $n_{j,i}(t) > n_{j,i}(L) = L/K$ and the ratio is monotone increasing in $n$, and the second term is bounded based on the following derivations

$$\min_j n_{j,i}(t_{m,j}) \geq \min_j n_{j,i}(t+1-t_0)$$

$$\geq \min_j n_{j,i}(t+1) - t_0$$

$$\geq \min_j n_{j,i}(t+1) - c_0 \min_j n_{j,i}(t+1)$$

$$= (1-c_0)\min_j n_{j,i}(t+1)$$

where the last inequality holds by applying Proposition 6 that holds on event $A_{\epsilon,\delta}$.

Therefore, we can rewrite the above expression as

$$(25) = E\left[\left(\exp\left\{\frac{\lambda^2\sigma^2}{2\min_j n_{j,i}(t+1)} \cdot \left(\frac{C(P_t'(m,j))^2 M(M+2)}{2(1-c_0)} + \right.\right.\right.\right.$$

$$\left.\left.\left.\left. \frac{d_{m,t}^2(M+2)|N_m(t)|}{\frac{L/K}{L/K+1}} + \frac{d_{m,t}^2(M+2)|M-N_m(t)|}{(1-c_0)}\right)\right\}1_{A_{\epsilon,\delta}}|\sigma(\{n_{m,i}(t)\}_{t,i,m})\right]$$

$$\leq E\left[\exp\left\{\frac{C\lambda^2\sigma^2}{2\min_j n_{j,i}(t+1)}\right\}1_{A_{\epsilon,\delta}}|\sigma(\{n_{m,i}(t)\}_{t,i,m})\right]$$

$$\leq \exp\left\{\frac{C\lambda^2\sigma^2}{2\min_j n_{j,i}(t+1)}\right\}$$

where the first inequality holds by the choice of $P_t'(m,j), d_{m,t}, L, c_0$ and $C$ and the second inequality uses the fact that $1_{A_{\epsilon,\delta}} \leq 1$ and $\min_j n_{j,i}(t+1) \in \sigma(\{n_{m,i}(t)\}_{t,i,m})$.

This completes the induction step and subsequently concludes the proof.

$$\square$$

**Proposition 9.** *Assume the parameter $\delta$ satisfies that $0 < \delta < c = f(\epsilon, M, T)$. In setting $s_1, s_2$, and $s_3$, for any $m, i$ and $t > L$ where $L$ is the length of the burn-in period, $\tilde{\mu}_{m,i}(t)$ satisfies that if if*

$n_{m,i}(t) \geq 2(K^2 + KM + M)$, *then with* $P(A_{\epsilon,\delta}) = 1 - 7\epsilon$,

$$P(\tilde{\mu}_{m,i}(t) - \mu_i \geq \sqrt{\frac{C_1 \log t}{n_{m,i}(t)}} | A_{\epsilon,\delta}) \leq \frac{1}{P(A_{\epsilon,\delta})} \frac{1}{t^2},$$

$$P(\mu_i - \tilde{\mu}_{m,i}(t) \geq \sqrt{\frac{C_1 \log t}{n_{m,i}(t)}} | A_{\epsilon,\delta}) \leq \frac{1}{P(A_{\epsilon,\delta}) t^2}.$$

*Proof.* By Proposition 7, we have $E[\tilde{\mu}_{m,i}(t) - \mu_i | A_{\epsilon,\delta}] = 0$, which allows us to consider the tail bound of the global estimator $\tilde{\mu}_i^m(t)$ conditional on event $A_{\epsilon,\delta}$ as follows.

Note that

$$P(\tilde{\mu}_{m,i}(t) - \mu_i \geq \sqrt{\frac{C_1 \log t}{n_{m,i}(t)}} | A_{\epsilon,\delta})$$

$$= E[1_{\tilde{\mu}_{m,i}(t) - \mu_i \geq \sqrt{\frac{C_1 \log t}{n_{m,i}(t)}}} | A_{\epsilon,\delta}]$$

$$= \frac{1}{P(A_{\epsilon,\delta})} E[1_{\tilde{\mu}_{m,i}(t) - \mu_i \geq \sqrt{\frac{C_1 \log t}{n_{m,i}(t)}}} 1_{A_{\epsilon,\delta}}]$$

$$= \frac{1}{P(A_{\epsilon,\delta})} E[1_{\exp\{\lambda(n)(\tilde{\mu}_{m,i}(t) - \mu_i)\} \geq \exp\{\lambda(n)\sqrt{\frac{C_1 \log t}{n_{m,i}(t)}}\}} 1_{A_{\epsilon,\delta}}]$$

$$\leq \frac{1}{P(A_{\epsilon,\delta})} E[\frac{\exp\{\lambda(n)(\tilde{\mu}_{m,i}(t) - \mu_i)\}}{\exp\{\lambda(n)\sqrt{\frac{C_1 \log t}{n_{m,i}(t)}}\}} 1_{A_{\epsilon,\delta}}] \qquad (26)$$

where the last inequality is by the fact that $1_{\exp\{\lambda(n)(\tilde{\mu}_{m,i}(t) - \mu_i)\} \geq \exp\{\lambda(n)\sqrt{\frac{C_1 \log t}{n_{m,i}(t)}}\}} \leq \frac{\exp\{\lambda(n)(\tilde{\mu}_{m,i}(t) - \mu_i)\}}{\exp\{\lambda(n)\sqrt{\frac{C_1 \log t}{n_{m,i}(t)}}\}}$.

By the assumption that $\delta < c$, we have Proposition 8 holds. Subsequently, by Proposition 8 and Lemma 2 which holds since $n_{m,i}(t) \geq 2(K^2 + KM + M)$, we have for any $\lambda$

$$E[\exp\{\lambda(\tilde{\mu}_i^m(t) - \mu_i)\} 1_{A_{\epsilon,\delta}} | \sigma(\{n_{m,i}(t)\}_{t,i,m})] \leq \exp\{\frac{\lambda^2}{2} \frac{C\sigma^2}{\min_j n_{j,i}(t)}\}$$

$$\leq \exp\{\frac{\lambda^2}{1} \frac{C\sigma^2}{n_{m,i}(t)}\}. \qquad (27)$$

Again, we utilize the law of total expectation and further obtain

$$(26) = \frac{1}{P(A_{\epsilon,\delta})} E[E[\frac{\exp\{\lambda(n)(\tilde{\mu}_{m,i}(t) - \mu_i)\}}{\exp\{\lambda(n)\sqrt{\frac{C_1 \log t}{n_{m,i}(t)}}\}} 1_{A_{\epsilon,\delta}} | \sigma(\{n_{m,i}(t)\}_{m,i,t})]]$$

$$= \frac{1}{P(A_{\epsilon,\delta})} E[E[\frac{\exp\{\lambda(n)(\tilde{\mu}_{m,i}(t) - \mu_i)\}}{\exp\{\lambda(n)\sqrt{\frac{C_1 \log t}{n_{m,i}(t)}}\}} 1_{A_{\epsilon,\delta}} | \sigma(\{n_{m,i}(t)\}_{m,i,t})]]$$

$$= \frac{1}{P(A_{\epsilon,\delta})} E[\frac{1}{\exp\{\lambda(n)\sqrt{\frac{C_1 \log t}{n_{m,i}(t)}}\}} E[\exp\{\lambda(n)(\tilde{\mu}_{m,i}(t) - \mu_i)\} 1_{A_{\epsilon,\delta}} | \sigma(\{n_{m,i}(t)\}_{m,i,t})]]$$

$$\leq \frac{1}{P(A_{\epsilon,\delta})} E[\frac{1}{\exp\{\lambda(n)\sqrt{\frac{C_1 \log t}{n_{m,i}(t)}}\}} \cdot \exp\{\frac{\lambda^2(n)}{1} \frac{C\sigma^2}{n_{m,i}(t)}\}]$$

$$\leq \frac{1}{P(A_{\epsilon,\delta})} \exp\{-2\log t\} = \frac{1}{P(A_{\epsilon,\delta}) t^2} \qquad (28)$$

where the first inequality holds by (44) and the second inequality holds by choosing $\lambda(n) = \frac{\sqrt{\frac{C_1 \log t}{n_{m,i}(t)}}}{2\frac{C\sigma^2}{n_{m,i}(t)}}$

and by the choice of parameter $C_1$ such that $\frac{C_1}{4C\sigma^2} \geq 2$ or equivalently $C_1 \geq 8C\sigma^2$.

In like manner, we obtain that by repeating the above steps with $\mu_i - \tilde{\mu}_{m,i}(t)$, we have

$$P(\mu_i - \tilde{\mu}_{m,i}(t) \geq \sqrt{\frac{C_1 \log t}{n_{m,i}(t)}}|A_{\epsilon,\delta}) \leq \frac{1}{P(A_{\epsilon,\delta})t^2} \tag{29}$$

which complete the proof.

$\square$

**Proposition 10.** *Assume the parameter $\delta$ satisfies that $0 < \delta < c = f(\epsilon, M, T)$. An arm $k$ is said to be sub-optimal if $k \neq i^*$ where $i^*$ is the unique optimal arm in terms of the global reward, i.e. $i^* = \arg\max \frac{1}{M} \sum_{j=1}^{M} \mu_i^j$. Then in setting $s_1, s_2$ and $s_3$, when the game ends, for every client $m$, $0 < \epsilon < 1$ and $T > L$, the expected numbers of pulling sub-optimal arm $k$ after the burn-in period satisfies with $P(A_{\epsilon,\delta}) = 1 - 7\epsilon$*

$$E[n_{m,k}(T)|A_{\epsilon,\delta}]$$
$$\leq \max\{[\frac{4C_1 \log T}{\Delta_i^2}], 2(K^2 + MK + M)\} + \frac{2\pi^2}{3P(A_{\epsilon,\delta})} + K^2 + (2M - 1)K$$
$$\leq O(\log T).$$

*Proof of Proposition 10.* We claim that what lead to pulling an sub-optimal arm $i$ are explicit by the decision rule of Algorithm 2, meaning that the result $a_t^m = i$ holds when any of the following conditions is met:

- Case 1: $n_{m,i}(t) \leq \mathcal{N}_{m,i}(t) - K$,

- Case 2: $\tilde{\mu}_{m,i} - \mu_i > \sqrt{\frac{C_1 \log t}{n_{m,i}(t-1)}}$,

- Case 3: $-\tilde{\mu}_{m,i^*} + \mu_{i^*} > \sqrt{\frac{C_1 \log t}{n_{m,i^*}(t-1)}}$,

- Case 4: $\mu_{i^*} - \mu_i < 2\sqrt{\frac{C_1 \log t}{n_{m,i}(t-1)}}$.

Then we formally consider the number of pulling arms $n_{m,i}(T)$ starting from $L+1$. For any $l > 1$, we have that based on the above listed conditions

$$n_{m,i}(T) \leq l + \sum_{t=L+1}^{T} 1_{\{a_t^m = i, n_{m,i}(t) > l\}}$$

$$\leq l + \sum_{t=L+1}^{T} 1_{\{\tilde{\mu}_i^m - \sqrt{\frac{C_1 \log t}{n_{m,i}(t-1)}} > \mu_i, n_{m,i}(t-1) \geq l\}}$$

$$+ \sum_{t=L+1}^{T} 1_{\{\tilde{\mu}_{i^*}^m + \sqrt{\frac{C_1 \log t}{n_{m,i^*}(t-1)}} < \mu_{i^*}, n_{m,i}(t-1) \geq l\}}$$

$$+ \sum_{t=L+1}^{T} 1_{\{n_{m,i}(t) < \mathcal{N}_{m,i}(t) - K, a_t^m = i, n_{m,i}(t-1) \geq l\}}$$

$$+ \sum_{t=L+1}^{T} 1_{\{\mu_i + 2\sqrt{\frac{C_1 \log t}{n_{m,i}(t-1)}} > \mu_{i^*}, n_{m,i}(t-1) \geq l\}}.$$

Consequently, the expected value of $n_{m,i}(t)$ conditional on $A_{\epsilon,\delta}$ reads as

$$
E[n_{m,i}(T)|A_{\epsilon,\delta}]
$$

$$
= l + \sum_{t=L+1}^{T} P(\tilde{\mu}_i^m - \sqrt{\frac{C_1 \log t}{n_{m,i}(t-1)}} > \mu_i, n_{m,i}(t-1) \geq l | A_{\epsilon,\delta})
$$

$$
+ \sum_{t=L+1}^{T} P(\tilde{\mu}_{i^*}^m + \sqrt{\frac{C_1 \log t}{n_{m,i^*}(t-1)}} < \mu_{i^*}, n_{m,i}(t-1) \geq l | A_{\epsilon,\delta})
$$

$$
+ \sum_{t=L+1}^{T} P(n_{m,i}(t) < \mathcal{N}_{m,i}(t) - K, a_t^m = i, n_{m,i}(t-1) \geq l | A_{\epsilon,\delta})
$$

$$
+ \sum_{t=L+1}^{T} P(\mu_i + 2\sqrt{\frac{C_1 \log t}{n_{m,i}(t-1)}} > \mu_{i^*}, n_{m,i}(t-1) \geq l | A_{\epsilon,\delta})
$$

$$
= l + \sum_{t=L+1}^{T} P(Case2, n_{m,i}(t-1) \geq l | A_{\epsilon,\delta}) + \sum_{t=L+1}^{T} P(Case3, n_{m,i}(t-1) \geq l | A_{\epsilon,\delta})
$$

$$
+ \sum_{t=L+1}^{T} P(Case1, a_t^m = i, n_{m,i}(t-1) \geq l | A_{\epsilon,\delta}) + \sum_{t=L+1}^{T} P(Case4, n_{m,i}(t-1) \geq l | A_{\epsilon,\delta})
$$

$$
\tag{30}
$$

where $l = \max \{[\frac{4C_1 \log T}{\Delta_i^2}], 2(K^2 + MK + M)\}$.

For the last term in (30), we have

$$
\sum_{t=L+1}^{T} P(Case4 : \mu_i + 2\sqrt{\frac{C_1 \log t}{n_{m,i}(t-1)}} > \mu_{i^*}, n_{m,i}(t-1) \geq l) = 0 \tag{31}
$$

since the choice of $l$ satisfies $l \geq [\frac{4C_1 \log T}{\Delta_i^2}]$ with $\Delta_i = \mu_{i^*} - \mu_i$.

For the first two terms, we have on event $A_{\epsilon,\delta}$

$$
\sum_{t=L+1}^{T} P(Case2, n_{m,i}(t-1) \geq l | A_{\epsilon,\delta}) + \sum_{t=1}^{T} P(Case3, n_{m,i}(t-1) \geq l | A_{\epsilon,\delta})
$$

$$
\leq \sum_{t=L+1}^{T} P(\tilde{\mu}_{m,i} - \mu_i > \sqrt{\frac{C_1 \log t}{n_{m,i}(t-1)}} | A_{\epsilon,\delta}) + \sum_{t=1}^{T} P(-\tilde{\mu}_{m,i^*} + \mu_{i^*} > \sqrt{\frac{C_1 \log t}{n_{m,i^*}(t-1)}} | A_{\epsilon,\delta})
$$

$$
\leq \sum_{t=1}^{T} (\frac{1}{t^2}) + \sum_{t=1}^{T} (\frac{1}{t^2}) \leq \frac{\pi^2}{3} \tag{32}
$$

where the first inequality holds by the property of the probability measure when removing the event $n_{m,i}(t-1) \geq l$ and the second inequality holds by (47) and (29) as stated in Proposition 9, which holds by the assumption that $\delta < c$.

For Case 1, we note that Lemma 1 implies that

$$
n_{m,i}(t) > \mathcal{N}_{m,i}(t) - K(K + 2M)
$$

with the definition of $N_{m,i}(t+1) = \max\{n_{m,i}(t+1), N_{j,i}(t), j \in \mathcal{N}_m(t)\}$.

Departing from the result that the difference between $N_{m,i}(t)$ and $n_{m,i}(t)$ is at most $K(K + 2M)$, we then present the following analysis on how long it takes for the value $-n_{m,i}(t) + N_{m,i}(t)$ to be smaller than $K$.

At time step $t$, if Case 1 holds for client $m$, then $n_{m,i}(t+1)$ is increasing by 1 on the basis of $n_{m,i}(t)$. What follows characterizes the change of $N_{m,i}(t+1)$. Client $m$ satisfying $n_{m,i}(t) \leq \mathcal{N}_{m,i}(t) - K$ will not change the value of $N_{m,i}(t+1)$ by the definition $N_{m,i}(t+1) = \max\{n_{m,i}(t+1), N_{j,i}(t), j \in$

$\mathcal{N}_m(t)\}$. Moreover, for client $j \in \mathcal{N}_m(t)$ with $n_{j,i}(t) < \mathcal{N}_{j,i}(t) - K$, i.e. $\mathcal{N}_{j,i}(t + 1)$ will not be affected by $n_{j,i}(t + 1) \leq n_{j,i}(t) + 1$. Thus, the value of $N_{m,i}(t + 1) = \max\{n_{m,i}(t + 1), N_{j,i}(t), j \in \mathcal{N}_m(t)\}$ is independent of such clients. We observe that for client $j \in \mathcal{N}_m(t)$ with $n_{j,i}(t) > \mathcal{N}_{j,i}(t) - K$, the value $N_{j,i}(t)$ will be the same if the client does not sample arm $i$, which leads to a decrease of 1 in the difference $-n_{m,i}(t) + N_{m,i}(t)$. Otherwise, if such a client samples arm $i$ which brings an increment of 1 to $N_{m,i}(t)$, the difference between $n_{m,i}(t)$ and $N_{m,i}(t)$ will remain the same. However, the latter has just been discussed and must be the cases as in Case 2 and Case 3, the total length of which has already been upper bounded by $\frac{\pi^2}{3}$ as shown in (32).

Therefore, the gap is at most $K(K + 2M) - K + \frac{\pi^2}{3}$, i.e.

$$\sum_{t=1}^{T} P(Case1, a_t^m = i, n_{m,i}(t - 1) \geq l|A) \leq K(K + 2M) - K + \frac{\pi^2}{3}. \tag{33}$$

Subsequently, we derive that

$$
\begin{aligned}
E[n_{m,i}(T)|A_{\epsilon,\delta}] &\leq l + \frac{\pi^2}{3} + K(K + 2M) - K + \frac{\pi^2}{3} + 0 \\
&= l + \frac{2\pi^2}{3} + K^2 + (2M - 1)K \\
&= \max\{[\frac{4C_1 \log T}{\Delta_i^2}], 2(K^2 + MK + M)\} + \frac{2\pi^2}{3} + K^2 + (2M - 1)K
\end{aligned}
$$

where the inequality results from (30), (31), (32), and (33).

This completes the proof steps.

$\square$

Next, we establish the concentration inequalities of the network-wide estimators when the rewards follow sub-exponential distributions, i.e. in setting $S_1, S_2$, and $S_3$.

**Proposition 11.** *Assume the parameter $\delta$ satisfies that $0 < \delta < c = f(\epsilon, M, T)$. In setting $S_1, S_2$, and $S_3$, for any $m, i, \lambda$ and $t > L$ where $L$ is the length of the burn-in period, the global estimator $\tilde{\mu}_i^m(t)$ is sub-exponentially distributed. Moreover, the conditional moment generating function satisfies that with $P(A_{\epsilon,\delta}) = 1 - 7\epsilon$, for $|\lambda| < \frac{1}{\alpha}$*

$$E[\exp\{\lambda(\tilde{\mu}_i^m(t) - \mu_i)\}1_{A_{\epsilon,\delta}}|\sigma(\{n_{m,i}(t)\}_{t,i,m})]$$
$$\leq \exp\{\frac{\lambda^2}{2} \frac{C\sigma^2}{\min_j n_{j,i}(t)}\}$$

*where $\sigma^2 = \max_{j,i}(\tilde{\sigma}_i^j)^2$ and $C = \max\{\frac{4(M+2)(1-\frac{1-c_0}{2(M+2)})^2}{3M(1-c_0)}, (M + 2)(1 + 4Md_{m,t}^2)\}$.*

*Proof.* Assume that parameter $|\lambda| < \frac{1}{\alpha}$. We prove the statement on the conditional moment generating function by induction. Let us start with the basis step.

Note that the definition of $A$ and the choice of $\delta$ again guarantee that for $t \geq L$, $|P_t - cE| < \delta < c$ on event $A$. This implies that for any $t \geq L$, $P_t > 0$ and thereby obtaining that if $t = L$

$$P'_{m,j}(t) = \frac{1}{M} \tag{34}$$

and if $t > L$

$$P'_{m,j}(t) = \frac{M - 1}{M^2}. \tag{35}$$

Consider the time step $t \le L + 1$. The quantity

$$E[\exp\{\lambda(\tilde{\mu}_i^m(t) - \mu_i)\}1_{A_{\epsilon,\delta}}|\sigma(\{n_{m,i}(t)\}_{t,i,m})]$$

$$= E[\exp\{\lambda(\tilde{\mu}_i^m(L+1) - \mu_i)\}1_{A_{\epsilon,\delta}}|\sigma(\{n_{m,i}(t)\}_{t,i,m})]$$

$$= E[\exp\{\lambda(\sum_{j=1}^{M} P'_{m,j}(L)\hat{\bar{\mu}}_{i,j}^m(h_{m,j}^L) - \mu_i)\}1_{A_{\epsilon,\delta}}|\sigma(\{n_{m,i}(t)\}_{t,i,m})]$$

$$= E[\exp\{\lambda(\sum_{j=1}^{M} \frac{1}{M}\bar{\mu}_i^j(h_{m,j}^L) - \mu_i)\}1_{A_{\epsilon,\delta}}|\sigma(\{n_{m,i}(t)\}_{t,i,m})]$$

$$= E[\exp\{\lambda \sum_{j=1}^{M} \frac{1}{M}(\bar{\mu}_i^j(h_{m,j}^L) - \mu_i^j)\}1_{A_{\epsilon,\delta}}|\sigma(\{n_{m,i}(t)\}_{t,i,m})]$$

$$\le \Pi_{j=1}^{M}(E[(\exp\{(\lambda\frac{1}{M}(\bar{\mu}_i^j(h_{m,j}^L) - \mu_i^j)\}1_{A_{\epsilon,\delta}})^M|\sigma(\{n_{m,i}(t)\}_{t,i,m})])^{\frac{1}{M}} \qquad (36)$$

where the third equality holds by (34), the fourth equality uses the definition $\mu_i = \frac{1}{M}\sum_{i=1}^{M}\mu_i^j$ and the last inequality results from the generalized hoeffding inequality as in Lemma 3.

Note that for any client $j$, by the definition of $\bar{\mu}_i^j(h_{m,j}^L)$ we have

$$E[(\exp\{(\lambda\frac{1}{M}(\bar{\mu}_i^j(h_{m,j}^L) - \mu_i^j)\}1_{A_{\epsilon,\delta}})^M|\sigma(\{n_{m,i}(t)\}_{t,i,m}))]$$

$$= E[\exp\{(\lambda(\bar{\mu}_i^j(h_{m,j}^L) - \mu_i^j)\}1_{A_{\epsilon,\delta}})|\sigma(\{n_{m,i}(t)\}_{t,i,m})]$$

$$= E[\exp\{(\lambda\frac{\sum_s(r_i^j(s) - \mu_i^j)}{n_{j,i}(h_{m,j}^L)}\}1_{A_{\epsilon,\delta}})|\sigma(\{n_{m,i}(t)\}_{t,i,m})]$$

$$= E[\exp\{\sum_s(\lambda\frac{(r_i^j(s) - \mu_i^j)}{n_{j,i}(h_{m,j}^L)}\}1_{A_{\epsilon,\delta}})|\sigma(\{n_{m,i}(t)\}_{t,i,m})]. \qquad (37)$$

It is worth noting that given $s$, $r_i^j(s)$ is independent of everything else, which gives us

$$(37) = \Pi_s E[\exp\{\lambda\frac{(r_i^j(s) - \mu_i^j)}{n_{j,i}(h_{m,j}^L)}\}1_{A_{\epsilon,\delta}}|\sigma(\{n_{m,i}(t)\}_{t,i,m})]$$

$$= \Pi_s E[\exp\{\lambda\frac{(r_i^j(s) - \mu_i^j)}{n_{j,i}(h_{m,j}^L)}\}|\sigma(\{n_{m,i}(t)\}_{t,i,m})] \cdot E[1_{A_{\epsilon,\delta}}|\sigma(\{n_{m,i}(t)\}_{t,i,m})]$$

$$= \Pi_s E_r[\exp\{\lambda\frac{(r_i^j(s) - \mu_i^j)}{n_{j,i}(h_{m,j}^L)}\}] \cdot E[1_{A_{\epsilon,\delta}}|\sigma(\{n_{m,i}(t)\}_{t,i,m})]$$

$$\le \Pi_s \exp\{\frac{(\frac{\lambda}{n_{j,i}(h_{m,j}^L)})^2\sigma^2}{2}\} \cdot E[1_{A_{\epsilon,\delta}}|\sigma(\{n_{m,i}(t)\}_{t,i,m})]$$

$$\le (\exp\{\frac{(\frac{\lambda}{n_{j,i}(h_{m,j}^L)})^2\sigma^2}{2}\})^{n_{j,i}(h_{m,j}^L)}$$

$$= \exp\{\frac{\frac{\lambda^2}{n_{j,i}(h_{m,j}^L)}\sigma^2}{2}\} \le \exp\{\frac{\lambda^2\sigma^2}{2\min_j n_{j,i}(h_{m,j}^L)}\} \qquad (38)$$

where the first inequality holds by the definition of sub-exponential random variables $r_i^j(s) - \mu_i^j$ with mean 0, the second inequality again uses $1_{A_{\epsilon,\delta}} \le 1$, and the last inequality is by the fact that $n_{j,i}(h_{m,j}^L) \ge \min_j n_{j,i}(h_{m,j}^L)$.

Therefore, we obtain that by plugging (38) into (36)

$$(36) \leq \Pi_{j=1}^{M} (\exp\{\frac{\lambda^2 \sigma^2}{2 \min_j n_{j,i}(h_{m,j}^L)}\})^{\frac{1}{M}}$$

$$= ((\exp\{\frac{\lambda^2 \sigma^2}{2 \min_j n_{j,i}(h_{m,j}^L)}\})^{\frac{1}{M}})^M = \exp\{\frac{\lambda^2 \sigma^2}{2 \min_j n_{j,i}(h_{m,j}^L)}\}$$

which completes the basis step.

Now we proceed to the induction step. Suppose that for any $s < t+1$ where $t+1 > L+1$, we have

$$E[\exp\{\lambda(\tilde{\mu}_i^m(s) - \mu_i)\} 1_{A_{\epsilon,\delta}} | \sigma(\{n_{m,i}(s)\}_{s,i,m})]$$
$$\leq \exp\{\frac{\lambda^2}{2} \frac{C\sigma^2}{\min_j n_{j,i}(s)}\} \tag{39}$$

The update rule of $\tilde{\mu}_i^m$ again and (25) implies that

$$E[\exp\{\lambda(\tilde{\mu}_i^m(t+1) - \mu_i)\} 1_{A_{\epsilon,\delta}} | \sigma(\{n_{m,i}(s)\}_{s,i,m})]$$
$$\leq \Pi_{j=1}^{M} (\exp\{\frac{\lambda^2 (P_t'(m,j))^2 (M+2)^2}{2} \frac{C\sigma^2}{\min_j n_{j,i}(t_{m,j})}\})^{\frac{1}{M+2}} \cdot$$

$$\Pi_{j \in N_m(t)} \Pi_s (E_r[\exp\{\lambda d_{m,t}(M+2)\frac{(r_i^j(s) - \mu_i^j)}{n_{j,i}(t)}\}] \cdot E[1_{A_{\epsilon,\delta}} | \sigma(\{n_{m,i}(t)\}_{t,i,m})])^{\frac{1}{M+2}} \cdot$$

$$\Pi_{j \notin N_m(t)} \Pi_s (E_r[\exp\{\lambda d_{m,t}(M+2)\frac{(r_i^j(s) - \mu_i^j)}{n_{j,i}(t_{m,j})}\}] \cdot E[1_{A_{\epsilon,\delta}} | \sigma(\{n_{m,i}(t)\}_{t,i,m})])^{\frac{1}{M+2}} \cdot \tag{40}$$

We continue bounding the last two terms by using the definition of sub-exponential random variables $(r_i^j(s) - \mu_i^j)$ and obtain

$$(40) \leq (\exp\{\frac{\lambda^2 (P_t'(m,j))^2 (M+2)^2}{2} \frac{C\sigma^2}{\min_j n_{j,i}(t_{m,j})}\})^{\frac{M}{M+2}} \cdot$$

$$\Pi_{j \in N_m(t)} \Pi_s (\exp \frac{\lambda^2 d_{m,t}^2 (M+2)^2 \sigma^2}{2 n_{j,i}^2(t)} \cdot E[1_{A_{\epsilon,\delta}} | \sigma(\{n_{m,i}(t)\}_{t,i,m})])^{\frac{1}{M+2}} \cdot$$

$$\Pi_{j \notin N_m(t)} \Pi_s (\exp \frac{\lambda^2 d_{m,t}^2 (M+2)^2 \sigma^2}{2 n_{j,i}^2(t_{m,j})} \cdot E[1_{A_{\epsilon,\delta}} | \sigma(\{n_{m,i}(t)\}_{t,i,m})])^{\frac{1}{M+2}}$$

$$= (\exp\{\frac{\lambda^2 (P_t'(m,j))^2 (M+2)^2}{2} \frac{C\sigma^2}{\min_j n_{j,i}(t_{m,j})}\})^{\frac{M}{M+2}} \cdot$$

$$\Pi_{j \in N_m(t)} \exp\{\frac{n_{j,i}(t)}{M+2} \frac{\lambda^2 d_{m,t}^2 (M+2)^2 \sigma^2}{2 n_{j,i}^2(t)}\} \cdot E[1_{A_{\epsilon,\delta}} | \sigma(\{n_{m,i}(t)\}_{t,i,m})] \cdot$$

$$\Pi_{j \notin N_m(t)} \exp\{\frac{n_{j,i}(t_{m,j})}{M+2} \frac{\lambda^2 d_{m,t}^2 (M+2)^2 \sigma^2}{2 n_{j,i}^2(t_{m,j})}\} \cdot E[1_{A_{\epsilon,\delta}} | \sigma(\{n_{m,i}(t)\}_{t,i,m})].$$

Building on that, we establish

$$(40) \leq (\exp\{\frac{\lambda^2 (P_t'(m,j))^2 (M+2)^2}{2} \frac{C\sigma^2}{\min_j n_{j,i}(t_{m,j})}\})^{\frac{M}{M+2}} \cdot$$

$$(\exp\{\frac{\lambda^2 d_{m,t}^2 (M+2)\sigma^2}{2\min_j n_{j,i}(t)}\})^{|N_m(t)|} \cdot E[1_{A_{\epsilon,\delta}} | \sigma(\{n_{m,i}(t)\}_{t,i,m})] \cdot$$

$$(\exp\{\frac{\lambda^2 d_{m,t}^2 (M+2)\sigma^2}{2\min_j n_{j,i}(t_{m,j})}\})^{|M-N_m(t)|} \cdot E[1_{A_{\epsilon,\delta}} | \sigma(\{n_{m,i}(t)\}_{t,i,m})]$$

$$= E[(\exp\{\frac{\lambda^2 (P_t'(m,j))^2 M(M+2)}{2} \frac{C\sigma^2}{\min_j n_{j,i}(t_{m,j})}\}) \cdot (\exp\{\frac{\lambda^2 d_{m,t}^2 (M+2)|N_m(t)|}{2\min_j n_{j,i}(t)}\})$$

$$\cdot (\exp\{\frac{\lambda^2 d_{m,t}^2 (M+2)\sigma^2 |M-N_m(t)|}{2\min_j n_{j,i}(t_{m,j})}\}) 1_{A_{\epsilon,\delta}} | \sigma(\{n_{m,i}(t)\}_{t,i,m})]$$

$$\leq E[(\exp\{\frac{\lambda^2 (P_t'(m,j))^2 M(M+2)}{2(1-c_0)} \frac{C\sigma^2}{\min_j n_{j,i}(t+1)}\}) \cdot (\exp\{\frac{\lambda^2 d_{m,t}^2 (M+2)|N_m(t)|\sigma^2}{2\frac{L/K}{L/K+1}\min_j n_{j,i}(t+1)}\})$$

$$\cdot (\exp\{\frac{\lambda^2 d_{m,t}^2 (M+2)|M-N_m(t)|\sigma^2}{2(1-c_0)\min_j n_{j,i}(t+1)}\}) 1_{A_{\epsilon,\delta}} | \sigma(\{n_{m,i}(t)\}_{t,i,m})]$$

where the first inequality uses the fact that $n_{j,i}(t) \geq \min_j n_{j,i}(t)$ and $n_{j,i}(t_{m,j}) \geq \min_j n_{j,i}(t_{m,j})$. For the second inequality, the first term is a result of $\frac{\min_j n_{j,i}(t)}{\min_j n_{j,i}(t+1)} \geq \frac{\min_j n_{j,i}(t)}{\min_j n_{j,i}(t)+1} \geq \frac{L/K}{L/K+1}$ since $n_{j,i}(t) > n_{j,i}(L) = L/K$ and the ratio is monotone increasing in $n$, and the second term is bounded through applying Proposition 6, which holds on event $A_{\epsilon,\delta}$ and leads to

$$\min_j n_{j,i}(t_{m,j}) \geq \min_j n_{j,i}(t+1-t_0)$$
$$\geq \min_j n_{j,i}(t+1) - t_0$$
$$\geq \min_j n_{j,i}(t+1) - c_0 \min_j n_{j,i}(t+1)$$
$$= (1-c_0) \min_j n_{j,i}(t+1).$$

Therefore, we can rewrite the above expression as

$$(40) = E[(\exp\{\frac{\lambda^2 \sigma^2}{2\min_j n_{j,i}(t+1)} \cdot (\frac{C\lambda^2 (P_t'(m,j))^2 M(M+2)}{2(1-c_0)} +$$

$$\frac{d_{m,t}^2 (M+2)|N_m(t)|}{\frac{L/K}{L/K+1}} + \frac{d_{m,t}^2 (M+2)|M-N_m(t)|}{(1-c_0)})\} 1_{A_{\epsilon,\delta}} | \sigma(\{n_{m,i}(t)\}_{t,i,m})]$$

$$\leq E[\exp\{\frac{C\lambda^2 \sigma^2}{2\min_j n_{j,i}(t+1)}\} 1_{A_{\epsilon,\delta}} | \sigma(\{n_{m,i}(t)\}_{t,i,m})]$$

$$\leq \exp\{\frac{C\lambda^2 \sigma^2}{2\min_j n_{j,i}(t+1)}\}$$

where the first inequality holds again by the choice of $P_t'(m,j), d_{m,t}, L$ and $c_0$ and the second inequality uses the fact that $1_{A_{\epsilon,\delta}} \leq 1$.

This completes the induction step and subsequently concludes the proof.

$\square$

**Proposition 12.** *Assume the parameter $\delta$ satisfies that $0 < \delta < c = f(\epsilon, M, T)$. In setting $S_1, S_2$, and $S_3$, for any $m, i$ and $t > L$ where $L$ is the length of the burn-in period, the deviation of $\tilde{\mu}_{m,i}(t)$*

*satisfies that if $n_{m,i}(t) \geq 2(K^2 + KM + M)$, then with $P(A_{\epsilon,\delta}) = 1 - 7\epsilon$,*

$$P(\tilde{\mu}_{m,i}(t) - \mu_i \geq \sqrt{\frac{C_1 \log t}{n_{m,i}(t)}} + \frac{C_2 \log t}{n_{m,i}(t)} | A_{\epsilon,\delta}) \leq \frac{1}{P(A_{\epsilon,\delta})} \frac{1}{T^4},$$

$$P(\mu_i - \tilde{\mu}_{m,i}(t) \geq \sqrt{\frac{C_1 \log t}{n_{m,i}(t)}} + \frac{C_2 \log t}{n_{m,i}(t)} | A_{\epsilon,\delta}) \leq \frac{1}{P(A_{\epsilon,\delta})} \frac{1}{T^4}.$$

*Proof.* By Proposition 7, we have $E[\tilde{\mu}_{m,i}(t) - \mu_i | A_{\epsilon,\delta}] = 0$, which allows us to consider the tail bound of the global estimator $\tilde{\mu}_i^m(t)$ conditional on event $A_{\epsilon,\delta}$. It is worth mentioning that by the choice of $C_1$ and $C_2$, we have

$$C_1^2 \cdot \frac{\alpha^2}{\tilde{\sigma}^4} \leq C_2^2.$$

where $\tilde{\sigma}^2$ is $\frac{2C\sigma^2}{n_{m,i}(t)}$.

Note that since we set $Rad = \sqrt{\frac{C_1 \ln T}{n_{m,i}(t)}} + \frac{C_2 \ln T}{n_{m,i}(t)}$, we obtain

$$P(|\tilde{\mu}_i^m(t) - \mu_i| > Rad | A_{\epsilon,\delta}) < P(|\tilde{\mu}_i^m(t) - \mu_i| > \sqrt{\frac{C_1 \ln T}{n_{m,i}(t)}} | A_{\epsilon,\delta}), \tag{41}$$

$$P(|\tilde{\mu}_i^m(t) - \mu_i| > Rad | A_{\epsilon,\delta}) < P(|\tilde{\mu}_i^m(t) - \mu_i| > \frac{C_2 \ln T}{n_{m,i}(t)} | A_{\epsilon,\delta}) \tag{42}$$

On the one hand, if $\frac{\sqrt{\frac{C_1 \log T}{n_{m,i}(t)}}}{\tilde{\sigma}^2} > \frac{1}{\alpha}$, i.e. $n_{m,i}(t) \leq C_1 \log T \frac{\alpha^2}{(\tilde{\sigma})^4}$, we have

$$P(|\tilde{\mu}_i^m(t) - \mu_i| > \frac{C_2 \ln T}{n_{m,i}(t)} | A_{\epsilon,\delta})$$

$$= E[1_{|\tilde{\mu}_i^m(t) - \mu_i| > \frac{C_2 \ln T}{n_{m,i}(t)}} | A_{\epsilon,\delta}]$$

$$= \frac{1}{P(A_{\epsilon,\delta})} E[1_{|\tilde{\mu}_i^m(t) - \mu_i| > \frac{C_2 \ln T}{n_{m,i}(t)}} 1_{A_{\epsilon,\delta}}]$$

$$= \frac{1}{P(A_{\epsilon,\delta})} E[1_{\exp\{\lambda(n)(|\tilde{\mu}_{m,i}(t) - \mu_i|)\} \geq \exp\{\lambda(n) \frac{C_2 \ln T}{n_{m,i}(t)}\}} 1_{A_{\epsilon,\delta}}]$$

$$\leq \frac{1}{P(A_{\epsilon,\delta})} E[\frac{\exp\{\lambda(n)(|\tilde{\mu}_{m,i}(t) - \mu_i|)\}}{\exp\{\lambda(n) \frac{C_2 \ln T}{n_{m,i}(t)}\}} 1_{A_{\epsilon,\delta}}] \tag{43}$$

where the last inequality is by the fact that $1_{\exp\{\lambda(n)(|\tilde{\mu}_{m,i}(t) - \mu_i|)\} \geq \exp\{\lambda(n) \frac{C_2 \ln T}{n_{m,i}(t)}\}} \leq \frac{\exp\{\lambda(n)(|\tilde{\mu}_{m,i}(t) - \mu_i|)\}}{\exp\{\lambda(n) \frac{C_2 \ln T}{n_{m,i}(t)}\}}$.

By the assumption that $\delta < c$, we have Proposition 11 holds. Subsequently, by Proposition 11 and Lemma 2 which holds since $n_{m,i}(t) \geq 2(K^2 + KM + M)$, we have for any $|\lambda| < \frac{1}{\alpha}$

$$E[\exp\{\lambda(\tilde{\mu}_i^m(t) - \mu_i)\} 1_{A_{\epsilon,\delta}} | \sigma(\{n_{m,i}(t)\}_{t,i,m})] \leq \exp\{\frac{\lambda^2}{2} \tilde{\sigma}^2\}. \tag{44}$$

Likewise, we obtain that by taking $\lambda = -\lambda$,

$$E[\exp\{\lambda(-\tilde{\mu}_i^m(t) + \mu_i)\} 1_{A_{\epsilon,\delta}} | \sigma(\{n_{m,i}(t)\}_{t,i,m})] \leq \exp\{\frac{\lambda^2}{2} \tilde{\sigma}^2\}. \tag{45}$$

With (44) and (45), we arrive at for any $|\lambda| < \frac{1}{\alpha}$ that

$$E[\exp\{\lambda(|\tilde{\mu}_i^m(t) - \mu_i|)\} 1_{A_{\epsilon,\delta}} | \sigma(\{n_{m,i}(t)\}_{t,i,m})] \leq 2\exp\{\frac{\lambda^2}{2} \tilde{\sigma}^2\}. \tag{46}$$

Again, we utilize the law of total expectation and further obtain that $|\lambda(n)| < \frac{1}{\alpha}$

$$(43) = \frac{1}{P(A_{\epsilon,\delta})} E[E[\frac{\exp\{\lambda(n)(|\tilde{\mu}_{m,i}(t) - \mu_i|)\}}{\exp\{\lambda(n)\frac{C_2 \ln T}{n_{m,i}(t)}\}} 1_{A_{\epsilon,\delta}} | \sigma(\{n_{m,i}(t)\}_{m,i,t})]]$$

$$= \frac{1}{P(A_{\epsilon,\delta})} E[\frac{1}{\exp\{\lambda(n)\frac{C_2 \ln T}{n_{m,i}(t)}\}} E[\exp\{\lambda(n)(|\tilde{\mu}_{m,i}(t) - \mu_i|)\} 1_{A_{\epsilon,\delta}} | \sigma(\{n_{m,i}(t)\}_{m,i,t})]]$$

$$\leq 2\frac{1}{P(A_{\epsilon,\delta})} E[\frac{1}{\exp\{\lambda(n)\frac{C_2 \ln T}{n_{m,i}(t)}\}} \cdot \exp\{\frac{\lambda^2(n)}{2}\tilde{\sigma}^2\}] \tag{47}$$

where the first inequality holds by (46).

Note that the condition $\frac{\sqrt{\frac{C_1 \log T}{n_{m,i}(t)}}}{\tilde{\sigma}^2} > \frac{1}{\alpha}$ implies that $n_{m,i}(t) < \frac{C_1 \ln T}{\frac{\tilde{\sigma}^2}{\alpha}}$ which is the global optima of the function in (47). This is true since $n_{m,i}(t) \leq C_1 \log T \frac{\alpha^2}{(\tilde{\sigma})^4} \leq \frac{(C_2 \log T)^2}{C_1}$. Henceforth, (47) is monotone decreasing in $\lambda(n) \in (0, \frac{1}{\alpha})$ and we obtain a minima when choosing $\lambda(n) = \frac{1}{\alpha}$ and using the continuity of (47).

Formally, it yields that

$$(47) \leq 2\frac{1}{P(A_{\epsilon,\delta})} E[\frac{1}{\exp\{\frac{1}{\alpha}\frac{C_2 \ln T}{n_{m,i}(t)}\}} \cdot \exp\{\frac{\frac{1}{2\alpha^2}}{1}\tilde{\sigma}^2\}]$$

$$= 2\frac{1}{P(A_{\epsilon,\delta})} E[\exp\{\frac{1}{2\alpha^2}\tilde{\sigma}^2 - \frac{1}{\alpha}\frac{C_2 \ln T}{n_{m,i}(t)}\}]$$

$$\leq 2\frac{1}{P(A_{\epsilon,\delta})} \exp\{-4\log T\} = \frac{2}{P(A_{\epsilon,\delta})T^4} \tag{48}$$

where the last inequality uses the choice of $C_2$ and the condition that $\frac{1}{2\alpha^2}\tilde{\sigma}^2 - \frac{1}{\alpha}\frac{C_2 \ln T}{n_{m,i}(t)} \leq -4 \ln T$ which holds by the following derivation. Notably, we have

$$\frac{1}{2\alpha^2}\tilde{\sigma}^2 - \frac{1}{\alpha}\frac{C_2 \ln T}{n_{m,i}(t)} \leq \frac{1}{2\alpha^2}\tilde{\sigma}^2 - \frac{C_2 \ln T}{\alpha}\frac{\frac{\tilde{\sigma}^2}{\alpha}}{C_1 \ln T}$$

$$= \frac{1}{2\alpha^2}\tilde{\sigma}^2 - \frac{C_2}{C_1}\frac{\tilde{\sigma}^2}{\alpha^2}$$

$$= (\frac{1}{2} - \frac{C_2}{C_1})(\tilde{\sigma}^2 \cdot \frac{\frac{C_1 \log T}{n_{m,i}(t)}}{\tilde{\sigma}^4}) = (\frac{1}{2} - \frac{C_2}{C_1})(\frac{\frac{C_1 \log T}{n_{m,i}(t)}}{\tilde{\sigma}^2})$$

$$= (\frac{1}{2} - \frac{C_2}{C_1}) \cdot \frac{C_1 \log T}{n_{m,i}(t)} \cdot \frac{1}{\frac{2C\sigma^2}{n_{m,i}(t)}} = (\frac{1}{2} - \frac{C_2}{C_1})\frac{C_1}{2C\sigma^2}\log T \leq -4\log T$$

where the first inequality uses $n_{m,i}(t) < \frac{C_1 \ln T}{\frac{\tilde{\sigma}^2}{\alpha}}$ and the last inequality is by the choices of parameters $(\frac{1}{2} - \frac{C_2}{C_1})\frac{C_1}{2C\sigma^2} \leq -4$.

On the other hand, if $\frac{\sqrt{\frac{C_1 \log T}{n_{m,i}(t)}}}{\tilde{\sigma}^2} < \frac{1}{\alpha}$, i.e. $n_{m,i}(t) \geq C_1 \log T \frac{\alpha^2}{(\tilde{\sigma})^4}$, we observe for $|\lambda(n)| < \frac{1}{\alpha}$

$$P(|\tilde{\mu}_i^m(t) - \mu_i| > \sqrt{\frac{C_1 \ln T}{n_{m,i}(t)}}|A_{\epsilon,\delta})$$

$$\leq 2\frac{1}{P(A_{\epsilon,\delta})} E[\frac{1}{\exp\{\lambda(n)\sqrt{\frac{C_1 \ln T}{n_{m,i}(t)}}\}} \cdot \exp\{\frac{\lambda^2(n)}{2}\tilde{\sigma}^2\}] \tag{49}$$

by a same argument from (43) to (47) replacing $\frac{C_2 \ln T}{n_{m,i}(t)}$ with $\sqrt{\frac{C_1 \ln T}{n_{m,i}(t)}}$. When choosing $\lambda(n) = \frac{\sqrt{\frac{C_1 \log T}{n_{m,i}(t)}}}{\tilde{\sigma}^2}$ that meets the condition $\lambda < \frac{1}{\alpha}$ under the assumption $\frac{\sqrt{\frac{C_1 \log T}{n_{m,i}(t)}}}{\tilde{\sigma}^2} < \frac{1}{\alpha}$ and noting that

$\frac{C_1}{2C\sigma^2} \geq 4$, we obtain

$$(49) \leq 2\frac{1}{P(A_{\epsilon,\delta})}E[\exp\{-\frac{C_1\log T}{\tilde{\sigma}^2 n_{m,i}(t)}\}]$$

$$= 2\frac{1}{P(A_{\epsilon,\delta})}E[\exp\{-\frac{C_1\log T}{n_{m,i}(t)}\frac{1}{\frac{2C\sigma^2}{n_{m,i}(t)}}\}]$$

$$\leq 2\frac{1}{P(A_{\epsilon,\delta})}\exp\{-4\log T\} = \frac{2}{P(A_{\epsilon,\delta})T^4}.$$

To conclude, by (41) and (42), we have

$$P(|\tilde{\mu}_i^m(t) - \mu_i| > Rad|A_{\epsilon,\delta}) \leq \frac{2}{P(A_{\epsilon,\delta})T^4}$$

which completes the proof. □

**Proposition 13.** *Assume the parameter $\delta$ satisfies that $0 < \delta < c = f(\epsilon, M, T)$. An arm $k$ is said to be sub-optimal if $k \neq i^*$ where $i^*$ is the unique optimal arm in terms of the global reward, i.e. $i^* = \arg\max \frac{1}{M}\sum_{j=1}^{M}\mu_i^j$. Then in setting $S_1, S_2$ and $S_3$, when the game ends, for every client $m$, $0 < \epsilon < 1$ and $T > L$, the expected numbers of pulling sub-optimal arm $k$ after the burn-in period satisfies with $P(A_{\epsilon,\delta}) = 1 - 7\epsilon$*

$$E[n_{m,k}(T)|A_{\epsilon,\delta}]$$
$$\leq \max([\frac{16C_1\log T}{\Delta_i^2}], [\frac{4C_2\log T}{\Delta_i}], 2(K^2 + MK + M)) + \frac{4}{P(A_{\epsilon,\delta})T^3} + K^2 + (2M-1)K$$
$$\leq O(\log T).$$

*Proof.* Recall that $Rad = \sqrt{\frac{C_1\ln T}{n_{m,i}(t)}} + \frac{C_2\ln T}{n_{m,i}(t)}$. We again have $a_t^m = i$ holds when any of the following conditions is met: Case 1: $n_{m,i}(t) \leq \mathcal{N}_{m,i}(t) - K$, Case 2: $\tilde{\mu}_{m,i} - \mu_i > \sqrt{\frac{C_1\ln T}{n_{m,i}(t)}} + \frac{C_2\ln T}{n_{m,i}(t)}$, Case 3: $-\tilde{\mu}_{m,i^*} + \mu_{i^*} > \sqrt{\frac{C_1\ln T}{n_{m,i^*}(t)}} + \frac{C_2\ln T}{n_{m,i^*}(t)}$, and Case 4: $\mu_{i^*} - \mu_i < 2(\sqrt{\frac{C_1\ln T}{n_{m,i}(t)}} + \frac{C_2\ln T}{n_{m,i}(t)})$.

By (30), the expected value of $n_{m,i}(t)$ conditional on $A_{\epsilon,\delta}$ reads as

$$E[n_{m,i}(T)|A_{\epsilon,\delta}]$$
$$= l + \sum_{t=L+1}^{T} P(Case2, n_{m,i}(t-1) \geq l|A_{\epsilon,\delta}) + \sum_{t=L+1}^{T} P(Case3, n_{m,i}(t-1) \geq l|A_{\epsilon,\delta})$$
$$+ \sum_{t=L+1}^{T} P(Case1, a_t^m = i, n_{m,i}(t-1) \geq l|A_{\epsilon,\delta}) + \sum_{t=L+1}^{T} P(Case4, n_{m,i}(t-1) \geq l|A_{\epsilon,\delta})$$
$$\tag{50}$$

where $l$ is specified as $l = \max\{[\frac{4C_1\log T}{\Delta_i^2}], 2(K^2 + MK + M)\}$ with $\Delta_i = \mu_{i^*} - \mu_i$.

For the last term in the above upper bound, we have

$$\sum_{t=L+1}^{T} P(Case4 : \mu_i + 2(\sqrt{\frac{C_1\ln T}{n_{m,i}(t)}} + \frac{C_2\ln T}{n_{m,i}(t)}) > \mu_{i^*}, n_{m,i}(t-1) \geq l) = 0 \tag{51}$$

since the choice of $l$ satisfies $l \geq \max([\frac{16C_1\log T}{\Delta_i^2}], [\frac{4C_2\log T}{\Delta_i}], 2(K^2 + MK))$.

For the first two terms, we have on event $A_{\epsilon,\delta}$

$$\sum_{t=L+1}^{T} P(Case2, n_{m,i}(t-1) \geq l|A_{\epsilon,\delta}) + \sum_{t=1}^{T} P(Case3, n_{m,i}(t-1) \geq l|A_{\epsilon,\delta})$$

$$\leq \sum_{t=L+1}^{T} P(\tilde{\mu}_{m,i} - \mu_i > \sqrt{\frac{C_1 \ln T}{n_{m,i}(t)} + \frac{C_2 \ln T}{n_{m,i}(t)}}|A_{\epsilon,\delta}) +$$

$$\sum_{t=1}^{T} P(-\tilde{\mu}_{m,i^*} + \mu_{i^*} > \sqrt{\frac{C_1 \ln T}{n_{m,i^*}(t)} + \frac{C_2 \ln T}{n_{m,i^*}(t)}}|A_{\epsilon,\delta})$$

$$\leq \sum_{t=1}^{T} (\frac{1}{P(A_{\epsilon,\delta})T^4}) + \sum_{t=1}^{T} (\frac{1}{P(A_{\epsilon,\delta})T^4}) \leq \frac{2}{P(A_{\epsilon,\delta})T^3} \quad (52)$$

where the first inequality holds by the property of the probability measure when removing the event $n_{m,i}(t-1) \geq l$ and the second inequality holds by Proposition 12, which holds by the assumption that $\delta < c$.

For Case 1, we note that Lemma 1 implies that

$$n_{m,i}(t) > N_{m,i}(t) - K(K + 2M)$$

with the definition of $N_{m,i}(t+1) = \max\{n_{m,i}(t+1), N_{j,i}(t), j \in \mathcal{N}_m(t)\}$.

Departing from the result that the difference between $N_{m,i}(t)$ and $n_{m,i}(t)$ is at most $K(K + 2M)$, we then present the following analysis on how long it takes for the value $-n_{m,i}(t) + N_{m,i}(t)$ to be smaller than $K$.

At time step $t$, if Case 1 holds for client $m$, then $n_{m,i}(t+1)$ is increasing by 1 on the basis of $n_{m,i}(t)$. What follows characterizes the change of $N_{m,i}(t+1)$. Client $m$ satisfying $n_{m,i}(t) \leq N_{m,i}(t) - K$ will not change the value of $N_{m,i}(t+1)$ by the definition $N_{m,i}(t+1) = \max\{n_{m,i}(t+1), N_{j,i}(t), j \in \mathcal{N}_m(t)\}$. Moreover, for client $j \in \mathcal{N}_m(t)$ with $n_{j,i}(t) < N_{j,i}(t) - K$, i.e. $N_{j,i}(t+1)$ will not be affected by $n_{j,i}(t+1) \leq n_{j,i}(t) + 1$. Thus, the value of $N_{m,i}(t+1) = \max\{n_{m,i}(t+1), N_{j,i}(t), j \in \mathcal{N}_m(t)\}$ is independent of such clients. We observe that for client $j \in \mathcal{N}_m(t)$ with $n_{j,i}(t) > N_{j,i}(t) - K$, the value $N_{j,i}(t)$ will be the same if the client does not sample arm $i$, which leads to a decrease of 1 in the difference $-n_{m,i}(t) + N_{m,i}(t)$. Otherwise, if such a client samples arm $i$ which brings an increment of 1 to $N_{m,i}(t)$, the difference between $n_{m,i}(t)$ and $N_{m,i}(t)$ will remain the same. However, the latter has just been discussed and must be the cases as in Case 2 and Case 3, the total length of which has already been upper bounded by $\frac{2}{P(A_{\epsilon,\delta})T^3}$ as shown in (52).

Therefore, the gap is at most $K(K + 2M) - K + \frac{2}{P(A_{\epsilon,\delta})T^3}$, i.e.

$$\sum_{t=1}^{T} P(Case1, a_t^m = i, n_{m,i}(t-1) \geq l|A_{\epsilon,\delta}) \leq K(K + 2M) - K + \frac{2}{P(A_{\epsilon,\delta})T^3}. \quad (53)$$

Subsequently, we derive that

$$E[n_{m,i}(T)|A_{\epsilon,\delta}]$$

$$\leq l + \frac{2}{P(A_{\epsilon,\delta})T^3} + K(K + 2M) - K + \frac{2}{P(A_{\epsilon,\delta})T^3} + 0$$

$$= l + \frac{2\pi^2}{3} + K^2 + (2M - 1)K$$

$$= \max([\frac{16C_1 \log T}{\Delta_i^2}], [\frac{4C_2 \log T}{\Delta_i}], 2(K^2 + MK + M)) + \frac{4}{P(A_{\epsilon,\delta})T^3} + K^2 + (2M - 1)K$$

where the inequality results from (50), (51), (52) and (53).

$\square$

### F.2   Proof of Theorems

**Theorem 1.** *For event $A_{\epsilon,\delta}$ and any $1 > \epsilon, \delta > 0$, we have $P(A_{\epsilon,\delta}) \geq 1 - 7\epsilon$.*

*Proof.* Recall that we define events

$$A_1 = \{\forall t \geq L, |P_t - cE| \leq \delta\},$$
$$A_2 = \{\forall t \geq L, \forall j, m, t + 1 - \min_j t_{m,j} \leq t_0 \leq c_0 \min_l n_{l,i}(t+1)\},$$
$$A_3 = \{\forall t \geq L, G_t \text{ is connected}\}$$

where $A_1, A_2, A_3$ belong to the $\sigma$-algebra in the probability space since the time horizon is countable, i.e. for the probability space $(\Omega, \Sigma, P)$, $A_1, A_2, A_3 \in \Sigma$.

Meanwhile, we obtain

$$\begin{aligned}
P(A_1) &= P(\{\forall t \geq L, |P_t - cE| \leq \delta\}) \\
&\geq P(\cap_i \{\forall t \geq L_{s_i}, |P_t - cE| \leq \delta\}) \\
&\geq 1 - \sum_i (1 - P(\cap_i \{\forall t \geq L_{s_i}, |P_t - cE| \leq \delta\})) \\
&\geq 1 - \sum_i (1 - (1 - \epsilon)) \\
&= 1 - 3\epsilon
\end{aligned} \tag{54}$$

where the first inequality includes all settings and $L \geq L_{s_i}$, the second inequality results from the Bonferroni's inequality and the third inequality holds by Proposition 1 and Proposition 3.

At the same time, note that

$$\begin{aligned}
P(A_2) &= P(\{\forall t \geq L, \forall j, m, t + 1 - \min_j t_{m,j} \leq t_0 \leq c_0 \min_l n_{l,i}(t+1)\}) \\
&\geq P(\cap_i \{\forall t \geq L_{s_i}, \forall j, m, t + 1 - \min_j t_{m,j} \leq t_0 \leq c_0 \min_l n_{l,i}(t+1)\}) \\
&\geq 1 - \sum_i (1 - P(\{\forall t \geq L_{s_i}, \forall j, m, t + 1 - \min_j t_{m,j} \leq t_0 \leq c_0 \min_l n_{l,i}(t+1)\})) \\
&\geq 1 - \sum_i (1 - (1 - \epsilon)) = 1 - 3\epsilon
\end{aligned} \tag{55}$$

where the first inequality is by the definition of $L$, the second inequality again uses the Bonferroni's inequality, and the third inequality results from Proposition 6.

Moreover, we observer that

$$\begin{aligned}
P(A_3) &= P(\{\forall t \geq L, G_t \text{ is connected}\}) \\
&\geq P(\cap_i \{\forall t \geq L_{s_i}, G_t \text{ is connected}\}) \\
&\geq 1 - \sum_i (1 - P(\{\forall t \geq L_{s_i}, G_t \text{ is connected}\})) \\
&\geq 1 - (1 - (1 - \epsilon)) - 0 = 1 - \epsilon
\end{aligned} \tag{56}$$

where the first inequality uses the definition of $L$, the second inequality is by the Bonferroni's inequality and the third inequality holds by Proposition 5 and the definition of $s_2, s_3$ where all graphs are guaranteed to be connected.

Consequently, we arrive at

$$\begin{aligned}
P(A_{\epsilon,\delta}) &= P(A_1 \cap A_2 \cap A_3) \\
&= 1 - P(A_1^c \cup A_2^c \cup A_3^c) \\
&\geq 1 - (P(A_1^c) + P(A_2^c) + P(A_3^c)) \\
&\geq 1 - (3\epsilon + 3\epsilon + \epsilon) = 1 - 7\epsilon
\end{aligned}$$

where the first inequality utilizes the Bonferroni's inequality, the second inequality results from (54), (55), and (56).

This concludes the proof and shows the validness of the statement.

$\square$

**Theorem 2.** *Let $f$ be a function specific to a setting and detailed later. For every $0 < \epsilon < 1$ and $0 < \delta < f(\epsilon, M, T)$, in setting $s_1$ with $c \geq \frac{1}{2} + \frac{1}{2}\sqrt{1 - (\frac{\epsilon}{MT})^{\frac{2}{M-1}}}$, $s_2$ and $s_3$, with the time horizon $T$ satisfying $T \geq L$, the regret of Algorithm 2 with $F(m, i, t) = \sqrt{\frac{C_1 \ln t}{n_{m,i}(t)}}$ satisfies that*

$$E[R_T | A_{\epsilon,\delta}] \leq L + \sum_{i \neq i^*} (\max\{[\frac{4C_1 \log T}{\Delta_i^2}], 2(K^2 + MK + M)\} + \frac{2\pi^2}{3P(A_{\epsilon,\delta})} + K^2 + (2M-1)K$$

*where the length of the burn-in period is explicitly*

$$L = \max \left\{ \underbrace{\frac{\ln \frac{T}{2\epsilon}}{2\delta^2}, \frac{4K \log_2 T}{c_0}}_{L_{s_1}}, \underbrace{\frac{\ln \frac{\delta}{10}}{\ln p^*} + 25\frac{1+\lambda}{1-\lambda}\frac{\ln \frac{T}{2\epsilon}}{2\delta^2}, \frac{4K \log_2 T}{c_0}}_{L_{s_2}}, \right.$$

$$\left. \underbrace{\frac{\ln \frac{\delta}{10}}{\ln p^*} + 25\frac{1+\lambda}{1-\lambda}\frac{\ln \frac{T}{2\epsilon}}{2\delta^2}, \frac{K \ln(\frac{MT}{\epsilon})}{\ln(\frac{1}{1 - \frac{2 \log M}{M-1}})}}_{L_{s_3}} \right\}$$

*with $\lambda$ being the spectral gap of the Markov chain in $s_2, s_3$ that satisfies $1 - \lambda \geq \frac{1}{2\frac{\ln 2}{\ln 2p^*}\ln 4+1}$, $p^* = p^*(M) < 1$ and $c_0 = c_0(K, \min_{i \neq i^*} \Delta_i, M, \epsilon, \delta)$, and the instance-dependent constant $C_1 = 8\sigma^2 \max\{12\frac{M(M+2)}{M^4}\}$.*

*Proof.* The optimal arm is denoted as $i^*$ satisfying

$$i^* = \arg\max_i \sum_{m=1}^{M} \mu_i^m.$$

For the proposed regret, we have that for any constant $L$,

$$R_T = \frac{1}{M}(\max_i \sum_{t=1}^{T} \sum_{m=1}^{M} \mu_i^m - \sum_{t=1}^{T} \sum_{m=1}^{M} \mu_{a_t^m}^m)$$

$$= \sum_{t=1}^{T} \frac{1}{M} \sum_{m=1}^{M} \mu_{i^*}^m - \sum_{t=1}^{T} \frac{1}{M} \sum_{m=1}^{M} \mu_{a_t^m}^m$$

$$\leq \sum_{t=1}^{L} |\frac{1}{M} \sum_{m=1}^{M} \mu_{i^*}^m - \frac{1}{M} \sum_{m=1}^{M} \mu_{a_t^m}^m| + \sum_{t=L+1}^{T} (\frac{1}{M} \sum_{m=1}^{M} \mu_{i^*}^m - \frac{1}{M} \sum_{m=1}^{M} \mu_{a_t^m}^m)$$

$$\leq L + \sum_{t=L+1}^{T} (\frac{1}{M} \sum_{m=1}^{M} \mu_{i^*}^m - \frac{1}{M} \sum_{m=1}^{M} \mu_{a_t^m}^m)$$

$$= L + \sum_{t=L+1}^{T} (\mu_{i^*} - \frac{1}{M} \sum_{m=1}^{M} \mu_{a_t^m}^m)$$

$$= L + ((T - L) \cdot \mu_{i^*} - \frac{1}{M} \sum_{m=1}^{M} \sum_{i=1}^{K} n_{m,i}(T)\mu_i^m)$$

where the first inequality is by taking the absolute value and the second inequality results from the assumption that $0 < \mu_i^j < 1$ for any arm $i$ and client $j$.

Note that $\sum_{i=1}^{K} \sum_{m=1}^{M} n_{m,i}(T) = M(T - L)$ where by definition $n_{m,i}(T)$ is the number of pulls of arm $i$ at client $m$ from time step $L + 1$ to time step $T$, which yields that

$$R_T \leq L + \sum_{i=1}^{K} \frac{1}{M} \sum_{m=1}^{M} n_{m,i}(T) \mu_{i^*}^m - \sum_{i=1}^{K} \frac{1}{M} \sum_{m=1}^{M} n_{m,i}(T) \mu_i^m$$

$$= L + \sum_{i=1}^{K} \frac{1}{M} \sum_{m=1}^{M} n_{m,i}(T)(\mu_{i^*}^m - \mu_i^m)$$

$$\leq L + \frac{1}{M} \sum_{i=1}^{K} \sum_{m:\mu_{i^*}^m - \mu_i^m > 0} n_{m,i}(T)(\mu_{i^*}^m - \mu_i^m)$$

$$= L + \frac{1}{M} \sum_{i \neq i^*} \sum_{m:\mu_{i^*}^m - \mu_i^m > 0} n_{m,i}(T)(\mu_{i^*}^m - \mu_i^m).$$

where the second inequality uses the fact that $\sum_{m:\mu_{i^*}^m - \mu_i^m \leq 0} n_{m,i}(T)(\mu_{i^*}^m - \mu_i^m) \leq 0$ holds for any arm $i$ and the last equality is true since $n_{m,i}(T)(\mu_{i^*}^m - \mu_i^m) = 0$ for $i = i^*$ and any $m$.

Meanwhile, by the choices of $\delta$ such that $\delta < c = f(\epsilon, M, T)$, we apply Proposition 10 which leads to for any client $m$ and arm $i \neq i^*$,

$$E[n_{m,i}(T)|A_{\epsilon,\delta}] \leq \max \left\{ \left[ \frac{4C_1 \log T}{\Delta_i^2} \right], 2(K^2 + MK + M) \right\} + \frac{2\pi^2}{3} + K^2 + (2M - 1)K. \quad (57)$$

As a result, the upper bound on $R_T$ can be derived as by taking the conditional expectation over $R_T$ on $A_{\epsilon,\delta}$

$$E[R_T|A_{\epsilon,\delta}]$$

$$\leq L + \frac{1}{M} \sum_{i \neq i^*} \sum_{m:\mu_{i^*}^m - \mu_i^m > 0} E[n_{m,i}(T)|A_{\epsilon,\delta}](\mu_{i^*}^m - \mu_i^m) \quad (58)$$

$$\leq L +$$

$$\frac{1}{M} \sum_{i \neq i^*} \sum_{m:\mu_{i^*}^m - \mu_i^m > 0} \left( \max \left\{ \left[ \frac{4C_1 \log T}{\Delta_i^2} \right], 2(K^2 + MK) \right\} + \frac{2\pi^2}{3} + K^2 + (2M - 1)K)(\mu_{i^*}^m - \mu_i^m) \right)$$

$$= L +$$

$$\frac{1}{M} \sum_{i \neq i^*} \left( \max \left\{ \left[ \frac{4C_1 \log T}{\Delta_i^2} \right], 2(K^2 + MK) \right\} + \frac{2\pi^2}{3} + K^2 + (2M - 1)K \right) \sum_{m:\mu_{i^*}^m - \mu_i^m > 0} (\mu_{i^*}^m - \mu_i^m)$$

$$(59)$$

where the second inequality holds by plugging in (57).

Meanwhile, we note that for any $i \neq i^*$,

$$\sum_{m:\mu_{i^*}^m - \mu_i^m > 0} (\mu_{i^*}^m - \mu_i^m) + \sum_{m:\mu_{i^*}^m - \mu_i^m \leq 0} (\mu_{i^*}^m - \mu_i^m)$$

$$= \sum_{m=1}^{M} (\mu_{i^*}^m - \mu_i^m)$$

$$= M\Delta_i > 0$$

and

$$\left| \sum_{m:\mu_{i^*}^m - \mu_i^m \leq 0} (\mu_{i^*}^m - \mu_i^m) \right| \leq M$$

which gives us that

$$\sum_{m:\mu_{i^*}^m - \mu_i^m > 0} (\mu_{i^*}^m - \mu_i^m)$$

$$= M\Delta_i - \sum_{m:\mu_{i^*}^m - \mu_i^m \leq 0} (\mu_{i^*}^m - \mu_i^m)$$

$$= M\Delta_i + |\sum_{m:\mu_{i^*}^m - \mu_i^m \leq 0} (\mu_{i^*}^m - \mu_i^m)|$$

$$\leq M\Delta_i + M = M(\Delta_i + 1). \tag{60}$$

Hence, the regret can be upper bounded by

(59)

$$\leq L + \sum_{i \neq i^*} (\Delta_i + 1)(\max\{[\frac{4C_1 \log T}{\Delta_i^2}], 2(K^2 + MK + M)\} + \frac{2\pi^2}{3} + K^2 + (2M - 1)K)$$

$$= O(\max\{L, \log T\})$$

where the inequality is derived from (60) and $L$ is the same constant as in the definition of $A_{\epsilon,\delta}$.

This completes the proof.

$\square$

**Theorem 3.** *Let $f$ be a function specific to a setting and defined in the above remark. For every $0 < \epsilon < 1$ and $0 < \delta < f(\epsilon, M, T)$, in settings $S_1$ with $c \geq \frac{1}{2} + \frac{1}{2}\sqrt{1 - (\frac{\epsilon}{MT})^{\frac{2}{M-1}}}$, $S_2$, $S_3$ with the time horizon $T$ satisfying $T \geq L$, the regret of Algorithm 2 with $F(m, i, t) = \sqrt{\frac{C_1 \ln T}{n_{m,i}(t)}} + \frac{C_2 \ln T}{n_{m,i}(t)}$ satisfies*

$$E[R_T | A_{\epsilon,\delta}] \leq L + \sum_{i \neq i^*} (\Delta_i + 1) \cdot (\max([\frac{16C_1 \log T}{\Delta_i^2}], [\frac{4C_2 \log T}{\Delta_i}], 2(K^2 + MK + M))$$

$$+ \frac{4}{P(A_{\epsilon,\delta})T^3} + K^2 + (2M - 1)K$$

*where $L, C_1$ are specified as in Theorem 2 and $\frac{C_2}{C_1} \geq \frac{3}{2}$.*

*Proof.* By the regret decomposition as in (58), we obtain that

$$E[R_T | A_{\epsilon,\delta}] \leq L + \frac{1}{M} \sum_{i \neq i^*} \sum_{m:\mu_{i^*}^m - \mu_i^m > 0} E[n_{m,i}(T) | A_{\epsilon,\delta}](\mu_{i^*}^m - \mu_i^m). \tag{61}$$

By Proposition 13, we have that with probability at least $1 - 7\epsilon$

$$E[n_{m,i}(T) | A_{\epsilon,\delta}]$$

$$\leq \max([\frac{16C_1 \log T}{\Delta_i^2}], [\frac{4C_2 \log T}{\Delta_i}], 2(K^2 + MK + M)) + \frac{4}{P(A_{\epsilon,\delta})T^3} + K^2 + (2M - 1)K. \tag{62}$$

Following (60) gives us that

$$\sum_{m:\mu_{i^*}^m - \mu_i^m > 0} (\mu_{i^*}^m - \mu_i^m) \leq M\Delta_i + M = M(\Delta_i + 1). \tag{63}$$

Therefore, we derive that with probability at least $P(A_{\epsilon,\delta}) = 1 - 7\epsilon$

$$E[R_T | A_{\epsilon,\delta}] \leq L + \sum_{i \neq i^*} (\Delta_i + 1) \cdot (\max([\frac{16C_1 \log T}{\Delta_i^2}], [\frac{4C_2 \log T}{\Delta_i}], 2(K^2 + MK + M))$$

$$+ \frac{4}{P(A_{\epsilon,\delta})T^3} + K^2 + (2M - 1)K$$

which completes the proof.

$\square$

**Theorem 4.** *Assume the same conditions as in Theorems 2 and 3. The regret of Algorithm 2 satisfies that*

$$E[R_T|A_{\epsilon,\delta}] \leq L_1 + \frac{4}{P(A_{\epsilon,\delta})T^3}+$$

$$(1 + \max\{\sqrt{C_1 \ln T}, C_2 \ln T\})(K(K + 2M) - K + \frac{2}{P(A_{\epsilon,\delta})T^3})+$$

$$K(C_2(\ln T)^2 + C_2 \ln T + \sqrt{C_1 \ln T}\sqrt{T(\ln T + 1)}) = O(\sqrt{T} \ln T).$$

*where $L_1 = \max(L, K(2(K^2 + MK + M)))$, $L, C_1$ is specified as in Theorem 2, and $\frac{C_2}{C_1^2} \geq \frac{3}{2}$. The involved constants depend on $\sigma^2$ but not on $\Delta_i$.*

*Proof.* Define $U_m^t(i)$ and $L_m^t(i)$ as $\tilde{\mu}_i^m(t) + Rad(i, m, t)$ and $\tilde{\mu}_i^m(t) - Rad(i, m, t)$, respectively, where $Rad$ is previously defined as $Rad(i, m, t) = \sqrt{\frac{C_1 \ln T}{n_{m,i}(t)} + \frac{C_2 \ln T}{n_{m,i}(t)}}$. We observe that by definition, the regret $R_T$ can be written as

$$R_T = \frac{1}{M} \sum_{t=1}^{T} \sum_{m=1}^{M} (\mu_{i^*} - \mu_{a_m^t}^t)$$

$$= \frac{1}{M} \sum_{t=1}^{T} \sum_{m=1}^{M} (\mu_{i^*} - U_m^t(a_m^t) + U_m^t(a_m^t) - L_m^t(a_m^t) + L_m^t(a_m^t) - \mu_{a_m^t}^t).$$

Subsequently, the conditional expectation of $R_T$ has the following decomposition

$$E[R_T|A_{\epsilon,\delta}]$$

$$= \frac{1}{M} \sum_{t=1}^{T} \sum_{m=1}^{M} (E[\mu_{i^*} - U_m^t(a_m^t)|A_{\epsilon,\delta}] + E[U_m^t(a_m^t) - L_m^t(a_m^t)|A_{\epsilon,\delta}] + E[L_m^t(a_m^t) - \mu_{a_m^t}^t|A_{\epsilon,\delta}])$$

$$= L_1 + \frac{1}{M} \sum_{t=L_1+1}^{T} \sum_{m=1}^{M} (E[\mu_{i^*} - U_m^t(i^*)|A_{\epsilon,\delta}] + E[U_m^t(i^*) - U_m^t(a_m^t)|A_{\epsilon,\delta}]+$$

$$E[U_m^t(a_m^t) - L_m^t(a_m^t)|A_{\epsilon,\delta}] + E[L_m^t(a_m^t) - \mu_{a_m^t}^t|A_{\epsilon,\delta}]) \tag{64}$$

where $L_1 = \max(L, 2(K^2 + MK + M))$.

For the first term, we derive its upper bound as follows.

Note that

$$E[\mu_{i^*} - U_m^t(i^*)|A_{\epsilon,\delta}]$$

$$\leq E[(\mu_{i^*} - U_m^t(i^*))1_{\mu_{i^*} - U_m^t(i^*)>0}|A_{\epsilon,\delta}]$$

$$= E[\mu_{i^*}1_{\mu_{i^*} - U_m^t(i^*)>0}|A_{\epsilon,\delta}] - E[U_m^t 1_{\mu_{i^*} - U_m^t(i^*)>0}|A_{\epsilon,\delta}]$$

$$\leq E[\mu_{i^*}1_{\mu_{i^*} - U_m^t(i^*)>0}|A_{\epsilon,\delta}]$$

$$\leq E[1_{\mu_{i^*} - U_m^t(i^*)>0}|A_{\epsilon,\delta}]$$

$$= P(\mu_{i^*} - U_m^t(i^*) > 0|A_{\epsilon,\delta})$$

$$= P(\mu_{i^*} - \tilde{\mu}_{i^*}^m(t) > Rad|A_{\epsilon,\delta})$$

$$\leq P(|\mu_{i^*} - \tilde{\mu}_{i^*}^m(t)| > Rad|A_{\epsilon,\delta}) \leq \frac{2}{P(A_{\epsilon,\delta})T^4} \tag{65}$$

where the first inequality uses the monotone property of $E[\cdot]$, the second inequality omits the latter negative quantity, the third inequality holds by the fact that $0 \leq \mu_i^* \leq 1$, and the last inequality is by Proposition 12.

In like manner, we have that the last term satisfies that

$$E[L_m^t(a_m^t) - \mu_{a_m^t}|A_{\epsilon,\delta}] \le \frac{2}{P(A_{\epsilon,\delta})T^4} \tag{66}$$

by the same logic as the above and substituting $i^*$ with $a_m^t$, and thus we omit the details here.

We then proceed to bound the second term. Based on the decision rule in Algorithm 2, we have either $E[U_m^t(i^*) - U_m^t(a_m^t)|A_{\epsilon,\delta}] < 0$ or $n_{m,i}(t) < N_{m,i}(t) - K$. This is equivalent to

$$\begin{aligned}
&E[U_m^t(i^*) - U_m^t(a_m^t)|A_{\epsilon,\delta}] \\
&= E[U_m^t(i^*) - U_m^t(a_m^t)1_{n_{m,i}(t) \ge N_{m,i}(t)-K}|A_{\epsilon,\delta}] + \\
&\qquad E[U_m^t(i^*) - U_m^t(a_m^t)1_{n_{m,i}(t) < N_{m,i}(t)-K}|A_{\epsilon,\delta}] \\
&\le E[U_m^t(i^*) - U_m^t(a_m^t)1_{n_{m,i}(t) < N_{m,i}(t)-K}|A_{\epsilon,\delta}] \\
&\le E[U_m^t(i^*)1_{n_{m,i}(t) < N_{m,i}(t)-K}|A_{\epsilon,\delta}].
\end{aligned} \tag{67}$$

By definition, $U_m^t(i^*) = \tilde{\mu}_{i^*} + Rad(i^*,m,t)$ implies that

$$U_m^t(i^*) \le 1 + Rad(i^*,m,t)$$

which leads to

$$\begin{aligned}
(67) &\le E[(1 + Rad(i^*,m,t))1_{n_{m,i}(t) < N_{m,i}(t)-K}|A_{\epsilon,\delta}] \\
&\le (1 + \max Rad(i^*,m,t))E[1_{n_{m,i}(t) < N_{m,i}(t)-K}|A_{\epsilon,\delta}] \\
&\le (1 + \max\{\sqrt{C_1 \ln T}, C_2 \ln T\})E[1_{n_{m,i}(t) < N_{m,i}(t)-K}|A_{\epsilon,\delta}]
\end{aligned}$$

and subsequently

$$\begin{aligned}
&\frac{1}{M}\sum_{t=L_1+1}^{T}\sum_{m=1}^{M} E[U_m^t(i^*) - U_m^t(a_m^t)|A_{\epsilon,\delta}] \\
&\le \frac{1}{M}\sum_{t=L_1+1}^{T}\sum_{m=1}^{M}(1 + \max\{\sqrt{C_1 \ln T}, C_2 \ln T\})E[1_{n_{m,i}(t) < N_{m,i}(t)-K}|A_{\epsilon,\delta}]. \tag{68}
\end{aligned}$$

Following (53) that only depends on whether clients stay on the same page that relies on the transmission, we obtain

$$\sum_t E[1_{n_{m,i}(t) < N_{m,i}(t)-K}|A_{\epsilon,\delta}] \le K(K+2M) - K + \frac{2}{P(A_{\epsilon,\delta})T^3}$$

which immediately leads to

$$(68) \le (1 + \max\{\sqrt{C_1 \ln T}, C_2 \ln T\}) \cdot (K(K+2M) - K + \frac{2}{P(A_{\epsilon,\delta})T^3})$$

Afterwards, we consider the third term and have

$$\begin{aligned}
&E[U_m^t(a_m^t) - L_m^t(a_m^t)|A_{\epsilon,\delta}] \\
&= E[2Rad(a_m^t,m,t)|A_{\epsilon,\delta}] \tag{69}
\end{aligned}$$

Putting (65, 66, 68, 69) all together, we deduce that

$$\begin{aligned}
(64) &\le L_1 + \frac{1}{M}\sum_{t=L_1+1}^{T}\sum_{m=1}^{M}(\frac{2}{P(A_{\epsilon,\delta})T^4} + E[2Rad(a_m^t,m,t)|A_{\epsilon,\delta}] + \frac{2}{P(A_{\epsilon,\delta})T^4}) \\
&\quad + (1 + \max\{\sqrt{C_1 \ln T}, C_2 \ln T\}) \cdot (K(K+2M) - K + \frac{2}{P(A_{\epsilon,\delta})T^3}) \\
&\le L_1 + \frac{4}{P(A_{\epsilon,\delta})T^3} + \frac{1}{M}\sum_{t>L_1}\sum_{m}(E[2Rad(a_m^t,m,t)|A_{\epsilon,\delta}]) + \\
&\quad (1 + \max\{\sqrt{C_1 \ln T}, C_2 \ln T\}) \cdot (K(K+2M) - K + \frac{2}{P(A_{\epsilon,\delta})T^3}).
\end{aligned}$$

Meanwhile, we observe that by definition

$$\frac{1}{M} \sum_{t > L_1} \sum_m (E[2Rad(a_m^t, m, t)|A_{\epsilon,\delta}])$$

$$= \frac{1}{M} \sum_i \sum_m \sum_{\substack{a_m^t = i \\ t > L_1}} (E[2Rad(i, m, t)|A_{\epsilon,\delta}])$$

$$= \frac{1}{M} \sum_i \sum_m \sum_{\substack{a_m^t = i \\ t > L_1}} E[2\sqrt{\frac{C_1 \ln T}{n_{m,i}(t)}} + \frac{C_2 \ln T}{n_{m,i}(t)} |A_{\epsilon,\delta}]. \tag{70}$$

By the sum of the Harmonic series, we have

$$\sum_{\substack{a_m^t = i \\ t > L_1}} \frac{C_2 \ln T}{n_{m,i}(t)} \le C_2 \ln T \ln n_{m,i}(T) + C_2 \ln T \le C_2 (\ln T)^2 + C_2 \ln T. \tag{71}$$

Meanwhile, by the Cauchy-Schwartz inequality, we obtain

$$\sum_{\substack{a_m^t = i \\ t > L_1}} \sqrt{\frac{C_1 \ln T}{n_{m,i}(t)}}$$

$$\le \sqrt{C_1 \ln T} \sqrt{(\sum_t 1)(\sum_t (\sqrt{\frac{1}{n_{m,i}(t)}})^2)}$$

$$\le \sqrt{C_1 \ln T} \sqrt{T(\ln T + 1)}$$

where the last inequality again uses the result on the Harmonic series as in (71).

Therefore, the cumulative value can be bounded as

$$(70) \le \frac{1}{M} \sum_i \sum_m (C_2 (\ln T)^2 + C_2 \ln T + \sqrt{C_1 \ln T} \sqrt{T(\ln T + 1)})$$

$$= K(C_2 (\ln T)^2 + C_2 \ln T + \sqrt{C_1 \ln T} \sqrt{T(\ln T + 1)})$$

Using the result of (70), we have

$$(64) \le L_1 + \frac{4}{P(A_{\epsilon,\delta})T^3} + K(C_2 (\ln T)^2 + C_2 \ln T + \sqrt{C_1 \ln T} \sqrt{T(\ln T + 1)}) +$$

$$(1 + \max\{\sqrt{C_1 \ln T}, C_2 \ln T\}) \cdot (K(K + 2M) - K + \frac{2}{P(A_{\epsilon,\delta})T^3})$$

$$= O(\max\{\sqrt{T} \ln T, (\ln T)^2\})$$

which completes the proof.

$\square$

# G   Choices of parameter $c_0$ in Theorem 2

**Parameter** $c_0$   We note that $c_0$ is a pre-specified parameter which are different in different settings. The choices of $c_0$ are as follows. Meanwhile, we need to study whether the possible choices of $c_0$ explode in terms of the order of $T$.

**Remark** (2). *The regret reads*

$$E[R_T|A_{\epsilon,\delta}] \le L + C_1 \sum_{i \ne i^*} ([\frac{4 \log T}{\Delta_i^2}]) + (K - 1)(2(K^2 + MK) + \frac{2\pi^2}{3} + K^2 + (2M - 1)K)$$

with $L$ denoted as $L = \max\{L_1, L_2, L_3\} = \max\{a_1, a_2, a_3, \frac{b_1}{c_0}, \frac{b_2}{c_0}, \frac{b_3}{c_0}\}$ and $C_1$ denoted as $\max\{\frac{e}{1-c_0}, f\}$, where parameters $a_1, a_2, a_3, b_1, b_2, b_3, e, f$ are specified as

$$a_1 = \frac{\ln\frac{2T}{\epsilon}}{2\delta^2}$$

$$b_1 = 4K\log_2 T$$

$$a_2 = \frac{\ln\frac{\delta}{10}}{\ln p^*} + 25\frac{1+\lambda}{1-\lambda}\frac{\ln\frac{2T}{\epsilon}}{2\delta^2}$$

$$b_2 = 4K\log_2 T$$

$$a_3 = \frac{\ln\frac{\delta}{10}}{\ln p^*} + 25\frac{1+\lambda}{1-\lambda}\frac{\ln\frac{2T}{\epsilon}}{2\delta^2}$$

$$b_3 = \frac{K\ln(\frac{MT}{\epsilon})}{\ln(\frac{1}{1-c})}$$

$$e = 16\frac{4(M+2)}{3M}$$

$$f = 16(M+2)(1+4Md_{m,t}^2).$$

This function of $c_0$ is non-differentiable which brings additional challenges and requires a case-by-case analysis.

Let $a = \{a_1, a_2, a_3\}$ and $b = \{b_1, b_2, b_3\}$. Then continue with the decision rule as in the previous discussion.

- Case 1: there exists $c_0$ such that $a \geq \frac{b}{c_0}$, i.e. $c \geq \frac{b}{a}$ and $\frac{b}{a} \leq 1$ Then $R_T$ is monotone increasing in $c$ due to $C_1$ and $c_0 = \frac{b}{a}$ gives us the optimal regret $R_T^1$.

- Case 2: if $a \leq \frac{b}{c_0}$, i.e. $c_0 \leq \frac{b}{a}$

    - if $\frac{e}{1-c_0} < f$, i.e. $c_0 \leq 1 - \frac{e}{f}$, then $c_0 = \min\left\{\frac{b}{a}, 1 - \frac{e}{f}\right\}$ is the minima.

    - else we have $c_0 \geq 1 - \frac{e}{f}$

        * if $1 - \frac{e}{f} > \frac{b}{a}$, it leads to contradiction and this can not be the case.

        * else $1 - \frac{e}{f} \leq \frac{b}{a}$, we obtain

        $$R_T \leq \frac{b}{c_0} + \frac{e}{1-c_0}\sum_{i\neq i^*}(\lceil\frac{4\log T}{\Delta_i^2}\rceil) + (K-1)(2(K^2+MK) + \frac{2\pi^2}{3} + K^2 + (2M-1)K)$$

        which implies that the optimal choice of $c_0$ is $\frac{\sqrt{b}}{\sqrt{b}+\sqrt{e\sum_{i\neq i^*}(\lceil\frac{4\log T}{\Delta_i^2}\rceil)}}$

        · if $1 - \frac{e}{f} \leq \frac{\sqrt{b}}{\sqrt{b}+\sqrt{e\sum_{i\neq i^*}(\lceil\frac{4\log T}{\Delta_i^2}\rceil)}} \leq \frac{b}{a}$, this gives us the final choice of $c_0$ and the subsequent local optimal regret $R_T^2$.

        · elif $\frac{\sqrt{b}}{\sqrt{b}+\sqrt{e\sum_{i\neq i^*}(\lceil\frac{4\log T}{\Delta_i^2}\rceil)}} < 1 - \frac{e}{f}$, the optimal choice of $c_0$ is $1 - \frac{e}{f}$ and the subsequent local optimal regret is $R_T^2$

        · else the optimal choice of $c_0$ is $\frac{b}{a}$ and the subsequent local optimal regret is $R_T^2$

- Compare $R_T^1$ and $R_T^2$ and choose the $c_0$ associated with the smaller value.

*The possible choices of $c_0$ are $\{\frac{b}{a}, \min\{\frac{b}{a}, 1-\frac{e}{f}\}, \frac{\sqrt{b}}{\sqrt{b}+\sqrt{e\sum_{i\neq i^*}([\frac{4\log T}{\Delta_i^2}])}}\}$, i.e.*

$$c_0 = \{\frac{b_1+b_2+b_3}{a_1+a_2+a_3}, \min\{\frac{b_1+b_2+b_3}{a_1+a_2+a_3}, 1-\frac{16\frac{4(M+2)}{3M}}{16(M+2)(1+4Md_{m,t}^2)}\}, \frac{\sqrt{b}}{\sqrt{b}+\sqrt{e\sum_{i\neq i^*}([\frac{4\log T}{\Delta_i^2}])}}\}$$

$$= \{\frac{b_1+b_2+b_3}{a_1+a_2+a_3}, \min\{\frac{b_1+b_2+b_3}{a_1+a_2+a_3}, 1-\frac{16\frac{4(M+2)}{3M}}{16(M+2)(1+4Md_{m,t}^2)}\}, \frac{\sqrt{b}}{\sqrt{b}+\sqrt{e\sum_{i\neq i^*}([\frac{4\log T}{\Delta_i^2}])}}\}$$

$$= \frac{8K\log T + \frac{K\ln(\frac{MT}{\epsilon})}{\ln(\frac{1}{1-c})}}{\frac{\ln\frac{2T}{\epsilon}}{2\delta^2} + 2\frac{\ln\frac{\delta}{10}}{\ln p^*} + 50\frac{1+\lambda}{1-\lambda}\frac{\ln\frac{2T}{\epsilon}}{2\delta^2}},$$

$$\min\{\frac{8K\log T + \frac{K\ln(\frac{MT}{\epsilon})}{\ln(\frac{1}{1-c})}}{\frac{\ln\frac{2T}{\epsilon}}{2\delta^2} + 2\frac{\ln\frac{\delta}{10}}{\ln p^*} + 50\frac{1+\lambda}{1-\lambda}\frac{\ln\frac{2T}{\epsilon}}{2\delta^2}}, 1-\frac{16\frac{4(M+2)}{3M}}{16(M+2)(1+4Md_{m,t}^2)}\},$$

$$\frac{\sqrt{8K\log T + \frac{K\ln(\frac{MT}{\epsilon})}{\ln(\frac{1}{1-c})}}}{\sqrt{8K\log T + \frac{K\ln(\frac{MT}{\epsilon})}{\ln(\frac{1}{1-c})}} + \sqrt{e\sum_{i\neq i^*}([\frac{4\log T}{\Delta_i^2}])}}$$

*which implies the choice of $c_0$ is between $\frac{\sqrt{8K\log T + \frac{K\ln(\frac{MT}{\epsilon})}{\ln(\frac{1}{1-c})}}}{\sqrt{8K\log T + \frac{K\ln(\frac{MT}{\epsilon})}{\ln(\frac{1}{1-c})}} + \sqrt{e\sum_{i\neq i^*}([\frac{4\log T}{\Delta_i^2}])}}$ and $1-$*

*$\frac{16\frac{4(M+2)}{3M}}{16(M+2)(1+4Md_{m,t}^2)}$. Meanwhile, we observe that the choice of $c_0$ satisfies*

$$E[R_T|A_{\epsilon,\delta}] \leq R_T(1-\frac{16\frac{4(M+2)}{3M}}{16(M+2)(1+4Md_{m,t}^2)}) = O(\log T).$$

