# OpenReview forum: "Decentralized Randomly Distributed Multi-agent Multi-armed Bandit with Heterogeneous Rewards"
_NeurIPS.cc/2023/Conference — NeurIPS 2023 spotlight_

### Official Review · Reviewer_H8V9 · 2023-07-05

**Soundness:** 3 good
**Presentation:** 3 good
**Contribution:** 3 good
**Rating:** 6
**Confidence:** 2

**Summary:**

This paper studies the problem of multi-agent multi-armed bandits where the arm rewards generated for each agent is heterogeneous across agents, and the goal is to find the global optimal arm, where the reward of the global optimal arm is the mean of the mean reward observed by agents. In addition, this work considers that the agents are connected through an incomplete and time-varying topology. Both heterogeneous agent-specific reward and time-varying underlying topology make sense from a practical perspective and also add intellectually interesting technical challenges to the multi-agent MAB problem. The authors propose a series of algorithms for graph generation and bandit learning and then analyze the regret of the proposed algorithms under multiple different assumptions and in both problem-dependent and problem-independent settings. The regret bounds are interesting and (near) optimal in most cases.

**Strengths:**

++ This paper addresses a very challenging setting of multi-agent MAB problem where reward generation is heterogeneous and the graph is time-varying.

++ The presentation of the paper is clear and solid and it clearly explains the challenges and missing points in the existing literature.

++ The proposed algorithms make sense intuitively.

++ The statements of the results sound solid (even though, I cannot confirm the details of the proof) and significant, with sufficient explanations in comparison with the literature.

**Weaknesses:**

-- Overall, this paper is a solid work with novel and non-trivial contributions. Unfortunately, the technical part of this paper is beyond my core expertise, and I can't note on the correctness of the proofs.

-- In the literature of multi-agent bandits, there is a substantial thread of the work that considers communication cost as a metric to improve upon. This work, however, does not touch at all the communication complexity in their study. Can authors comment on whether or not it makes sense to add communication cost analysis to their work?

-- In this work, the authors, by default, assume that the goal of the game is to find the global optimal arm. However, one may suggest that in a multi-agent setting, each agent may care more about their local optimal arm. Can the authors comment on how the algorithms may or may not work when the objective is to find local optimal arms?

**Questions:**

see two questions above.

---

> ### Author Rebuttal · Authors · 2023-08-09
>
> Thank you very much for the review and for providing your suggestions and comments. We would like to present the responses to the questions as follows.
>
> Q1. This work, however, does not touch at all the communication complexity in their study. Can authors comment on whether or not it makes sense to add communication cost analysis to their work?
>
> A1. We will add the analysis on the communication cost to the paper, consistent with the existing literature on multi-agent MAB, and present the analysis as follows (we would also like to refer to A6 in response to the comment Q6 from reviewer a3u4 where we provide elaborations on communication cost.)
>
> Assuming a constant cost of establishing a communication link, as defined in [1], denoted as $c_1$, the communication cost $C_T$ can be calculated as $C_T = c_1 \cdot \sum_{t=1}^T |E_t|$, which is proportional to $\sum_{t=1}^T |E_t|$. Alternatively, following the framework proposed in [2], the communication cost can be defined as the total number of communications among clients, which can be represented as $C_T = \sum_{t=1}^T |E_t|$, similar to the previous definition. Regarding the quantity $C_T = \sum_{t=1}^T |E_t|$, the number of edges $E_t$ could be $O(M)$ for sparse graphs, and at most $O(M^2)$. This implies that the communication cost, in a worst-case scenario, is of order $O(TM^2)$. This cost aligns with the existing work of research on decentralized distributed multi-agent MAB problems without a focus on communication cost, where edge-wise communication is a standard practice.
>
> Recent advancements have been made in reducing communication costs, such as achieving $O(\sqrt{T}M^2)$ in [2] for centralized settings or $O(M^3\log T)$ through Global Information Synchronization (GIS) communication protocols assuming time-invariant graphs in [1]. The incorporation of random graph structures introduces its own complexities, which poses challenges to reductions in communication costs. These advancements introduce a promising direction for communication efficiency within the context of random graphs.
>
> We value and appreciate your perspective on communication cost which enriches the paper and suggests a promising avenue for future research – enhancing communication cost efficiency based on various communication protocols in decentralized distributed multi-agent MAB problems.
>
>
>
>
>
> Q2. The authors, by default, assume that the goal of the game is to find the global optimal arm. Can the authors comment on how the algorithms may or may not work when the objective is to find local optimal arms?
>
> A2. The regret measure is defined with respect to the reward of the globally optimal arm, and the algorithms are designed to minimize this regret. As the regret is proportional to $T$ - the number of pulls of the global optimal arm, a $\log T$ regret implies the identification of the global optimal arm, given that the time horizon $T$ is sufficiently large to ensure $T >> \log T$.
>
> If the objective is to find local optimal arms, locality can be interpreted in two ways: (a) local as in global local arm, or (b) optimal arm with respect to a subset of the nodes, e.g., a node and all adjacent nodes. Regarding the former interpretation, the algorithm guarantees a global optimal solution on average if $T$ is large; it is thus likely to find an optimal arm if $T$ is large. If $T$ is not large, there is no such guarantee and thus the algorithm will identify a good arm within a limited number of pulls. With respect to the latter interpretation, if local is defined as at a node, then this entails executing standard MAB at each node. If locality is defined with respect to 1-neighborhood of a node, then this is the centralized distributed version of MAB. If locality is defined based on some other sets, then the algorithm can be executed recursively on each set of interest. We do not provide an analysis of such a recursive version of the algorithm.
>
> [1] T. Li and L. Song. Privacy-preserving communication-efficient federated multi-armed bandits. IEEE Journal on Selected Areas in Communications, 40(3):773–787, 2022.
>
> [2] C. Li and H. Wang. Communication efficient federated learning for generalized linear bandits. Advances in Neural Information Processing Systems, 35:38411–38423, 2022.

---

> > ### Comment · Reviewer_H8V9 · 2023-08-14
> > **Response**
> >
> > Thanks for your response. It would be great if you could add that communication analysis to the updated version. Also, there are some works in the context of multi-agent bandits with homogeneous rewards that achieve much lower communication costs (see below), where the algorithm design follows some other approaches, e.g., leader-follower, so, considering those might lead to better trade-offs between regret and communication cost.
> >
> > [1] Wang, Po-An, et al. "Optimal algorithms for multiplayer multi-armed bandits." International Conference on Artificial Intelligence and Statistics. PMLR, 2020.
> > [2] Wang, Xuchuang, et al. "Achieving Near-Optimal Individual Regret & Low Communications in Multi-Agent Bandits." The Eleventh International Conference on Learning Representations. 2022.

---

> > > ### Author Response · Authors · 2023-08-17
> > >
> > > Thank you for reviewing the response and sharing the valuable advice and references. We will incorporate the communication cost analysis and the new references into the revised version of the paper, as detailed in the following paragraphs.
> > >
> > > Modifications to the introduction in the paper
> > >
> > > Line 47: In the field of decentralized multi-agent MAB, it is commonly assumed that the mean reward value of an arm for different clients is the same, or equivalently, homogeneous. This assumption can be seen in works such as [..., 1, 2].
> > >
> > > Modifications to Section 3 (including [1, 2]; the references [5, 6] suggested by reviewer NP2Q)
> > >
> > > Communication cost analysis: Assuming a constant cost of establishing a communication link, as defined in [1, 3], denoted as $c_1$, the communication cost $C_T$ can be calculated as $C_T = c_1 \cdot \sum_{t=1}^T |E_t|$, which is proportional to $\sum_{t=1}^T |E_t|$. Alternatively, following the framework proposed in [2, 4, 5, 6], the communication cost can be defined as the total number of communications among clients, which can be represented as $C_T = \sum_{t=1}^T |E_t|$, similar to the previous definition. Regarding the quantity $C_T = \sum_{t=1}^T |E_t|$, the number of edges $E_t$ could be $O(M)$ for sparse graphs, and at most $O(M^2)$. In the random graph model, the expected number of edges is $\frac{M(M-1)}{2}c$, which implies $O(M^2)$ in the worst case scenario and that the total communication cost, in a worst-case scenario, is of order $O(TM^2)$. This analysis holds also for the random connected graph case where $c$ represents the probability of having an edge. This cost aligns with the existing work of research on decentralized distributed multi-agent MAB problems without a focus on communication cost, where edge-wise communication is a standard practice.
> > >
> > > Added text to Conclusions and Future Work
> > >
> > > Recent advancements have been made in reducing communication costs with respect to the dependency in multi-agent MAB with homogeneous rewards (in the generalized linear bandit setting [4], the groundtruth of the unknown parameter is the same for all clients), such as achieving $O(\sqrt{T}M^2)$ in [4] for centralized settings or $O(M^3\log T)$ through Global Information Synchronization (GIS) communication protocols assuming time-invariant graphs in [3]. Likewise, [5, 6] improve the communication cost of order $\log T$ or $o(T)$ through asynchronous communication protocols and balancing the trade-off between regret and communication cost. More recently, [2] establishes a novel communication protocol, TCOM, which is of order $\log \log T$ by means of concentrating communication around sub-optimal arms and performing aggregation of estimators across time steps. Furthermore, [1] develops a new leader-follower communication protocol, which selects a leader that communicates to the followers. Here the  communication cost is independent of $T$ which is much smaller. The incorporation of random graph structures and heterogeneous rewards introduces its own complexities, which poses challenges to reductions in communication costs. These great advancements introduce a promising direction for communication efficiency as a next step within the context herein.
> > >
> > >
> > > References:
> > >
> > > [1] Wang, Po-An, et al. "Optimal algorithms for multiplayer multi-armed bandits." International Conference on Artificial Intelligence and Statistics. PMLR, 2020.
> > >
> > > [2] Wang, Xuchuang, et al. "Achieving Near-Optimal Individual Regret \& Low Communications in Multi-Agent Bandits." The Eleventh International Conference on Learning Representations. 2022.
> > >
> > >
> > > [3] T. Li and L. Song. Privacy-preserving communication-efficient federated multi-armed bandits. IEEE Journal on Selected Areas in Communications, 40(3):773–787, 2022.
> > >
> > > [4] C. Li and H. Wang. Communication efficient federated learning for generalized linear bandits. Advances in Neural Information Processing Systems, 35:38411–38423, 2022.
> > >
> > > [5] Sankararaman et.al. Social Learning in Multi Armed Bandits, SIGMETRICS 2019
> > >
> > > [6] Chawla et.al. The Gossiping Insert-Eliminate Algorithm for multi-agent Bandits, AISTATS 2020

---

### Official Review · Reviewer_zM1d · 2023-07-06

**Soundness:** 3 good
**Presentation:** 3 good
**Contribution:** 3 good
**Rating:** 7
**Confidence:** 4

**Summary:**

Update: I've read the authors' rebuttal as well as all the reviews from other reviewers. The authors' clarification are helpful and mitigate my concerns. I've raised the score from 6 to 7 and I think this manuscript is acceptable for publication.

This paper aims at making contributions to decentralized multi-agent MAB problem research by studying both heterogeneous rewards (with heavy-tailed reward distributions) and time-varying random graphs, where the distributions of rewards and graphs are independent of time. It is claimed that this is the first work to consider this problem and to investigate it with heavy-tailed reward distributions. To be more specific, the paper investigates 1) heterogeneous sub-exponential and sub-gaussian distributed rewards and 2) random graphs including the possibly unconnected E-R model and random connected graphs, and applies them to the decentralized multi-agent MAB framework using a proposed UCB-type solution. It is considered as an extension of relevant existing literature (especially [Dubey et al.,2020] and and [Zhu et al., 2021b]) on decentralized multi-agent MAB  research area.

**Strengths:**

1. Paper presentation is clean and clear, and overall speaking, the manuscript is reader friendly and straightforward to follow.
2. There are quite nontrivial novel theoretical analysis results included in the paper, which seems to be solid.
3. There are quite a couple of helpful remarks/explanations comparing with existing literature.

**Weaknesses:**

1. I'd like to see more more explanations/elaborations about the motivation/practical backgrounds why those types graphs are considered and what are some example real world applications that might have those types of graphs. For example, this paper consider the uniformly distributed connected graph generated using Algorithm 1, but could the author provide any example real world applications that have that kind of graphs? Especially about the part of adding extra minimum number of edges to make the randomly sampled graph to be connected . That seems to be a purely artificial/simulation based approach, and I don't obviously see much real world applications that demonstrate such behavior. The paper introduction briefly mention random failure graph in real world applications as motivation, but obviously random failed graphs have essential difference compared with such formulated graphs. More elaborations/explanations about the motivations/applications would be helpful to convince that such an artificial formulation is not just for convenient mathematical analysis, but also very beneficial to real world applications, rather than just a mathematical game.

2. In algorithm 1, I saw a constant probability "1/2" in the graph sampling part. Why this specific "1/2" is used here? Would any constant probability 0<p<1 work the same here for the analysis and theoretical results (as well as obtained bounds when comparing with literature)? It would be great if the authors could help clarify/confirm this in the manuscript.

3. I saw undirected graph is considered/assumed in this paper. How hard/difficult to consider the more generic scenario of directed graphs and what are the main challenges? It would be valuable to see some discussions on this for future research directions.

4. I'd like to see some scaling and complexity discussions. Is the proposed method naturally scalable to large multi-agent systems? It would be good to see some experiments and analysis on this, ideally comparing with relevant literature.

**Questions:**

Please see my questions in my above comment when taking about the weakness.

**Limitations:**

N/A.

---

> ### Author Rebuttal · Authors · 2023-08-09
>
> Thank you so much for the review and for sharing your comments and perspectives. We would like to include the following perspectives.
>
> Q1. the motivation/practical backgrounds of graphs
>
> A1. We would like to refer to A5 in response to the comment Q5 from reviewer a3u4 where we provide detailed elaboration on the motivations. In what follows, we present the practical applications of the graph under consideration, along with the generation process.
>
> a) uniformly distributed connected graph
>
> Note that in some real-world scenarios, the graph may be changing, yet it can still retain a robust connectivity property. For instance, on a cluster, certain connections between servers might be dynamically disrupted, but the overall cluster of servers remains functional. This means that servers can communicate with each other through multi-step paths, and the way servers are interconnected within the cluster might be time-dependent. This rationale clarifies our choice to sample from the connected graph family, which in line with established literature.  Simultaneously, the uniform distribution aligns with the foundational Erdős-Rényi (E-R) model $G(M,m)$, wherein a graph is uniformly selected from the set of all graphs with $M$ nodes and $m$ edges.
>
> b) adding extra minimum number of edges to make the randomly sampled graph to be connected
>
> We would like to add that this step aims to ensure the connectedness of the initialization graph, denoted as $G_0$, before constructing the subsequent Markov chain within the for loop. Initially, we assess whether $G^{init}$ is already connected; if it is, then there is no need to add any edges. If it is not connected, starting from a vertex we iteratively add edges to the graph, checking its connectivity after each addition. This process continues until the graph becomes connected. By following this approach, we ensure that the complexity of the process remains within $M(M^2+M)$ (the worst case scenario), wherein the first term accounts for the steps taken and the second term accounts for the graph connectivity verification. This method generates an initialized connected graph, denoted as $G_0$, based on $G^{init}$ in polynomial time, preceding the subsequent for loop. While the number of edges added might not necessarily be globally optimal (we are not backtracking), it suffices that it is locally optimal, in order to efficiently obtain a connected graph.
> We can rephrase this as "adding an additional minimal number of edges oriented from a single vertex." In the sensor cluster, we consistently establish connections between a designated sensor and other sensors until the entire cluster becomes connected.
>
> c) random failure graph in real world applications
>
> For a practical application, our objective is to illustrate the professional connection relationships among individuals on social media through a use case. In this context, we establish an edge between two users when they are mutually connected or follow each other on a platform. However, there is a possibility that a user might accidentally tap the "remove connection" or "unfollow" button, leading to the disruption of the connection. Even though these accidental events are infrequent and random, they can still happen. It is important to note that such behaviors are independent among users, and the likelihood of them occurring may be similar for users with similar features. Therefore, we can model this process using an Erdős-Rényi (E-R) model, denoted as $G(M,c)$, where the parameter $c$ signifies the probability of such disconnections not taking place. Essentially, this parameter reflects the likelihood of these accidental edge removals, which is assumed to be quite small in this use case.
>
>
> Q2. In algorithm 1, Why this "1/2" is used here? Would any 0<p<1 work?
>
> We achieve an observation of a graph with a probability of $(\frac{1}{2})^{\frac{M(M-1)}{2}}$ by assigning a probability of $\frac{1}{2}$ to the generation of each edge, considering that we have $\frac{M(M-1)}{2}$ potential edge candidates. Simultaneously, the total number of possible graphs with $M$ nodes is $2^{\frac{M(M-1)}{2}}$. This implies that a uniform distribution over all possible graphs is equivalent to generating each graph observation with a probability of $\frac{1}{2^{\frac{M(M-1)}{2}}}$. As a result, the edge-wise sampling with a probability of $\frac{1}{2}$ creates a uniform distribution over all possible graphs, which is a necessary precondition for simulating the original underlying distribution $P(\cdot)$ constrained within the connected graph family (our intended distribution $f(\cdot) = P(\cdot | G \text{ is connected})$) as described in [1]. The selection of $p = \frac{1}{2}$ transforms the problem of graph sampling based on $P(\cdot)$, with its exponentially large sample space, into a manageable edge-wise sampling task.
>
> The graph generation and the corresponding convergence rate analysis rely on obtaining a sample based on $P(\cdot)$, a condition that is met when selecting $p = \frac{1}{2}$ (if and only if), so for settings $s_2$, $s_3$, $S_2$, and $S_3$, the current analysis and results are valid when specifying $p = \frac{1}{2}$ at the initialization in Algorithm 1. Moreover, it would be promising to explore the modifications needed in Algorithm 1 when $p \neq \frac{1}{2}$ during the initialization step. This could involve adjusting the distribution $p^{\frac{M(M-1)}{2}}$ to match the original underlying distribution $P(\cdot)$ through importance sampling. Additionally, investigating other types of random graphs with non-uniform distributions $p^{\frac{M(M-1)}{2}}$ rather than the uniform distributions $P(\cdot)$ paves the way for future research directions in random graph simulation.
>
> Thank you again for the comments and reviews. The responses to Q3 and Q4 are included in the official comments.
>
> [1] C. Gray, L. Mitchell, and M. Roughan. Generating connected random graphs. Journal of Complex Networks, 7(6):896–912, 2019.

---

> > ### Author Response · Authors · 2023-08-10
> > **Perspectives on directed graphs and complexity analyses**
> >
> > Q3. How hard/difficult to consider the more generic scenario of directed graphs and what are the main challenges?
> >
> > This paper aligns with most existing literature on random graph generation and on decentralized distributed multi-agent MAB. A recent separate line of research focuses on strongly connected graphs.
> >
> > We highlight differences and challenges for directed graphs across key aspects.
> >
> > Graph Generation:
> > In the E-R model, individual directed edge sampling occurs via a Bernoulli distribution with $c$ probability. The sampling frequency adjusts to $M(M-1)$ from $\frac{M(M-1)}{2}$ in undirected graphs.
> >
> > For random connected graphs, Algorithm 1 requires adjustment to account for a uniform distribution over all strongly connected graphs. Edge selection involves sampling all arcs,  followed by verification to ensure the resulting graph is strongly connected. This change preserves prior time complexity. The convergence rate analysis in this strongly connected variant would require additional effort, though the techniques employed should be similar to the current analyses applied to undirected graphs, following Propositions 3 and 4 in Appendix. We can demonstrate that the acceptance ratio $\alpha_t$ of the underlying Metropolis-Hastings sampling remains unchanged from the current version.
> >
> > A challenge lies in establishing a new transition kernel $\pi(\cdot,\cdot)$ for strongly connected graphs, as in Proposition 4. This points to future work, leveraging existing analytical techniques in this work.
> >
> > Main Algorithm:
> > Asymmetry in empirical matrices $P_t$ and $P^{\prime}_t$ necessitates modifications. Updates for client $m$ switch to ${P_t(j,m)}$ and ${P^{\prime}_t(j,m)}$ from ${P_t(m,j)}$ and ${P^{\prime}_t(m,j)}$. Network-wide estimators $\Tilde{\mu}$ include arcs, involving arc-related substitutions.
> >
> > Integrating asymmetrical information into the algorithm is a challenge that can enhance consensus for strongly connected graphs.
> >
> > Regret Analysis:
> > Taking Theorem 2 as an example, achieving a regret bound of $O(\log T)$ requires: 1) empirical matrix convergence, 2) client consensus: consensus on the number of pulled arms, and 3) concentration inequalities for network-wide estimators. Step 2 involves defining sufficient conditions on strong connectivity in the context of the E-R model. The remaining steps align with the current analysis framework.
> >
> > Q4. I'd like to see some scaling and complexity discussions. Is the proposed method naturally scalable to large multi-agent systems? It would be good to see some experiments and analysis on this, ideally comparing with relevant literature.
> >
> > A4. The method is theoretically suitable for multi-agent systems of any size due to universal regret bounds regardless of $M$. For Erdős-Rényi (E-R) graphs, time complexity is $O(M^2)$, but independent edge sampling allows parallelization for larger systems, notably accelerating the process. For random connected graphs, initialization complexity is $O(M^3)$ (discussed in A1), but its impact on overall efficiency is limited given the time horizon's context.
> >
> > At each step, graph generation's time complexity is $O(M^2 + M + |E_t|) \leq O(M^2)$ where the first term accounts for edge selection, and the second and third terms are for graph connectivity verification using Algorithm 1 (benefitting from the use of Markov chains for graph generation). The main algorithm's stages result in $O(M^2 + MK)$ overall complexity, where the $MK$ complexity results from the stages like UCB (Upper Confidence Bound) computations and estimator updates, while $M^2$ arises from transmissions and graph generation.
> >
> > [Zhu et al., 2021b] proposes $Gossip_{UCB}$ method for decentralized multi-agent MAB with time-invariant connected graphs which exhibits a time complexity of $O(MK + M^2)$. Our method is consistent with this work, and beyond that, we account for more complicated random graphs, which surprisingly does not bring an additional time complexity ($M^3$ in the initialization step in the worst-case scenario to obtain a connected graph is also applicable to this work). The $Decentralized , MP-UCB$ algorithm designed for time-invariant connected graphs and homogeneous heavy-tailed rewards in [Dubey et al.,2020] has $O(MK + M^2)$ complexity per step, with initialization at $O(3^{\frac{M}{3}})$. Ours retains complexity order, with significantly lower initialization and extension to random graphs.
> >
> > Most procedures operate synchronously, except random connected graph generation ($O(M^2)$).
> > Our approach balances parallelism and random connected graph generation, yielding reasonable $O(M^2)$ complexity for exponentially large sample spaces.
> >
> > Space complexities align with existing methods at $O(M^2 +MK)$ due to estimator storage.
> >
> > Regarding experimentation, while we were not able to complete the experiments within the response period, we plan numerical studies to assess runtime and performance against existing methods.

---

> ### Comment · Area_Chair_abKL · 2023-08-18
>
> Dear Reviewer,
>
> Please reply to the rebuttal and indicate whether it clears your concerns, or at least acknowledge whether you have read the response. This is important to the authors.
>
> Thanks,
>
> Your Area Chair

---

> > ### Comment · Reviewer_zM1d · 2023-08-18
> >
> > I've read the authors' rebuttal as well as all the reviews from other reviewers. The authors' clarification are helpful and mitigate my concerns. I've raised the score from 6 to 7 and I think this manuscript is acceptable for publication.

---

### Official Review · Reviewer_3jx9 · 2023-07-07

**Soundness:** 3 good
**Presentation:** 3 good
**Contribution:** 4 excellent
**Rating:** 7
**Confidence:** 4

**Summary:**

This paper presents a novel algorithm for solving decentralized multi-agent multi-armed bandit problems for two graph dynamics under both sub-exponential and sub-gaussian reward distributions. Their whole algorithm framework can be divided into a Burn-in period and a Learning period, ensuring a process warm-up phase before getting UCB-type estimators. The paper includes detailed theorems and proofs in both the main manuscript and appendix with regret analysis.

**Strengths:**

1. Authors fill in the gap in the multi-agent multi-armed bandit problem with the intersection between heavy-tailed reward distributions and heterogeneous settings with time-varying graphs.
2. The paper is clearly written and easy to follow.


**Weaknesses:**

1. The paper is full of details but without a conclusion to re-emphasize the main statement and contribution of this paper.
2. Please refer to the questions


**Questions:**

1. In line 368, is there any typo mentioning the same conditions as in Theorem 2 and $\textbf{4}$?
2. In line 312, the 3rd and 4th terms are exactly the same. Is there any typo?
3. Could you please elaborate more about how you get L1 in theorem 4?

---

> ### Author Rebuttal · Authors · 2023-08-09
>
> Thank you very much for the review and providing the suggestions and insights. We would like to add responses as follows.
>
> Q1. The paper is full of details but without a conclusion
>
> A1. We will incorporate a concluding section into the paper, as shown in the official comment.
>
> Q2. In line 368, is there any typo mentioning the same conditions as in Theorem 2 and 4?
>
> A2. Thank you for bringing this to our attention, and we apologize for the typo. The correct phrase should be "Assume the same conditions as Theorem 2 and 3," where Theorem 2 and Theorem 3 refer to settings $s_1,s_2,s_3$ and $S_1, S_2, S_3$, respectively. This has been revised in the revised paper.
>
> Q3. In line 312, the 3rd and 4th terms are exactly the same. Is there any typo?
>
> A3. The 3rd and 4th terms are the same. This term is due to the convergence rate of the simulated graph based on Algorithm 1 and of the empirical matrix $P_t$, given the uniform random connected graph that has a dependency on the spectral gap $\lambda$. Since both setting $s_2$ and $s_3$ are under the uniform random connected graph assumption generated by Algorithm 1, and they only differ in the number of clients $M$, this term appears twice in the length of the burn-in period.  One alternative approach is to retain only one instance of this term in the expression for $L$, written as $L = \max\{\frac{{}\ln{\frac{2T}{\epsilon}}}{2\delta^2}, \frac{4K\log_{2}T}{c_0}, \frac{\ln{\frac{\delta}{10}}}{\ln{p^*}} +25\frac{1+\lambda}{1-\lambda}\frac{\ln{\frac{2T}{\epsilon}}}{2\delta^2}, \frac{K\ln(\frac{MT}{\epsilon})}{c_0\ln(\frac{1}{1-\frac{2\log M}{M-1}})}\}$.  An alternative expression for $L$  keeping both terms is presented in the official comment.
>
> Thank you again for the point. We hope this could provide a more detailed explanation of the rationale behind the choice of $L$.
>
> Q4. Could you please elaborate more about how you get L1 in theorem 4?
>
> A4. A sketch is as follows. First and foremost, we would like to add that the upper bound on $L_1$ is $\max(L, K \cdot 2(K^2+MK))$ based on the proof steps outlined in Appendix, which aligns with the concept of instance-free regret bounds. This upper bound does not rely on the $\frac{1}{\Delta_i}$ factor that could be problematic with small $\Delta_i$. The terms $[\frac{16C_1\log T}{\Delta_i^2}]$ and $[\frac{4C_2\log T}{\Delta_i}]$ are not utilized and $L_1$ should be $\max(L, K \cdot 2(K^2+MK))$ in the statement. We will update this term in the statement of Theorem 4.
>
> The proof sketch of Theorem 4 is as follows. We directly decompose $E[R_T|A]$ based on any $L_1$ as $E[R_T|A] = L_1 + \frac{1}{M}\sum_{t=L_1}^T\sum_{m=1}^M(E[\mu_{i^*} - U_m^t(i^*)|A] + E[U_m^t(i^*) - U_m^t(a_m^t)|A] +  E[U_m^t(a_m^t) - L_m^t(a_m^t) |A] + E[L_m^t(a_m^t) - \mu_{a_m^t}^t|A])$ where $U_m^t(a_m^t), L_m^t(a_m^t)$ are the upper and lower confidence bounds of the network-wide estimators. This is the sum of (i) the differences between the confidence bounds and the true mean values, (ii) the differences in the upper confidence bounds between the global optimal and sub-optimal arms and (iii) the differences between the upper and lower confidence bounds $\sum_t U_m^t(a_m^t) - L_m^t(a_m^t)$ which is the dominating term. The term (i) relies on the concentration inequalities for the network-wide estimators. The second term (ii) is based on the number of time steps when the clients do not follow UCB, which is relevant to the criterion $n_{m,i}(t) \leq N_{m,i}(t) - K$. The third term (iii) is proportional to the cumulative sum of $\sqrt{\frac{C_1\ln{T}}{n_{m,i}(t)}} + \frac{C_2\ln{T}}{n_{m,i}(t)}$ which has an upper bound $O(\sqrt{T}\log T)$, i.e. $\sum_t U_m^t(a_m^t) - L_m^t(a_m^t) \propto \sum_t\sqrt{\frac{C_1\ln{T}}{n_{m,i}(t)}} + \frac{C_2\ln{T}}{n_{m,i}(t)} = O(\sqrt{T}\log T)$ by the Harmonic series and the Cauchy Schwartz's inequality. The choice of $L_1$ is a necessary condition for the concentration inequalities in (i).
>
> The first term, $L$, in the expression for $L_1$ originates from the definition of event $A$, where $t > L$ is assumed with respect to the graph's randomness in the context of considering the regret $E[R_T|A]$ conditioned on $A$. The event $A$ becomes crucial for the concentration inequalities of the network-wide estimators (i), and the rationale behind introducing term $L$ is discussed in A3.
>
> The second term, $L_1 \geq K \cdot 2(K^2+MK)$, is necessitated by the condition that, for $t \geq L_1$, $n_{m,i}(t) \geq n_{m,i}(L_1) \geq \frac{L_1}{K} \geq 2(K^2+MK)$. This condition ensures the validity of the subsequent lemma in Appendix: For any $m,i$, if $n_{m,i}(t) \geq 2(K^2+2KM)$ and graph $G_t$ is connected, then we have $n_{m,i}(t) \leq 2\min_{j}n_{j,i}(t)$
> where the min is taken over all the clients. This lemma guarantees that the clients would stay on the same page (in terms of the variance proxies of the network-wide estimators), which is important to the concentration inequalities of network-wide estimators (i).
>
> This lemma is equally essential for both Theorem 2 and Theorem 3. The rationale behind the absence of the condition $L \geq K \cdot 2(K^2+MK)$ in Theorem 2 and Theorem 3 lies in the different analytical strategies. These two theorems rely on an alternative decomposition, namely $E[R_T|A] \propto  L + \sum_i\Delta_iE[n_{m,i}(T)| A]$, which enables us to achieve instance-dependent results. The absence of the condition $L \geq K \cdot 2(K^2+MK)$ in Theorem 2 and Theorem 3 arises from bounding the terms based on this alternative decomposition. The condition $n_{m,i}(t) \geq 2(K^2+MK)$ is incorporated into the upper bound of the second term $E[n_{m,i}(T)| A]$. This upper bound reads as
> $\max({[\frac{4C_1\log T}{\Delta_i^2}], 2(K^2+MK) })$ and $\max([\frac{16C_1\log T}{\Delta_i^2}], [\frac{4C_2\log T}{\Delta_i}], 2(K^2+MK))$ in Theorem 2 and Theorem 3, respectively.
>
> We hope the discussions on $L_1$ could elaborate the derivation of $L_1$ in Theorem 4 and illustrate why it is unique to this particular theorem.

---

> > ### Author Response · Authors · 2023-08-10
> > **Perspectives on the concluding section and the alternative of L**
> >
> > Concluding section:
> >
> > In this paper, we consider a decentralized multi-agent multi-armed bandit problem in a fully stochastic environment that generates time-varying random graphs and heterogeneous rewards following sub-gaussian and sub-exponential distributions, which has not yet been studied in existing works. To the best of our knowledge, this is the first theoretical work on random graphs including the E-R model and random connected graphs, and the first work on heterogeneous rewards with heavy tailed rewards. To tackle this problem, we develop a series of new algorithms, which first simulate graphs of interest, then run a warm-up phase to handle graph dynamics and initialization, and proceed to the learning period using a combination of upper confidence bounds (UCB) with a consensus mechanism that relies on newly proposed weight matrices and updates, and using a stopping time to handle randomly delayed feedback. Our technical novelties in the results and the analyses are as follows. We prove high probability instance-dependent regret bounds at the order of $\log T$ in both sub-gaussian and sub-exponential cases, consistent with the regret bound in the existing works that only consider the expected regret. Moreover, we establish a nearly tight instance-free regret bound of order $\sqrt{T}\log T$ for both sub-exponential and sub-gaussian distributions, up to a $\log T$ factor. We leverage probabilistic graphical methods on random graphs and draw on theories related to rapidly mixing Markov chains, which allows us to eliminate the doubly stochasticity assumption through new weight matrices and a stopping time. We construct new network-wide estimators and invent new concentration inequalities for them, and subsequently incorporate the seminal UCB algorithm into this distributed setting.
> >
> >
> > Alternative of $L$:
> >
> > Alternatively, we may rephrase the expression for $L$ as follows - by keeping the repeated term ($\frac{4K\log_{2}T}{c_0}$ which is another repetitive term that was omitted in the earlier expression; it is associated with the edge probability $c$, which defines the gap between consecutive communications and remains consistent across both $s_1$ and $s_2$).
> > $$
> >  L  = \max{L_{s_1},L_{s_2},L_{s_3}}
> > $$
> > where $L_{s_1} = \max(\frac{{}\ln{\frac{2T}{\epsilon}}}{2\delta^2}, \frac{4K\log_{2}T}{c_0})$,
> >
> > $L_{s_2} = \max(\frac{\ln{\frac{\delta}{10}}}{\ln{p^*}} +25\frac{1+\lambda}{1-\lambda}\frac{\ln{\frac{2T}{\epsilon}}}{2\delta^2}, \frac{4K\log_{2}T}{c_0})$,
> >
> > and $L_{s_3} = \max(\frac{\ln{\frac{\delta}{10}}}{\ln{p^*}} +25\frac{1+\lambda}{1-\lambda}\frac{\ln{\frac{2T}{\epsilon}}}{2\delta^2}, \frac{K\ln(\frac{MT}{\epsilon})}{c_0\ln(\frac{1}{1-\frac{2\log M}{M-1}})})$ are the length of burn-in period of setting $s_1, s_2$ and $s_3$, respectively. There are two terms we have not introduced, $\frac{{}\ln{\frac{2T}{\epsilon}}}{2\delta^2}$ and $\frac{K\ln(\frac{MT}{\epsilon})}{c_0\ln(\frac{1}{1-\frac{2\log M}{M-1}})}$ where the first is related to the concentration inequality of the empirical matrix $P_t$ in the E-R model and the second term is related to the gap between two consecutive communications when $c$ is small in setting $s_3$ and $S_3$ where $M > 11$.

---

> > > ### Comment · Reviewer_3jx9 · 2023-08-15
> > >
> > > Thank you for your response and clarification. The conclusion section will definitely make the work more complete.

---

> > > > ### Author Response · Authors · 2023-08-17
> > > >
> > > > Thank you for reviewing the responses and providing valuable suggestions and comments. We will incorporate the concluding section verbatim into the revised paper.

---

### Official Review · Reviewer_NP2Q · 2023-07-10

**Soundness:** 3 good
**Presentation:** 3 good
**Contribution:** 3 good
**Rating:** 7
**Confidence:** 3

**Summary:**

The paper studies stochastic multi-armed bandit over a network of agents communicating locally. Specifically, for each of the K arms, each agent has an associated underlying arm-mean. The agents can only communicate with their neighbors in each round to exchange information. The goal of the agents is to pull arms so that the regret with respect to the average arm-mean averaged over the agents is minimized. This is a challenging problem since the agents can only infer their arm mean from the observed rewards and must rely on information sharing to learn the average of the arm means averaged over the network.

In order to model information sharing, the paper assumes that the communication graph over agents is a stochastic process that possibly changes over time. This allows for the information about an arm collected by an agent to percolate over the network so that all agents can potentially learn of the mean of an arm averaged over agents. However, the agents never have full information on the graph connectivity and at each time, can only observe who their neighbors are.

In this scenario, the paper proposes an algorithm based on initial coordinated exploration and information sharing followed by UCB based on estimated network sample average over arms. The proposed algorithm is elegant and practical.

**Strengths:**

A clear articulation of the problem and an elegant algorithm to solve it. Good analysis of the algorithm so that a reader can follow along. High probability bounds that do not average over the randomness in the graph generation process that is typical of results in this area.

**Weaknesses:**

Lack of experimental evidence. Given one of the core contributions of the paper is an elegant algorithm, presenting synthetic simulations will substantiate the contributions of the paper.

**Questions:**

Can the authors substantiate the insights using synthetic simulations? For example, it will be nice if the authors can

-  "visualize" the flow of information in the algorithm,
-  the impact of heterogeneity to the resulting regret
-  Impact of the graph generation process on regret. For example complete graphs is the easiest while a ring graph is the hardest? Perhaps the graph spectrum plays a role in the regret?

Of-course these are a lot of changes, but will appreciate intuition/insights on these questions and will make the paper more complete.

***EDIT Post Rebuttal:***
--
 I thank the authors for the thorough response and would encourage them to add it to the camera ready. I would further propose to add two related references to the additional list of references.

Sankararaman et.al. Social Learning in Multi Armed Bandits, SIGMETRICS 2019

Chawla et.al. The Gossiping Insert-Eliminate Algorithm for multi-agent Bandits, AISTATS 2020

---

> ### Author Rebuttal · Authors · 2023-08-09
>
> Thank you so much for reviewing the paper and for sharing the suggestions and perspectives. We would like to add some responses as follows.
>
> Q1. Lack of experimental evidence.
>
> A1. We will add an experimental illustration of the proposed algorithms using synthetic simulations to the paper, and compare these algorithms to existing ones as benchmarks. While we were not be able to conduct a comprehensive experiment during the response period, we present the details which we will use for running the simulation experiments. The official comment below the rebuttal includes details on data generation, models, benchmarks [1,2], and model evaluation.
>
> Q2. Can the authors substantiate the insights using synthetic simulations? For example,
>
> a) it will be nice if the authors can "visualize" the flow of information in the algorithm
>
> A. We will add a flowchart to the paper to visualize the flow of the algorithm.
>
> b) the impact of heterogeneity to the resulting regret
>
> A. Indeed, heterogeneity does impact the regret, and it is a great point to examine the degree of this impact through numerical experiments.
>
> Theoretically, the current methods and regret bounds are applicable to both homogeneous and heterogeneous settings. However, algorithms developed for homogeneous settings may not be effective in heterogeneous settings. Developing solutions that provide guarantees for heterogeneous settings is one of our motivations. It is true that smaller regret is possible in homogeneous settings, and the regret bounds do not reflect such differences between the two settings. This necessitates a direct comparison of exact regrets through numerical experiments.
>
> We introduce a variable denoted as $h$, which refers to the discrepancy, representing the heterogeneity of the problem setting and defined as $h = \max_{m,j,i}|\mu_i^m - \mu_i^j|$. During the generation of synthetic datasets, we generate various settings with different discrepancies, such as $h = 0, 0.1, 0.2, \ldots, 1$, while keeping all other parameters/settings constant. Subsequently, we execute the algorithm across these datasets and compare the regrets among different settings.
>
> c) Impact of the graph generation process on regret. For example complete graphs is the easiest while a ring graph is the hardest? Perhaps the graph spectrum plays a role in the regret?
>
> A. We appreciate the perspective on the dependency of regret on graph generation. In deterministic graph scenarios, [3] investigates regret under various graph structures, including star, circular ring, complete, and d-regular graphs. This study demonstrates that the circular ring graph presents the most challenging scenario with the largest regret, while the complete graph is the simplest. Notably, their regret bound is influenced by the graph's diameter and thus by the graph's spectrum.
>
> In our cases involving random graphs generated by MCMC which might have different spectrum at different time steps, the regret bounds exhibit a dependence on the spectrum of the associated Markov chain, denoted as $\lambda$ depending on the targeted distribution $f$ of the graphs. In our study, we focus on random connected graphs, where $f(\cdot) = P(\cdot | G \text{ is connected})$. This relationship can be expressed as $1 - \lambda  \geq \frac{1}{2\frac{\ln{2}}{\ln{2p^*}}\ln{4} + 1}$, where $p^* = p^*(M) < 1$ is specific to the distribution $f$.
>
> Numerical experiments could establish such dependencies for different graphs besides the studied graphs here. To observe the algorithm's regret given complete graphs, we set the parameter $c$ in the E-R model $G(M,c)$ to $1$, with the regret being computed at the end of the game. If we consider circular ring graphs or star graphs, generating the graph becomes feasible in polynomial time due to the small sample space ($O(M^2)$ or $O(M)$). Following this, experiments could be conducted using these graphs in combination with our main algorithm, allowing us to examine the resulting regret accordingly. For the case of ring graphs, a family of random ring graphs might be established by adapting Algorithm 1 [4]. In this case, the target distribution would be $f(\cdot) = P(\cdot | G \text{ is a connected ring graph})$. Developing an MCMC simulation method for generating graphs based on this distribution, as well as for generating other types of graphs like d-regular graphs, presents a potential direction for future exploration. One conjecture is to initialize the distribution as $P(|G \text{ is a ring})$, which requires knowledge of how to randomly generate a ring graph. Following this, the MCMC method outlined in Algorithm 1 could be employed.
>
> Thank you once again for the suggestions and perspectives on the numerical experiments. We hope the above illustrations could be informative. We will add the numerical experiments to the paper to establish the tightness of the dependencies of the regret bounds on the various aspects of the settings.
>
> [1] A. Dubey and A. Pentland. Cooperative multi-agent bandits with heavy tails. In International Conference on Machine Learning, 2730–2739, 2020.
>
> [2] Z. Zhu, J. Zhu, J. Liu, and Y. Liu. Federated bandit: A gossiping approach. In Abstract Proceedings of the 2021 ACM SIGMETRICS/International Conference on Measurement and Modeling of Computer Systems, 3–4, 2021.
>
> [3]  T. Li and L. Song. Privacy-preserving communication-efficient federated multi-armed bandits.466
> IEEE Journal on Selected Areas in Communications, 40(3):773–787, 2022
>
> [4] I. Gitler, E. Reyes, and R. H. Villarreal. Ring graphs and complete intersection toric ideals. Discrete mathematics, 310(3):430–441, 2010.

---

> > ### Author Response · Authors · 2023-08-10
> > **Perspectives on numerical experiment details**
> >
> > Numerical Experiment Details
> >
> > Dataset generation: The process of data generation involves both reward generation and graph generation. First we generate different numbers of arms and clients, denoted as $K$ and $M$, respectively. Specifically, we generate rewards using both sub-gaussian and sub-exponential distributions, varying the mean values $\mu_i^m$ by random sampling within $[0,1]$ and introducing multiple levels of heterogeneity denoted as $h = \max_{i,j,m}|\mu_i^m-\mu_j|$. Additionally, we include multiple variance (proxy) $\sigma^2$ of the distributions by again random sampling. In terms of graph generation, we generate E-R models with varying values of $c$, and we generate random connected graphs using Algorithm 1. Furthermore, as elaborated in A2, we would consider the generation of random ring graphs or star graphs.
> >
> > Models and benchmarks: We will implement the proposed algorithms. We consider \citep{dubey2020cooperative}
> > as the benchmark, since the work assumes heavy-tailed rewards and deterministic graphs in the analysis, and additionally it conducts experiments with a fixed E-R model.  Meanwhile, we also compare the new method with~\citep{zhu2021federated} which focuses on deterministic graphs and sub-gaussian rewards and motivates our work.
> >
> > Model evaluation:
> > The evaluation metric is the regret measure as defined in the paper. An additional possibility is to include communication cost as another performance measure, based on the reviewers' suggestions and existing literature. This will help us understand the trade-off between regret and communication cost under different models.

---

> > > ### Comment · Reviewer_NP2Q · 2023-08-11
> > > **Thank you for the response**
> > >
> > >  I thank the authors for the thorough response and would encourage them to add it to the camera ready. I would further propose to add two related references to the additional list of references.
> > >
> > > Sankararaman et.al. Social Learning in Multi Armed Bandits, SIGMETRICS 2019
> > >
> > > Chawla et.al. The Gossiping Insert-Eliminate Algorithm for multi-agent Bandits, AISTATS 2020

---

> > > > ### Author Response · Authors · 2023-08-17
> > > >
> > > > Thank you for reviewing the responses and for sharing the valuable suggestions and references. We will incorporate the discussions and references [5, 6] into the paper and benchmarks, as elaborated in the following paragraphs verbatim. The works [5, 6], in addition to establishing the elegant algorithms and optimal regret bounds that explicitly characterize the dependency of regret bounds on the conductance of the graph,  also pave the way for integrating communication efficiency and privacy guarantee optimization into our settings with random graphs and heterogeneous rewards. The characterization of the trade-off between regret and communication cost in the regret bound theoretically demonstrates the improvement resulting from replacing ring graphs with complete graphs. The techniques include asynchronous communication protocols and communication cost optimization, in line with both federated learning (the number of transmitted estimators is $O(M)$ or $\log (MK)$, instead of the common $O(MK)$) and decentralized multi-agent MAB (the number of communication rounds having a budget of $O(\log T)$ or $o(T)$, instead of $O(T)$). We appreciate the references and have added them to the communication efficiency discussion that is also suggested by reviewer a3u4 and reviewer H8V9. In the meantime, transmitting arm selections, rather than reward estimators based on historical arm pulls, significantly enhances client privacy.
> > > >
> > > > Modifications to Introduction
> > > >
> > > > Line 47: In the field of decentralized multi-agent MAB, it is commonly assumed that the mean reward value of an arm for different clients is the same, or equivalently, homogeneous. This assumption can be seen in works such as [..., 3, 5, 6].
> > > >
> > > > Line 66: Besides rewards, the underlying graph assumptions are essential to the decentralized multi-agent MAB problem, as increased communication among clients leads to better identification of global optimal arms and smaller regret. The existing works [5, 6] relate regret with graph structures and characterize the dependency of regret on the graph complexity with respect to conductance. When considering two special cases, [6] demonstrates the theoretical improvement achieved by circular ring graphs compared to complete graphs, and [3] numerically shows that the circular ring graph presents the most challenging scenario with the largest regret, while the complete graph is the simplest.
> > > >
> > > > Modifications to Remark (Specification of the parameters)
> > > >
> > > > Line 335: We observe that the regret bound is dependent on the transition kernel $\pi$ and the spectral gap $\lambda$ of the underlying Markov chain associated with $\pi$. This indicates the significance of graph complexities and distributions within the framework of the random graph model when deriving the regret bounds, in a similar manner as the role of graph conductance in the regret bounds established in [5, 6] for time-invariant graphs.
> > > >
> > > > Modifications to our experimental details
> > > >
> > > > Models and benchmarks: We will implement the proposed algorithms. In term of heavy-tailed rewards, we will consider [1]
> > > > as the benchmark, since the work assumes heavy-tailed rewards and deterministic graphs in the analysis, and additionally it conducts experiments with a fixed E-R model. Meanwhile, we will also compare the new method with [2] which focuses on deterministic graphs and sub-gaussian rewards and motivates our work. We will also include [5, 6] as benchmarks, where the regret bounds in [5, 6] fully characterize the dependencies on the sub-optimality gap and the graph complexity in terms of conductance. These works and [3] also consider communication efficiency and privacy, and the experiments will also cover complete graph scenarios.
> > > >
> > > > Variables in the settings: We will explore various settings to analyze the tightness of the regret bounds' dependencies on a range of factors. These factors include the degree of setting heterogeneity ($h$), the distributions of graphs with different complexities ($f$), the number of clients ($M$), the number of arms ($K$), and the reward distributions ($\sigma^2$, $\mu$, and $s$), where $s$ denotes whether the distribution is light-tailed or heavy-tailed.
> > > >
> > > > Evaluation:
> > > > The evaluation metric will be the regret measure as defined in the paper. We will also include the communication cost as another performance measure, which is thoroughly considered in [3, 5, 6], based on the reviewers' suggestions and existing literature. This will help us understand the trade-off between regret and communication cost under different models. Meanwhile, the privacy guarantee will also be regarded as an evaluation metric, as optimized in [5, 6], following the privacy measure defined in [3] or a newly defined measure. Additionally, runtime can provide insights into the time complexity of the models.
> > > >
> > > > [5] Sankararaman et.al. Social Learning in Multi Armed Bandits, SIGMETRICS 2019
> > > >
> > > > [6] Chawla et.al. The Gossiping Insert-Eliminate Algorithm for multi-agent Bandits, AISTATS 2020

---

### Official Review · Reviewer_a3u4 · 2023-07-24

**Soundness:** 4 excellent
**Presentation:** 3 good
**Contribution:** 4 excellent
**Rating:** 7
**Confidence:** 2

**Summary:**

This paper studies a multiplayer graph bandit setting. In particular, they consider  heterogenous rewards in which the mean rewards for each agent are different, and are allowed to be both sub-gaussian and sub-exponential. They also consider random graphs between agents, which dictates which agents are allowed to communicate their rewards with each other. This graph structure is allowed to vary with time. They consider two types of graph generations as well as two types fo reward families for a total of 4 settings.

They propose an algorithm with the following components (1) graph generation, (2) DrFed-UCB (burn in period), and (3) DrFed-UCB (learning period).

Their regret analysis achieves optimal instance dependent regret on the order of log T with high probability.

**Strengths:**

This paper considers a setting that is very general and interesting as well. For instance rewards can vary as well as the communication graphs which are generally assumed to be fixed. Furthermore, they allow for heavy tailed reward dsitributions where as standard bandit literature only considers subgaussian rewards. Their results seem to be very strong for this rather general setting.

**Weaknesses:**

 I think the organization of the settings as well as the key results can be better (Maybe each setting gets its own theorem latex environment). Moreover, I'd use section 3.2 to list the regret bounds, and outline the analysis in a section following that (theorem 1 can go in that section). Kind of related to this point I think the statements of theorem 2 , theorem 3, theorem 4 are too detailed. Maybe just state the regret bound and define the constants in paragraph below it or in the proof.

**Questions:**

I'm not sure I understand S_1, S_2, and S_3. This is described in the first paragraph of sction 3.2 and it's worded quite confusingly. for isntance I'm not sure what M is referring to in describing Seettings 1.1, 1.2, 2.1, and 2.2. Also ,whcih setting is theorem 4 for? S_1, S_2 or S_3?

**Limitations:**

The graph generation is limited to two distributions (ER and uniform). Furthermore, their algorithm can be costly in communication.

---

> ### Author Rebuttal · Authors · 2023-08-09
>
> Thank you very much for reviewing the paper and providing the detailed comments and suggestions. For the questions and limitations, we would like to include some responses as follows.
>
> Q1. I think the organization of the settings as well as the key results can be better (Maybe each setting gets its own theorem latex environment).
>
> A1. We have improved the presentation of these settings in the revised paper, as detailed in Table 1 in A4, to provide a clearer structure for the setting information. In addition, we will use different latex environments for the results corresponding to different settings.
>
> Q2. Moreover, I'd use section 3.2 to list the regret bounds, and outline the analysis in a section following that (theorem 1 can go in that section).
>
> A2. We will include a summary of instance-dependent and instance-free regret bounds in settings $s_1,s_2,s_3,S_1,S_2,S_3$ in Section 3.2,  and put the formal statements of theorems and the analyses to Section 3.3.
>
> Q3. I think the statements of theorem 2, theorem 3, theorem 4 are too detailed.
>
> A3. In the new Section 3.3, we will revise Theorems 2, 3, and 4 by providing a concise version that emphasizes the order of the regret bound. Additionally, we will include remarks immediately after the theorems, paraphrasing the details about the constants and parameters, to enhance the clarity and presentation of the results.
>
> Q4. $S_1, S_2$, and $S_3$ the first paragraph of section 3.2; M; Theorem 4
>
> A4. We apologize for the unclear presentation and will include a table in the paper, which is shown in the uploaded pdf, to outline the notations for each setting (Setting 1.1, 1.2, 2.1, 2.2; $s_1, s_2, s_3, S_1,S_2,S_3$). When defining Setting 1.1, 1.2, 2.1 and 2.2, we are considering the possible combinations of graphs and reward assumptions for any $M$. If we further impose constraints on $M$ within setting 1.2 to be either $[1,10]$ or $[11, \infty)$, we obtain setting $s_2$ and $s_3$ respectively. Likewise, we define $S_2$ and $S_3$ by applying the same $M$ conditions in setting 2.2.
>
> Theorem 4 holds for setting $S_1,S_2$ and $S_3$ with sub-exponential rewards, as well as $s_1,s_2$ and $s_3$ with sub-gaussian rewards. We apologize for the typo in the sentence "Assume the same conditions as in Theorem 2 and 4," which should read "Assume the same conditions as in Theorem 2 and 3."  In Theorem 2, we address settings $s_1, s_2$, and $s_3$, and in Theorem 3, we address settings $S_1, S_2, S_3.$ We will include these specific setting specifications in the revised statement of Theorem 4.
>
> Q5. The graph generation is limited to two distributions (ER and uniform).
>
> A5. Thank you for your perspective and suggestion regarding the graph assumptions. We would like to add that this paper considers these two types of random graphs for the following reasons, both from the edge and the graph perspective.
>
> The E-R model, comprising $G(M,c)$ and $G(M,m)$, is a seminal random graph model. Particularly, $G(M,c)$ is commonly used for subsets of graph edges, as in mean field games, majority voting, and multi-agent MAB experiments [1] where it's fixed after generation. This inspires its integration into the decentralized distributed multi-agent MAB problem. While other models like edge randomness [2] exist, they are less prevalent in the literature.
>
> On the other hand, the E-R model $G(M,m)$ generates graphs uniformly with $M$ nodes and $m$ edges. In decentralized multi-agent MAB problems, connectivity is often assumed for obtaining global information in finite time. This motivates us to explore a uniform distribution in the connected graph family. We adapt MCMC techniques where matching and coupling are well cited applications [3], from graph generation, inspired by [4], to generate uniformly distributed connected graphs, providing the very first finite-time convergence rate analysis. This offers insights into the domain of random graph generation.
>
> We appreciate the comment on the possibilities of other random graphs, which inspires us to consider other distributions besides uniform distributions, random weighted graphs [2] and waxman random graphs [5], as future directions.
>
> Q6. Furthermore, their algorithm can be costly in communication.
>
> A6. In federated learning, research has explored communication costs defined as proportional to model sizes [6]. In the context of multi-agent MAB, we may consider defining costs as the number of estimators, yielding $O(MK)$ due to $M$ clients each maintaining $K$ estimators. Existing work on decentralized multi-agent MAB [1,2] has exhibited the same order, although communication costs have not been addressed.
>
> Alternatively, for multi-agent MAB with communication cost, cumulative building costs have been defined [7], where each edge incurs a constant cost. This implies communication cost is proportional to the number of edges or communication rounds. In our random graph model, the expected edge count is $\frac{M(M-1)}{2}c$, implying worst-case $O(M^2)$. Comparatively, this cost aligns with $O(M^3\frac{\log T}{T})$ in [7]; it's more favorable for $M\log T \leq T$ (e.g., small $M$ or large $T$), and less favorable otherwise (e.g., $M = O(T)$). Our algorithm's communication cost is similar to that in [7] for small $M$ or large $T$. Despite having chances of a larger communication cost, we extend the time-invariant connected graph in [7] to time-dependent random graphs.
>
> The GIS protocol in [7] indicates potential to enhance efficiency ($O(M^3\frac{\log T}{T})$), particularly with small $M$ or large $T$. We appreciate your comment and for raising this important aspect for further consideration.
>
> From a time perspective, parallel synchronization can optimize communication. Clients could independently communicate information they possess ((although this might raise concerns about channel congestion if only one communication channel is available per client). This strategy could accelerate communication.

---

> > ### Author Response · Authors · 2023-08-10
> >
> > References:
> >
> > [1] A. Dubey and A. Pentland. Cooperative multi-agent bandits with heavy tails. In International Conference on Machine Learning, 2730–2739, 2020.
> >
> > [2] A. Frieze and M. Karo ́nski. Introduction to random graphs. Cambridge University Press, 2016.
> >
> > [3] M. Jerrum and A. Sinclair. The markov chain monte carlo method: an approach to approximate counting and integration. Approximation algorithms for NP-hard problems, pages 482–520,1996.
> >
> > [4] C. Gray, L. Mitchell, and M. Roughan. Generating connected random graphs. Journal of Complex Networks, 7(6):896–912, 2019.
> >
> > [5] M. Roughan, J. Tuke, and E. Parsonage. Estimating the parameters of the waxman random graph. In Algorithms and Models for the Web Graph: 16th International Workshop, WAW 2019, Brisbane, QLD, Australia, July 6–7, 2019, Proceedings 16, pages 71–86. Springer, 2019.
> >
> > [6] C. Wu, F. Wu, L. Lyu, Y. Huang, and X. Xie. Communication-efficient federated learning via knowledge distillation. Nature communications, 13(1):2032, 2022.
> >
> > [7] T. Li and L. Song. Privacy-preserving communication-efficient federated multi-armed bandits. IEEE Journal on Selected Areas in Communications, 40(3):773–787, 2022.

---

> > > ### Comment · Reviewer_a3u4 · 2023-08-10
> > > **Response to Authors**
> > >
> > > Thank you for taking the time to respond to my suggestions. I'm happy with the responses, references and the changes you plan on making.

---

> > > > ### Author Response · Authors · 2023-08-17
> > > >
> > > > Thank you for reviewing the responses and providing valuable feedback and suggestions. We will integrate the valuable suggestions and make the corresponding changes in the revised version of the paper.

---

### Author Rebuttal · Authors · 2023-08-09

We would like to thank all the reviewers for their valuable suggestions and comments. We have carefully considered the comments and provided additional perspectives accordingly. Thank you so much for your time and effort in reading the responses.

Additionally, in the uploaded file, we incorporate a table that presents the settings more clearly.

---

### Decision · Program_Chairs · 2023-09-21

**Decision:**

Accept (spotlight)

**Comment:**

This paper introduces an innovative approach to the decentralized multi-agent multi-armed bandit (MAMAB) problem, incorporating the study of heterogeneous rewards and time-varying random graphs with time-independent reward and graph distributions. Notably, the authors delve into both sub-exponential and sub-gaussian reward settings, an area that is currently underrepresented in existing literature.

All reviewers concurred that this paper represents a significant contribution to the field of decentralized MAMAB and have thus recommended it for acceptance. The authors are urged to incorporate all the points discussed during the rebuttal phase into the final version of the paper. This should include a comparison and discussion on how the regret depends on various problem-dependent variables.

Lastly, congratulations on this well-executed piece of work!